# Real-time interleaved spectroscopic photoacoustic and ultrasound (PAUS) scanning with simultaneous fluence compensation and motion correction

Geng-Shi Jeng [1,2], Meng-Lin Li [3,4], MinWoo Kim[1], Soon Joon Yoon [1], John J. Pitre Jr[1], David S. Li[5], Ivan Pelivanov[1✉] & Matthew O'Donnell [1]

For over two decades photoacoustic imaging has been tested clinically, but successful human trials have been limited. To enable quantitative clinical spectroscopy, the fundamental issues of wavelength-dependent fluence variations and inter-wavelength motion must be overcome. Here we propose a real-time, spectroscopic photoacoustic/ultrasound (PAUS) imaging approach using a compact, 1-kHz rate wavelength-tunable laser. Instead of illuminating tissue over a large area, the fiber-optic delivery system surrounding an US array sequentially scans a narrow laser beam, with partial PA image reconstruction for each laser pulse. The final image is then formed by coherently summing partial images. This scheme enables (i) automatic compensation for wavelength-dependent fluence variations in spectroscopic PA imaging and (ii) motion correction of spectroscopic PA frames using US speckle tracking in real-time systems. The 50-Hz video rate PAUS system is demonstrated in vivo using a murine model of labelled drug delivery.

[1] Department of Bioengineering, University of Washington, Seattle, WA, USA. [2] Institute of Electronics, National Chiao Tung University, Hsinchu, Taiwan. [3] Department of Electrical Engineering, National Tsing Hua University, Hsinchu, Taiwan. [4] Institute of Photonics Technologies, National Tsing Hua University, Hsinchu, Taiwan. [5] Department of Chemical Engineering, University of Washington, Seattle, WA, USA. ✉email: ivanp3@uw.edu

Nearly every object has a unique signature based on the optical properties of its molecular constituents. Consequently, optical spectroscopy is one of the most important analytic tools in all of science and technology. For biomedical applications, it can help quantify molecular components within complex solutions and structures based on each constituent's optical absorption spectra[1]. Spatially resolved optical spectroscopy is not used routinely in vivo because high tissue scattering typically limits ballistic photon penetration to a millimeter or less. Although recent progress in diffuse optical tomography (DOT) and 2D topography is encouraging, especially for functional brain imaging[2], this method faces fundamental challenges since image reconstruction is an ill-conditioned and under-determined problem and requires the detailed 3D structure of the object under study[3–5]. Overall, infrared spectroscopy and tomography methods can provide a few cm light penetration into biological tissue, but their spatial resolution is limited by optical scattering and remains very poor (8–10 mm)[3,4] compared to other clinical imaging modalities (PET, CT, MRI, and ultrasound (US)).

Photoacoustics (PA) has been proposed to overcome this barrier. As Alexander Graham Bell showed in 1881[6], optical absorption can be measured acoustically. PA methods were confined to laboratory devices until the 1990s when several groups leveraged modern pulsed laser technologies to image optical absorbers in vivo deep within highly scattering tissue[7–10].

The principle is straightforward[11,12]. Pulsed laser radiation diffusing within the tissue is absorbed by different structures, producing local heating proportional to the absorption coefficient. Through thermal expansion, heat generates an ultrasound transient (PA signal) propagating long distances. Signals are recorded at the tissue surface with an array transducer to reconstruct the absorber distribution. Because signal amplitude is proportional to the light absorption coefficient, molecular imaging within the tissue is an important potential feature.

For 25 years, PA imaging has been an active area and is now one of the hottest topics in biomedical optics. PA tomography (PAT) and microscopy (PAM) have numerous diagnostic applications[13–15], providing both endogenous[16–18] and exogenous molecular contrast in structural and functional images[18–20]. In particular, PA spectroscopy has produced functional images in the brain and heart preclinically (mostly in mice)[14,21,22], and molecular images using the unique spectra of nanoengineered contrast agents[19,22–24]. These tools have been extended to clinical breast cancer screening[25–28], skin lesion diagnosis[29,30], biopsy guidance[31,32], gastrointestinal imaging[33], and tumor metastases and lymph node screening[34].

Even with PA's remarkable success, validated clinical protocols, as well as system integration with clinical-grade probes, have been limited[35,36]. The PA imaging geometry has been highly optimized for small animals[37–39], with several commercial systems developed specifically for this application (https://www.ithera-medical.com/applications/preclinical-research/), (https://photosound.com/products/), (https://www.tomowave.com/products/lois-3d-pre-clinical-system/) These systems provide almost uniform illumination of a mouse body with nearly full-view signal detection, producing reconstructed PA images with spatial resolution approaching mathematical limits and minimum artifacts. Most of these conditions are hard to replicate for humans in routine clinical use for a very simple reason. In particular, a mouse cross-section is of the order of a few light penetration depths and can be easily surrounded by the transducer array. Illumination from all directions distributes light throughout the object, reducing the depth-dependence of laser fluence. The typical size of the human body is more than 40 light penetration depths and cannot be surrounded by the transducer array for efficient full-view reconstruction. Tissue illumination can be performed from one side only; thus, laser fluence will decay rapidly with depth. There is a fundamental difference in PA imaging of humans compared to mice, with the one major exception being the human breast.

The PA signal amplitude is proportional not only to absorption, but also to laser fluence (i.e., light level at a target)[11,12]. Because tissue attenuation depends on wavelength[40], the absorption spectrum estimated from a PA image can be very inaccurate (i.e., the shape can change dramatically and the wavelength of maximum absorption shift), especially deep within tissue (see Fig. 1a, c)[41,42].

Although compensating for wavelength-dependent light fluence is important even for the case of uniform multi-sided illumination, this is a much more serious problem for a single-sided imaging geometry, which is typical of the clinical environment. To reconstruct the true absorption coefficient spectrum at large depths, local wavelength-dependent light fluence must be compensated. Unfortunately, this requires a precise map of tissue optical properties, which cannot be measured or calculated during imaging. Several methods have attempted to estimate and compensate fluence variations[41–49], but none work well clinically where real-time or near real-time corrections are needed. In most cases, fluence is compensated using an approximate exponential function equalizing intensities. This can help, but measured spectra do not necessarily represent true molecular constituent spectra. This is especially true for spectroscopic measurements at depth determining blood oxygenation or the concentration of targeted contrast agents.

Tissue motion affects clinical spectroscopic PA imaging, where the same object must be probed at several wavelengths. Measurements at each wavelength require a unique laser pulse, with multiple pulses producing a spectrum. The repetition rate is determined primarily by the maximum permissible exposure (MPE) into the body[50]. When tissue is irradiated for more than 10 s, the maximum laser fluence can be 20 mJ cm$^{-2}$ at 700 nm wavelength, growing to 100 mJ cm$^{-2}$ at 1064 nm. Similarly, the maximum irradiance is 200 mW cm$^{-2}$ at 700 nm, growing to 1 W cm$^{-2}$ at 1064 nm. Thus, at the maximum fluence rate a 10 Hz repetition rate optimizes the fluence, i.e. maximizes the amplitude of generated PA signals.

As suggested in ref. [51], five wavelengths are needed for stable spectral decomposition with FDA-approved ICG (indocyanine green) contrast agents. Using five wavelengths yields a 2 Hz effective frame rate. Given typical physiologic motion, 2 Hz spectroscopic imaging has large artifacts. As illustrated in Fig. 1b, motion corrupts measurements of the local concentration of absorbers over the spectroscopic sequence, resulting in inaccuracies at best and total destruction of the spectrum at tissue interfaces (see "Results"). To avoid blurred images and inaccurate spectroscopic data, scan rates should be increased and/or frames aligned with motion correction. Previous efforts include respiratory or data-driven gating[52,53], model-based estimation[54], and tissue boundary tracking (e.g., skin surface)[55]. Gating-based methods typically reject images during large motion, slowing the effective frame rate and limiting accuracy for fast processes. Correcting motion, rather than rejecting it, to preserve spectroscopic frame rates has not been demonstrated.

Here we introduce a different approach (see Fig. 2) leveraging a unique diode-pumped wavelength tunable (700–900 nm) laser emitting about 1 mJ pulses at 1000 Hz, with wavelength switching in less than 1 ms for any arbitrary wavelength order (i.e., wavelength need not be sequentially stepped between bounds) (Supplementary Note 1). Thus, every pulse in a sequence can be at a different wavelength without sacrificing repetition rate. To maximize exposure, we illuminate with a narrow (~1 mm in diameter)

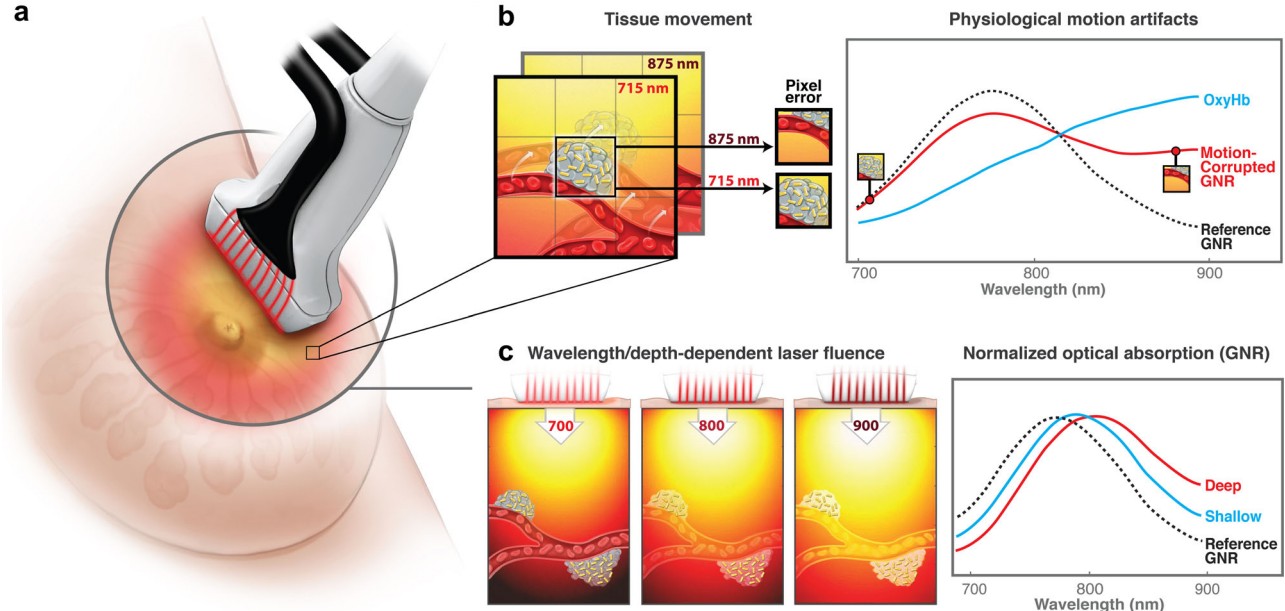

**Fig. 1 Effects of wavelength-dependent optical fluence variations and tissue motion on quantitative spectroscopic measurements. a** Schematic of conventional interleaved photoacoustic-ultrasound (PAUS) imaging: pulsed laser radiation irradiates the tissue simultaneously from all fibers surrounding the US detector, thus creating broad-beam illumination. **b** Spectroscopic PAUS requires sequential multi-wavelength illumination of moving tissue. As shown herein, tissue motion during spectroscopic acquisition can corrupt measurements of the concentration of different chromophores (blood and gold nanorods (GNR), for example) in an image pixel. **c** Wavelength- and depth-dependent optical fluence in tissue can strongly affect measurements of optical absorption spectra. In this example, the GNR spectrum changes with increasing image depth.

laser beam and switch it from fiber-to-fiber at 1000 Hz around the US probe (Fig. 2), resulting in one loop forming a single-wavelength frame in only 20 ms. The next loop uses another wavelength without delay; the procedure repeats overall wavelengths. In other words, we use fast-scanning (or fast-sweep) over the image area instead of broad-beam illumination over the same area. The laser is extremely stable and externally triggerable at a variable rate. Thus, it can be integrated with conventional US sequences, enabling PA measurements interleaved with all US modes (e.g., harmonic and color flow imaging and elastography) at 50 Hz frame rates for both modalities (optical delivery details are in Supplementary Note 2; the specific scan protocol is in Results and Supplementary Note 3). In addition, we dramatically reduced the laser footprint and its cost. Most importantly, we will show that the fast-sweep concept has significant advantages over conventional broad-beam illumination because it enables simple methods for wavelength-dependent laser fluence compensation and motion correction (Results). In addition, a novel off-axis fiber delivery system is developed as opposed to the clinically incompatible galvo-based scan approach[32] previously proposed by our group. Leveraging this configuration, we also present a theoretical framework to quantify wavelength-dependent fluence variation among different fibers (Supplementary Note 6). Moreover, here we demonstrate real-time implementation (parallel with data acquisition) of the entire data processing cycle, including data transfer, interleaved PAUS beamforming, laser fluence compensation and motion correction (Supplementary Note 3). We share experimental raw data and provide all processing routines in Supplementary Data and Software Library with detailed instructions on how to run the scripts so anyone can implement real-time spectroscopic PAUS imaging in future studies.

## Results

### Fast-sweep spectroscopic PAUS for laser fluence and motion correction. This method leverages recent developments in the laser industry. First, a diode-pumped wavelength tunable

(700–900 nm) laser was customized for fast-sweep imaging, in contrast to customizing the imaging system to the laser. It is compact (Supplementary Note 1), with a footprint potentially fitting within a cart-based ultrasound system. It emits pulses of about 1 mJ energy at a 1000 Hz repetition rate, with wavelength switching in less than 1 ms for arbitrary sequences. Thus, 1 kHz operation does not change between single-wavelength and spectroscopic approaches.

Unlike previous delivery systems coupling laser pulses into all fibers in a bundle simultaneously[13], we couple light into individual fibers sequentially (see Fig. 2a, b, d). Using a rotating wedge, the laser beam is projected onto a circle at the focus of a collimating lens. The wedge motor's absolute position controller synchronizes emission (i.e. coordinate of the laser spot on the circle) with the centers of 20 fibers in the bundle. With absolute position control, a precise rate is not needed for external laser triggering, ensuring maximal light delivery to each fiber. Motor speed variations only slightly alter the overall frame rate of 50 Hz.

Ten fibers are uniformly spaced along each elevational edge of the US imaging array (Fig. 2a). A custom-designed 15-MHz, 128-element US linear-array probe was integrated with the 20-fiber array to form the PAUS probe (Supplementary Note 2).

We integrated all controls, including laser pulse activation/sequencing, motor scanning, and PAUS image acquisition, with a commercial scanner (Vantage, Verasonics, WA, USA). The motor encoder triggers the US system to launch interleaved US and PA pulse sequences and the US system externally triggers the laser, synchronizing all sub-systems (Fig. 2a). Unlike triggering a scanner with a fixed rep-rate laser, externally triggering the laser with the scanner guarantees jitter-free synchronization by referencing both the imaging sequence and acquisition to the same clock. The scan protocol forming simultaneous PA and US images at a fixed wavelength is described in "Methods".

Note here that MPE limits[50] depend on the irradiation time. If a spot on the tissue surface is irradiated for less than 10 s, the irradiance can be increased. For example, for a 1 s exposure

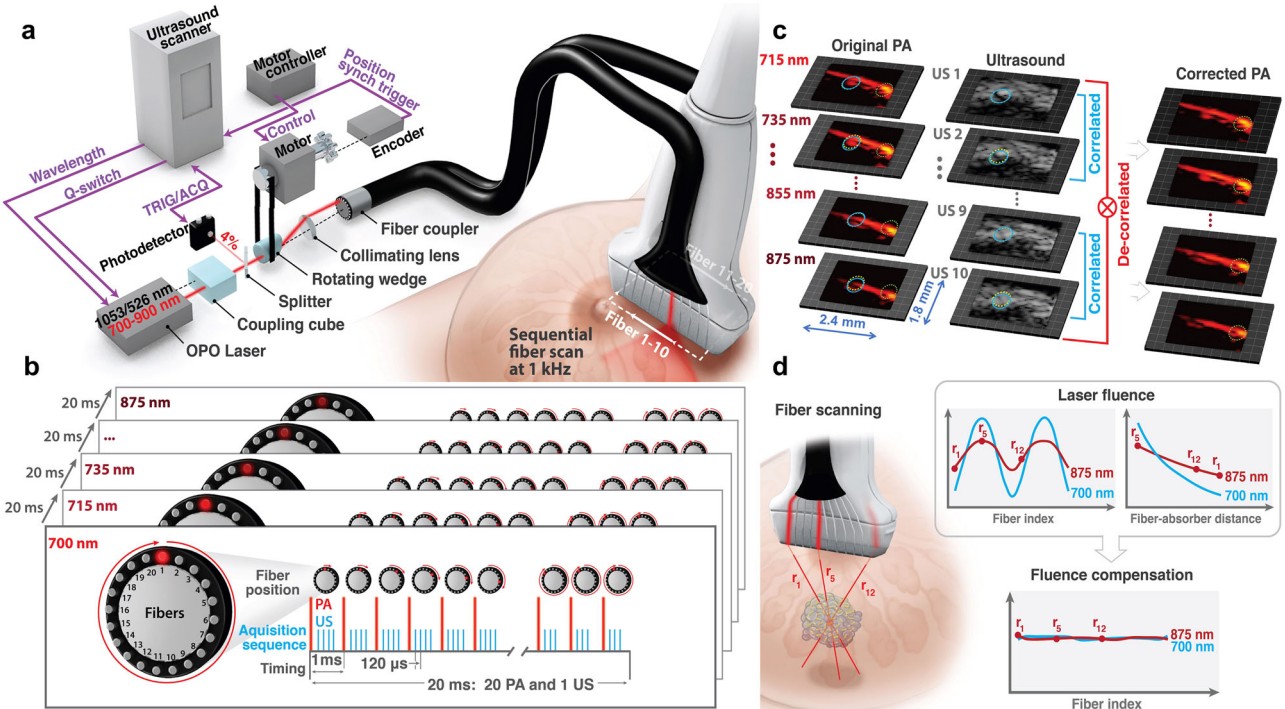

**Fig. 2 A 50 Hz frame-rate spectroscopic swept-beam PAUS system with real-time compensation for wavelength-dependent fluence and motion correction. a** The PAUS system includes a kHz-rate, compact, wavelength-tunable (700–900 nm) diode-pumped laser (Laser-Export, Russia), an integrated fiber delivery system (TEM-Messtechnik, Germany), and an US scanner (Vantage, Verasonics, WA, USA). The laser, externally triggered by the US scanner, emits pulses of about 1 mJ energy at a variable (up to 1 kHz) repetition rate, with wavelength switching times less than 1 ms. Thus, every laser pulse in a sequence can be at a different wavelength without sacrificing the kHz repetition rate. The integrated fiber delivery system includes 20 fibers arranged on two sides of a transducer array. The spinning motor rotates the laser beam over the ring, thus sequentially coupling laser pulses to different fibers while sending a trigger signal to the US scanner. Upon triggering, the US system initiates the interleaved B and PA imaging sequence by sending a trigger to the laser. **b** Timing diagram and pulse sequence for interleaved, spectroscopic PAUS. Ten wavelengths (i.e., 700, 715–875 nm every 20 nm) are used. For each wavelength, the laser beam irradiates tissue sequentially from 20 fibers, with several scanned, focused US beams interleaved for each laser firing. A frame composed of 1 B-mode and 20 PA sub-images is produced within 20 ms (50 Hz effective imaging frame rate). **c** The interframe tissue motion map is obtained using US speckle-tracking[56], then applied to all pixels of co-registered PA images. Blue and yellow circles show local motion between adjacent frames, whereas a green circle fixed in location clearly shows the efficacy of motion correction. **d** Light emerging from different fibers propagates different distances to a target. The amplitude of a partial PA image obtained for single fiber irradiation follows the dependence shown in upper left plot. Considering the distance from each fiber to a typical absorber in the imaging field, the PA amplitude follows the form shown in the upper right plot due to light absorption and scattering in tissue. The procedure is repeated for all wavelengths used.

(5 full, 10-wavelength, spectroscopic cycles of our PAUS system), the irradiance can be increased 5 times from 200 mW cm$^{-2}$ to 1.1 W cm$^{-2}$ at 700 nm. Thus, imaging in bursts can be used to increase the frame rate from 10 to 50 Hz, as proposed and demonstrated in this work. Because MPE depends on irradiation time, we maximize the frame rate to be 50 Hz by sequentially scanning fibers at a 1-kHz rate under permitted limits. When highly absorbing contrast agents are used, close illumination from successive spots may cause overheating on the tissue surface due to light and thermal diffusion. In this case, rearranging the fiber illumination sequence to 1, 15, 2, 14, etc. can be easily done at the entrance to the fiber coupler.

To enable stable spectral decomposition, 10 laser wavelengths (i.e., 700, 715–875 nm every 20 nm) comprised the spectroscopic sequence. It can be customized in number of wavelengths, number of pulses per wavelength, wavelength sequence, and spectral resolution. Wavelength spacing is arbitrary, including a variable pitch, with 2 nm resolution defined by the spectral line width. For noise minimized spectral estimates, we turned off (0% energy) the laser at 700 nm to estimate noise levels (Methods).

Interleaved data acquisition provides simultaneous anatomic (US) and PA images at a 50 Hz frame rate. This is sufficient for US speckle tracking[56] of individual pixels to map tissue motion between sequential images at different wavelengths (Fig. 2c). Motion can be compensated (Methods and Supplementary Note 9), as shown in the 3rd column of Fig. 2c, for all 10 wavelengths. After compensation, every pixel carries information from all wavelengths without motion artifacts and, therefore, enables spectral identification of molecular constituents. The images in Fig. 2c are in vivo data from a small animal. Even at a 50 Hz spectroscopic frame rate, pixel displacements can be about a millimeter whereas the pixel size is less than 100 µm.

Even with no motion artifacts, the PA image amplitude is still proportional to the product of light absorption and laser fluence, where fluence is a function of depth and optical wavelength in biological tissue. Here, we use partial PA images from every fiber to estimate laser fluence. Indeed, when light emerges from different fibers, it propagates different distances to a target. Figure 2d (upper left plot) shows how PA signal amplitude changes with fiber index. Converting fiber index to fiber-absorber distance, PA signal loss with distance due to light attenuation is shown in Fig. 2d (upper right plot). Note that fluence losses with depth will differ for different wavelengths. As shown in Methods, such measurements can drive accurate and robust mapping of laser fluence independent of the wavelength-dependent absorption curve for a specific absorber. After evaluation, fluence can be

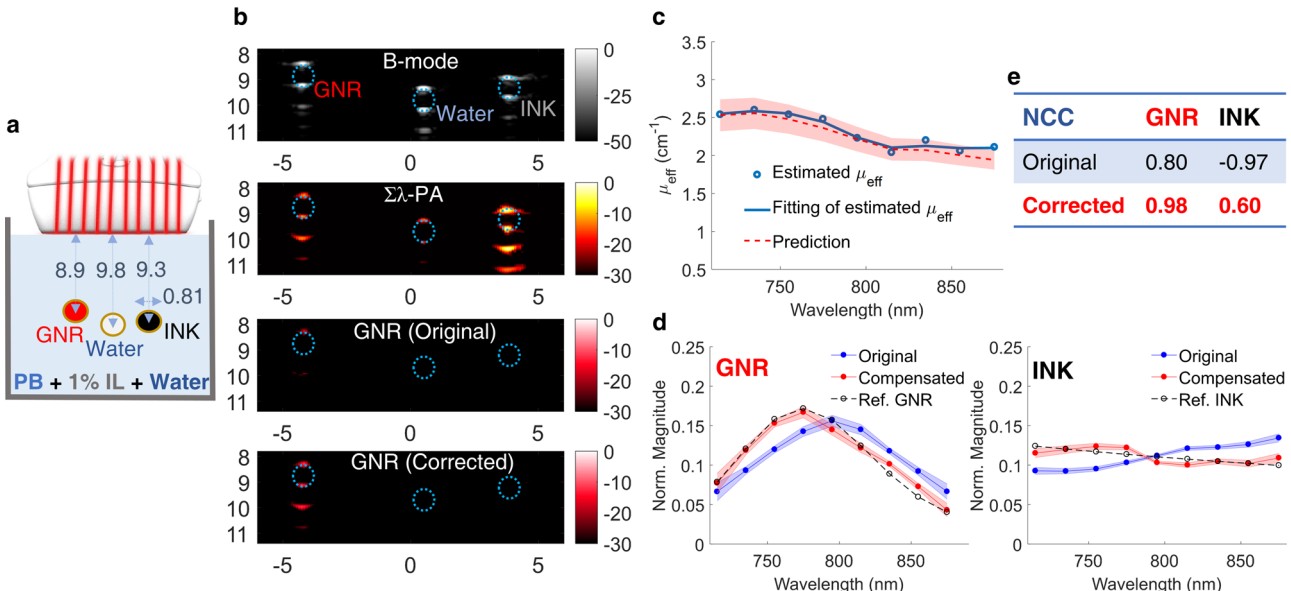

**Fig. 3 Phantom spectroscopic PA imaging with optical fluence compensation.** GNR (gold nanorods), de-ionized water, and black ink were injected into three different tubes immersed in an absorbing and highly scattering background (1% Intralipid suspension mixed with Prussian blue nanoparticles). **a** Experimental setup. **b** First row: B-mode image. Second row: wavelength-compounded, Σλ-PA, image. Third/Fourth row: GNR-weighted PA image without/with optical fluence compensation. The horizontal and vertical axes represent lateral and axial dimensions in units of mm, respectively. Wavelength-dependent fluence was estimated using PA signals at the top of the tube filled with black ink. **c** The estimated effective optical attenuation coefficient ($\mu_{eff}$, blue line) is compared to predictions using the UV–VIS measured absorption coefficient of Prussian blue aqueous solution and the reduced scattering coefficient of Intralipid known from the literature[58] (red shaded region). **d** Measured spectra (normalized to the area under the spectrum) of GNR (left) and black ink (right) are expressed as mean ± standard deviation. For each panel, the spectrum after fluence compensation (red solid line) is compared to the uncompensated spectrum (blue line) and the reference UV–VIS result (dashed line). **e** Improvements after fluence compensation for GNR and black ink solutions are quantified in terms of the averaged normalized correlation coefficient (NCC).

decoupled from the PA image to obtain the true light absorption spectrum of molecular absorbers. If fluence is ignored, accurate spectral decomposition is nearly impossible at large depths.

Two types of spectroscopic PA images are produced: wavelength-compounded images (referred to as Σλ-PA) and component-weighted images (Supplementary Note 4). Σλ-PA uses coherent summation over all wavelengths, resulting in improved SNR compared to individual wavelength images. On the other hand, component-weighted images are realized pixel-wise by the product of the Σλ-PA signal with the correlation of the spectrum after fluence compensation with the ground-truth spectrum of a molecular constituent (Methods and Supplementary Note 4), similar to other spectral decomposition approaches[57]. As a result, component-weighted imaging can differentiate an exogenous agent of interest from other absorbers with dissimilar spectra.

**Spectroscopic PAUS with compensation for wavelength-dependent laser fluence.** We conducted phantom experiments to test wavelength-dependent fluence compensation. Three identical polytetrafluoroethylene tubes were immersed (Fig. 3a) in 400 ml of 1% Intralipid solution (with known scattering[42,58]) with 0.47 ml Prussian blue nanoparticles (with known absorption) added to create an optical background with known wavelength-dependent properties. The first tube was filled with a solution of gold nanorods (GNR), the second with water as a control, and the third with Higgins black ink (Methods for more details and Supplementary Note 5 for measured spectra of all solutions).

US B-mode, Σλ-PA, and GNR-weighted PA images are shown in Fig. 3b. Given the transducer's limited view and bandwidth (Supplementary Note 2), both B-mode and PA images are only visible at the tube top and bottom. This artifact is well known[59] and is outside the scope of this paper. Other artifacts in both US

and PA images below the tube bottom are acoustic reverberations inside the tube. Finally, the walls of the water-filled tube produce weak PA signals, clearly demonstrating non-zero absorption at levels far below those of GNR and ink.

Due to wavelength- and depth-dependent fluence variations, the GNR-weighted PA image is poorly correlated with the true absorption spectrum (third row in Fig. 3b). The measured absorption spectra of GNR and Higgins black ink solutions (blue curves in Fig. 3d) look very different from the ground truth (dashed curves in Fig. 3d, Supplementary Figs. 5a, b). The GNR spectrum is significantly red shifted. Moreover, the ink spectrum is inverted from the ground truth; that is, its slope with respect to wavelength is the negative of the true slope. Note that these dramatic changes are at less than 1 cm depth, for effective light attenuation in the medium less than 3 cm$^{-1}$; that is, under optical conditions typical in humans[40]. Although this problem was widely discussed in the literature, no robust real-time solution has been proposed to date.

Leveraging the fast-sweep approach, we adopted a light diffusion model (Methods and Supplementary Note 6) to evaluate the laser fluence distribution from differences in PA image amplitudes (due to different propagation paths between a target and different fibers) across all fiber illuminations. Results obtained in different concentration Intralipid solutions validate that the background wavelength-dependent effective light attenuation coefficient, $\mu_{eff}$, can be accurately reconstructed and is consistent with reported results[42,60] (Supplementary Note 7).

We estimated the laser fluence distribution in the phantom solution using PA signals from the top of the Higgins black ink tube (Fig. 3b—right). The estimated wavelength dependence of $\mu_{eff}$ (Fig. 3c) is very similar to that obtained by combining Intralipid scattering[58] with the measured Prussian blue absorption spectrum (Supplementary Note 5). Using the estimated $\mu_{eff}$,

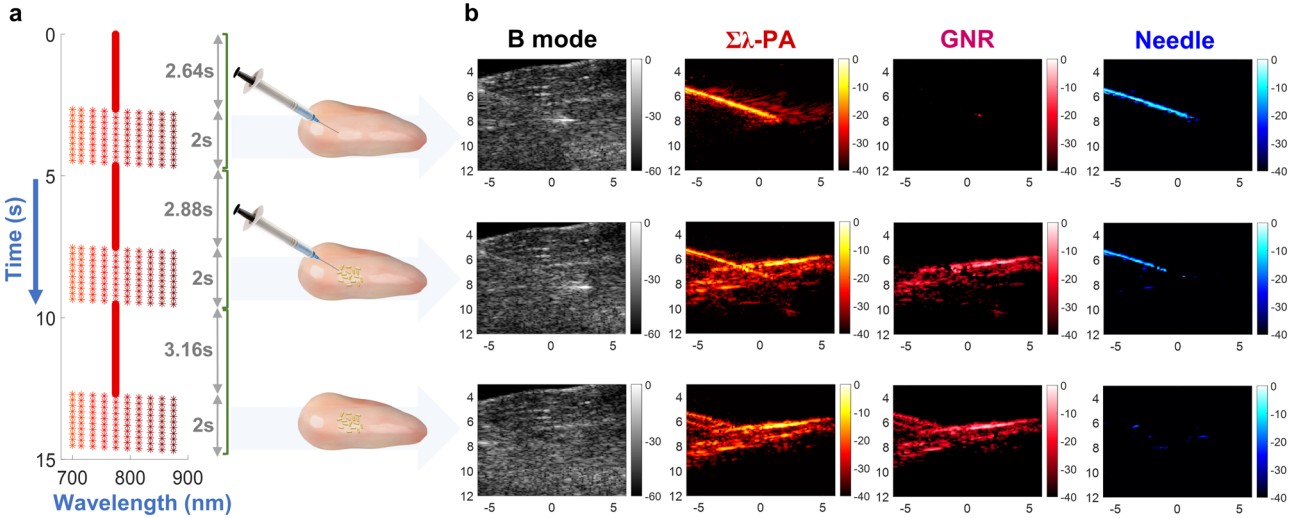

**Fig. 4 Ex vivo spectroscopic PA imaging for needle guidance with nanoparticle injection into chicken breast. a** Laser sequence and timing diagram for the following three operations: needle insertion into the chicken breast, GNR injection, and needle pullout. For each operation, the laser pulse sequence was designed to scan with a fixed wavelength of 775 nm, followed by 10 cycles sweeping over 10 wavelengths. **b** PAUS imaging results including needle insertion into chicken breast (top row), GNR injection (center row), and needle pullout (bottom row). Four simultaneous but different imaging modalities are compared, including: B-mode (left column), wavelength-compounded Σλ-PA (second column), GNR-weighted PA (third column), and needle-weighted PA (fourth column). GNR and needle-weighted PA images are produced by the product of Σλ-PA and measured spectra correlated with individual reference spectra. The reference spectrum of GNR was measured by UV–VIS whereas that of the needle was determined by PA measurements in de-ionized water (Supplementary Note 5). The horizontal and vertical axes represent lateral and axial dimensions in units of mm, respectively.

laser fluence compensated spectra are significantly improved (red lines in Fig. 3d), as evidenced by the increased normalized cross-correlation coefficient (NCC, defined in Supplementary Note 4) between corrected and ground-truth spectra (Fig. 3e), especially the NCC sign change for Higgins black ink. Compared to original Σλ-PA and GNR-weighted images, the compensated image completely preserves GNR signals and rejects those from all other tubes (Fig. 3b—fourth row).

**Ex vivo spectroscopic PAUS to guide interventional procedures.** Real-time US is commonly used for interventional procedures[61,62], often guiding drug injections to help visualize the needle relative to anatomy and deliver the drug to the desired target. The drug itself cannot be visualized unless the injection creates bubbles. Such bubbles typically disappear quickly, and the distribution of the drug is not always clear. Additionally, it takes great skill to orient the US image plane relative to the needle since a specular reflection is used for visualization. Nevertheless, real-time US guidance of many interventional procedures is a rapidly growing field that could expand greatly by overcoming these limitations.

PA guidance of needle injections has also been demonstrated[32,63]. Because the PA signal is quite independent of needle position relative to the transducer, the image plane need not be precisely oriented, potentially making the technique more accessible. PA spectroscopic imaging can also add a molecular dimension because therapeutic agents such as drugs can be molecularly labeled. Many small animal studies have shown the potential of spectroscopic PA molecular imaging[22–24,57]. Nevertheless, these methods have not translated well into clinical tools.

Here, we demonstrate how fast-sweep PAUS provides robust imaging for interventional procedure guidance using a simple example of GNR injection ex vivo (chicken breast—Fig. 4). This image-guided procedure has three sequential steps: (i) needle insertion into tissue, (ii) injection of a GNR solution, and (iii) needle pullout. A custom pulse sequence was developed (Fig. 4a),

where PA images at different wavelengths are interleaved with real-time US image acquisition.

For the first 132 frames, a fixed 775 nm wavelength helped guide initial needle insertion. After full needle insertion, multi-wavelength operation commenced repeating 10 wavelengths over 10 cycles. The same sequence was replicated during GNR injection, where the variable wavelength component started after all nanoparticles were delivered. Finally, the needle was removed, and the same sequence repeated. Supplementary Movie 1 presents a video of the entire experiment. The laser repetition rate was about 1000 Hz without any breaks for wavelength switching, producing complete PA and US images for each wavelength at a 50 Hz frame rate.

Both Σλ-PA and component-weighted images were reconstructed (Supplementary Note 4). Additional details on both laser fluence compensation and motion correction are in Supplementary Notes 8 and 9, respectively. Wavelength-compounded Σλ-PA images have the best SNR, combining all 10 wavelengths over the spectral range. However, this format does not display specific molecular constituents and can contain artifacts. Using spectral decomposition, the PA image of a specific molecular constituent can be displayed at high SNR[22–24,26,57]. An alternate approach is to correlate the light absorption spectrum at every pixel with the spectra of molecular constituents in tissue. This correlation-based method does not require numerical minimization (i.e. inversion), which is very sensitive to background absorption and noise[57,64]. It solves the forward problem, which, by definition, is more stable.

The upper row of Fig. 4b shows PA images (with motion correction and fluence compensation) after full needle insertion but before injection. The Σλ-PA image clearly shows the needle, but some additional bright spots are also present. It has high SNR because it coherently combines all 10 wavelengths over the spectral range; however, it is not specific to molecular constituents and contains artifacts. When spectrally correlated with the GNR spectrum, the PA image shows nearly nothing over a 40 dB dynamic range. Indeed, nanoparticles had not been injected yet. The needle spectrum correlation (Fig. 4b—upper right) clearly shows the needle with few artifacts.

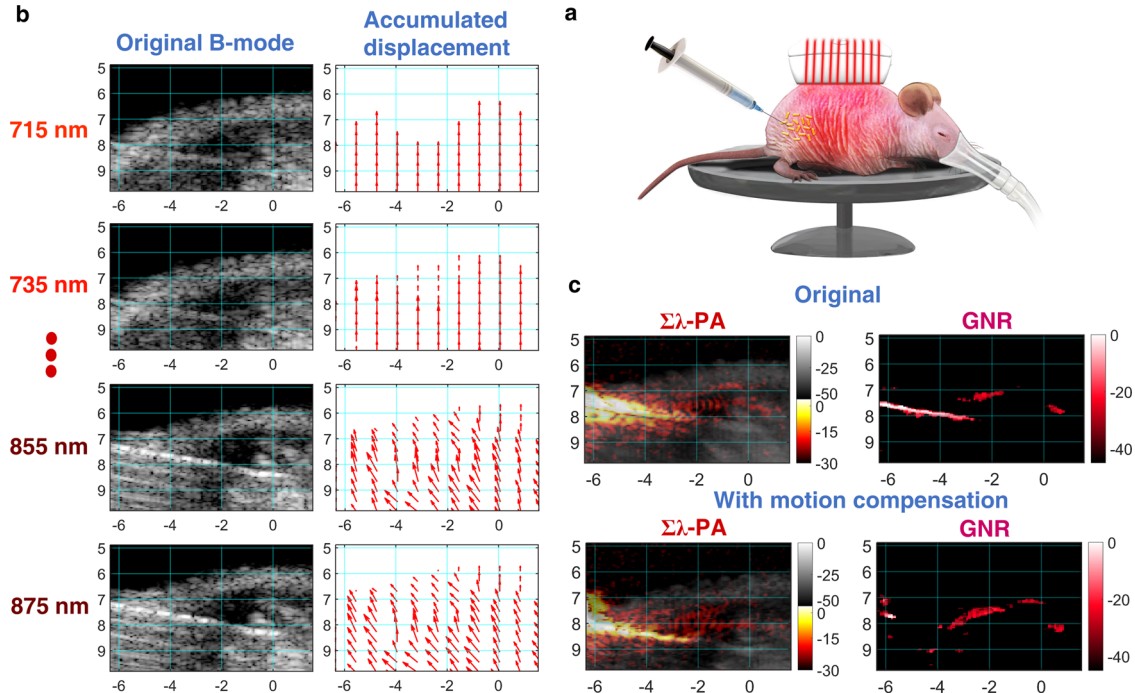

**Fig. 5 Case #1 of in vivo spectroscopic imaging for GNR injection into a mouse right leg muscle.** Effects of motion artifacts on quantitative PA measurements are illustrated. **a** Experimental setup. **b** Inter-wavelength motion artifacts during nanoparticle injection (See movements in B-mode images with wavelength). Inter-wavelength motion vector is estimated from two successive B-mode images using PatchMatch-based speckle tracking, and then accumulated over each wavelength sequence to produce the final motion-corrected PA images. For easy visualization, vectors are spatially decimated. **c** The original uncompensated wavelength-compounded Σλ-PAUS (left) and corresponding GNR-weighted PA (right) images. Motion artifacts corrupt GNR-weighted PA images as the needle is not rejected. **d** The needle is completely rejected in GNR-weighted PA images and GNR detectability is also improved in motion-corrected images.

The middle row of Fig. 4b shows PA images after injection. Additional signals are evident in the Σλ-PA image. Although the PA image differs greatly from that before injection, the B-mode image is nearly identical, demonstrating how poorly US monitors injections. Component-weighted PA images clearly differentiate the needle from GNR.

Finally, when the needle is removed (bottom row in Fig. 4b), the Σλ-PA image is almost identical to that of the GNR-correlated one (second and third columns, respectively), and no needle is observed. It is interesting that the needle pullout leaves a trace of GNR in the needle channel.

**In vivo spectroscopic PAUS in a small animal model.** PA spectroscopic imaging has been extensively studied in small animal models[14,16,19,22–24,57]. However, small animals greatly simplify imaging conditions. The transducer array can surround the animal, recording the PA signal with large spatial and temporal bandwidth for accurate PA reconstruction. Such conditions are very difficult to duplicate for most human applications, with breast as the notable exception.

In the last two sections, we addressed wavelength-dependent fluence variations. Here we use a small animal model to address tissue motion, the second major limitation on clinical translation. The specific mouse model is described in Methods. The GNR solution was injected into the mouse's right leg muscle using the same protocol described for ex vivo studies above. In particular, the laser pulse sequence was scanned at 775 nm during needle insertion, followed by incrementally sweeping 10 wavelengths over 10 cycles during GNR injection (Supplementary Movie 2).

Pixel-wise estimates of motion vectors from real-time US images (Methods and Supplementary Note 9) show that motion differs from one pixel to another (Fig. 5b) and changes during the

imaging sequence. Motion artifacts blur the Σλ-PA image (top left panel in Fig. 5c). Furthermore, the needle is not removed from the GNR-weighted PA image (Fig. 5c—top right panel). After motion compensation, the Σλ-PA image is highly improved, but it is still not clear if any GNR particles are present. The motion-compensated GNR-weighted image does not contain the needle (right bottom panel in Fig. 5c) and the sensitivity of the GNR-weighted PA image is greatly improved, with more GNR particles clearly detected.

In another example, more GNR particles were injected for easy visualization. PA images at individual wavelengths are presented in Fig. 6a, as well as motion-compensated Σλ-PA (Fig. 6b, left panel) and motion-compensated GNR-weighted (Fig. 6b, right panel) images. Interestingly, not all bright points in the Σλ-PA image appear in the GNR-weighted image.

After motion compensation, the measured GNR spectrum closely matches ground truth and, therefore, no fluence compensation is required. However, the correct spectrum cannot be obtained without motion compensation. Unlike small animal studies, human imaging includes both significant scattering and physiologic motion. Clearly, fast-sweep PAUS addresses two significant barriers to clinical translation of spectroscopic PA imaging at the expense of limited view and bandwidth. Indeed, fast-sweep PAUS trades off PA image quality for spectroscopic identification of an injected agent using US-based methods to guide interventional procedures.

**Discussion**

Real-time US is commonly used to guide many interventional procedures. Indeed, more than two million patients in the USA are treated annually with US guidance, helping reduce procedural errors and costs[65]. For injections of therapeutic agents; however, the agent itself cannot be directly visualized and procedure

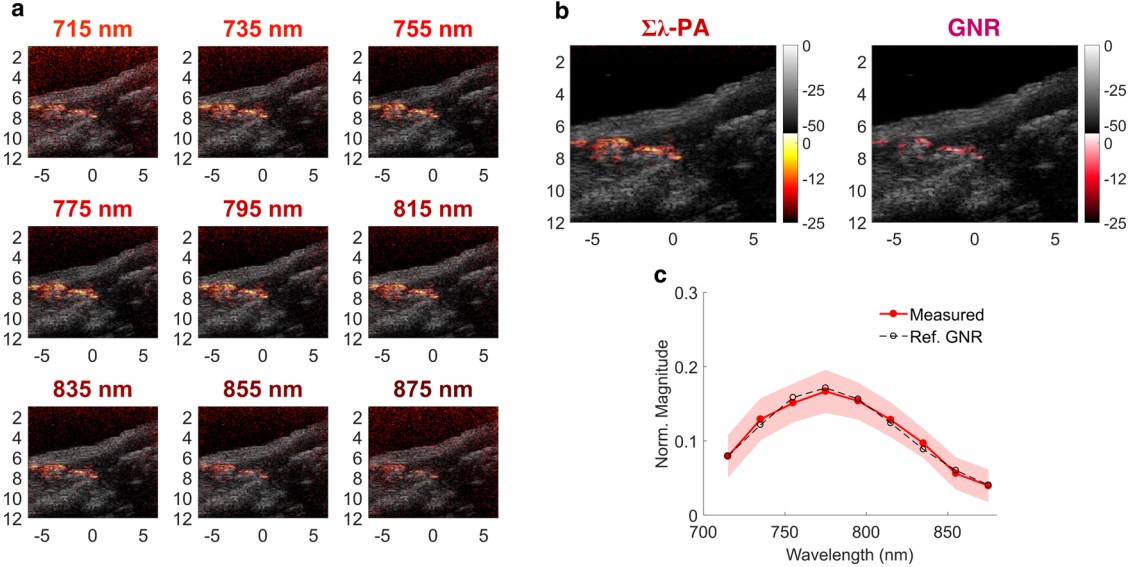

**Fig. 6 Case #2 of in vivo spectroscopic imaging for GNR injection into a mouse right leg muscle.** Quantitative GNR spectral measurement after motion compensation is illustrated. **a** Multi-wavelength PAUS image from 715 to 875 nm. **b** Left: wavelength-compounded $\Sigma\lambda$-PA image; right: GNR-weighted PA image. **c** Measured spectra (normalized to the area under the spectrum) of PA signals larger than −15 dB are expressed as mean ± standard deviation compared to the UV–VIS result (dashed line).

outcomes cannot be assessed. Here, we proposed integrating real-time spectroscopic PA imaging with the US to add a molecular dimension to procedure guidance, helping visualize molecularly labeled substances such as drugs and monitoring procedure outcomes. We explored real-time spectroscopic PAUS imaging of injected nanoparticles mimicking this extremely important medical procedure. Although the current implementation of the fast-sweep PAUS system cannot be considered a clinical instrument, the tools and solutions embodied in this system should greatly simplify its clinical translation.

Spectroscopic PA imaging systems using bulky solid-state lasers are difficult to translate clinically for many reasons. Practically, their size and cost limit easy integration with clinical US. The fast-sweep system can be integrated into the footprint of a conventional clinical US scanner and easily interleaved with all US modalities. Both motion correction and fluence compensation can be included in spectroscopic PA imaging without sacrificing real-time operation (see Supplementary Notes 3, 6, and 9, and also Supplementary Data and Software Library for real-time scripts). Overall, fast-sweep PAUS can improve clinical workflow for many US applications by adding spectroscopic identification and monitoring of labeled agents such as drugs to all modalities, including B-mode, ultrafast and harmonic imaging, color Doppler, and shear-wave elastography.

Compensating wavelength-dependent laser fluence is key to fast-sweep PAUS. The PA signal is proportional to both the local light absorption coefficient and laser fluence. This effect may be not critical for some small animal models probed with nearly uniform illumination and imaged at a shallow depth, but in vivo human measurements with single-sided illumination require fluence compensation for PA spectroscopy. Indeed, a mouse body cross-section is only a few light penetration depths and can be easily surrounded by the transducer array. Illumination from all directions reduces the depth-dependence of laser fluence[19,39,40]. The human body is more than 40 light penetration depths. Tissue illumination can be performed from one side only for the vast majority of applications; thus, laser fluence will decay rapidly with depth.

Without compensation, PA image spectra can be markedly different from true spectra (Fig. 3). Indeed, for the model system presented here, the GNR spectrum was significantly red shifted

and the black ink spectrum slope even changed sign. This can lead to erroneous conclusions about molecular contributors to the PA signal. With it, spectroscopic PA images can be decoupled from wavelength-dependent fluence variations, helping identify constituents based on known optical absorption spectra (Figs. 3, 4). Because all pixels with amplitudes exceeding the noise floor can contribute to fluence estimates, this procedure is almost guaranteed to be stable for optically macro-homogeneous media (Supplementary Notes 6-8 for details).

For over twenty-five years, the nearly unique properties of US speckle have been exploited for dense estimates (i.e., full displacement vector at every pixel) of tissue motion. Using a speckle-tracking algorithm appropriate for real-time use, dense displacement fields were estimated from interleaved US images at the 50 Hz frame rate. Since both modalities use the same array, US and PA pixels are co-registered. Thus, US-derived displacements can correct PA images for interframe motion (see Supplementary Note 9), aligning PA images from all wavelengths in a sequence.

For our 10-wavelength sequence, motion artifacts are clearly very serious in vivo (Fig. 5). For high spatial resolution (i.e., pixel-wise) spectroscopic imaging, motion corrupts spectral measurements, which cannot always be recovered using spatial averaging. For instance, detecting multiple molecular constituents or separating exogenous agents (like molecular-targeted nanoparticles) from endogenous absorbers (like blood) is challenging if motion is not properly corrected.

Motion compensation may also help tackle limited PA penetration (and, therefore, typical low PA SNR). Without considering the motion, signal averaging will not significantly increase SNR and will spatially blur spectroscopic information. With it, however, multiple frames can be averaged to greatly enhance SNR and increase image depth. In addition, motion compensation may be very important for fast processes, as often encountered in interventional procedures. Although motion artifacts can sometimes be rejected in small animal models[52], they must be addressed in clinical imaging.

Motion estimation and compensation have also been used to distinguish deep PA signals corrupted by a systemic image background (e.g., bulk tissue absorption), where induced motion is leveraged to distinguish deep signals from the background[66].

Although effective for increasing sensitivity at large depths, this approach has not been used for quantitative spectroscopic measurements.

The current system has been reprogrammed to produce US images at a 5 kHz frame rate (plane wave imaging[67]) but with image quality markedly reduced from the current approach. Hybrid sequences can be developed to trade-off image quality with frame rate, providing robust tracking for any significant physiologic motion. If faster rates are needed, then dense motion fields can be interpolated to any time and space point to compensate motion, even for PA sub-images acquired at the same wavelength but with different fibers.

Building on motion correction and fluence compensation results, we proposed two PA modalities: Σλ-PA and component-weighted images (see Figs. 3–6). Wavelength compounding improves PA sensitivity whereas component weighting improves specificity. Conventional decomposition uses spectral inversion of all known molecular absorbers in the medium. It may be unstable even for fluence compensated PA images due to typically low SNR and image artifacts. In contrast, we use the correlation of the measured absorption spectrum with that of a known component. We used component-weighted imaging to identify GNR and a needle (see Fig. 4). For multiple molecular constituents, it can be performed for every constituent. Because this procedure is correlation-based and does not use inversion, we believe that it can be more stable. We note, however, that correlation cannot yield absolute constituent concentrations. Possibly both methods can be combined whenever absolute concentrations are needed.

Other fluence compensation methods have been developed that may complement the approach presented here. In particular, eigenspectra PA tomography[68] can assess tissue background oxygenation. Although it was not demonstrated for a limited view/bandwidth geometry with strong exogenous optical absorbers such as needles or contrast agents present, we believe that it can be tested for our geometry with some modifications applied. It would be very interesting to combine our method with that proposed in[68] for both background and contrast agent spectroscopy.

An additional advantage of the patterned multi-fiber illumination proposed in this study is a strong reduction of clutter artifacts. Clutter[69] is usually dominated by US signals generated in skin. They propagate within the tissue, mixing with PA signals and creating strong artifacts. In our case, 'clutter' US beams are generated at a 6 mm elevation distance above the imaging plane (see Supplementary Fig. 2). Given the transducer frequency response, clutter beams never reach the image plane by diffraction[70] at distances less than 60 mm from the tissue surface.

Although fast-sweep spectroscopic PA imaging has significant advantages, it also has limitations. As noted, the probe's limited view and finite bandwidth produce image artifacts, especially for large objects with uniform absorption[59,71]. Recently, a deep-learning image reconstruction developed for the fast-scanning PAUS system has been shown promising for mitigating such artifacts[72]. The second issue is that the small footprint of individual laser firings reduces SNR compared to broad illumination. For the sequence used here, the SNR is reduced approximately by the square root of the number of fibers. That is, the current system has approximately 13 dB lower SNR compared to broad illumination with a 50 Hz high-power laser delivering the same surface fluence. Because of our high frame rates and laser stability, however, SNR can be recovered with averaging. For example, a Σλ-PA image can recover nearly 9 dB. With motion compensation, longer averaging periods can also improve SNR.

Previous PA imaging approaches using kHz-rate light sources[73,74] were demonstrated only for a single (or a few) wavelength. To improve SNR, simple signal averaging with broad-beam illumination was applied, leading to dramatic (equal to $\sqrt{N}$, $N$—number of pulses to average) overall SNR reduction compared to conventional high energy, 10–50 Hz rate system. In addition, PA and US modalities were not synchronized in those systems to enable motion correction.

Given its advantages and limitations, real-time spectroscopic PAUS imaging is appropriate for many clinical applications but is not appropriate for some, such as deep imaging within relatively high scattering tissue where SNR is a significant concern. There are many potential clinical applications, but two obvious short-term targets are guiding interventional procedures, such as the needle injections presented here, and monitoring the patency of full thickness skin grafts.

US-guided needle-based procedures are challenging because alignment is difficult, therapeutic agents delivered through the needle cannot be visualized, and US contrast agents delivered simultaneously are short-lived and can only help confirm the delivery site. On the other hand, both drugs and cells can be labeled with FDA-approved PA contrast agents such as ICG and methylene blue (and many others) for procedure guidance. These agents are molecular, so they persist for long periods (hours to days) to help monitor drug/cell migration. Thus, real-time spectroscopic PAUS imaging can not only guide drug/cell delivery, but also monitor diffusion and migration over long periods and correlate movement with outcomes.

Finally, although motion artifacts and wavelength-dependent fluence variations are the largest technical hurdles to clinical adoption of spectroscopic PA imaging, there are many other challenges, as detailed in[13]. In addition, regulatory issues, educating personnel, and high PA system cost are additional barriers to widespread clinical translation.

## Methods

**Fast-sweep PAUS imaging system and image reconstruction**. The real-time spectroscopic PAUS imaging system contains: (i) a compact, high pulse repetition rate (from single shot to 1000 Hz), wavelength tunable (700–900 nm) Ti:Sapphire laser (Laser-Export, Moscow, Russia, Supplementary Note 1); (ii) a fiber delivery system (TEM Messtechnik, Germany) integrated with (iii) a 15-MHz, 128-element US linear-array probe (Vermon, France, Supplementary Note 2); and (iv) an US scanner (Vantage, Verasonics, WA, USA). The laser can operate at rates up to 1 kHz, support an external trigger, and switch operating wavelength between laser shots (i.e. switch in less than one ms). Thus, complicated pulse trains can be programmed with arbitrary wavelength sequences and variable number of pulses, pulse energy, and repetition rate.

A new, fast-sweep scanning approach (Fig. 2a) was implemented. An optical wedge aligned in the laser beam and positioned close to the focus of a convex lens is rotated by a stepping motor at approximately 50 Hz, generating a circularly rotating beam behind the collimating lens. A fiber coupler, consisting of 20 multimode fibers with a 550 μm core diameter each secured around a 5.5 mm diameter tube with a fixed distance between fibers, was placed at a fixed distance from the lens. A position encoder within the motor triggered the Verasonics scanner, which in turn triggered the laser. Thus, independent of the motor spinning rate and its precision, the laser was triggered when aligned with fiber centers. The opposite end of the fiber bundle surrounded the US probe with a beam diameter of about 1 mm at a 1 mm distance. Ten fibers are uniformly spaced along each elevation edge of the US array (Fig. 2a, 1.5 mm fiber pitch, spanning 13.5 mm to cover the entire 12.8 mm lateral image range; Supplementary Note 2). Thus, unlike broad-beam illumination, the laser irradiated tissue with a narrow beam from each fiber but swept through all 20 fibers in about 20 ms, resulting in a 50 Hz effective PA frame rate.

As shown in Fig. 2b, for a fixed laser wavelength, the image frame contains 20 PA sub-images corresponding to sequential illumination over 20 fibers covering the complete lateral image range, and multiple ultrasound scan lines constructing the complete B-mode image per single-wavelength PA image. For each laser firing (Fig. 2b), corresponding PA signals are received by all 128 US channels and processed to form a PA sub-image, followed by several sequential US beams at successive lateral positions focused to the same depth on transmit and dynamically focused on receive. An integrated PAUS image is then produced by coherently summing all 20 PA sub-images to form one full PA image and interleaving it with the ultrasound B-mode image produced by combining all individual US scan lines. All signals were sampled at 62.5 MHz. Both B-mode and PA images were formed using coherent delay-and-sum beamforming, followed by envelope detection. The flexibility of our pulse sequence enables multi-beam acquisition (dual receive beams in this study) to maintain high US frame rates. By interleaving laser firings

with US pulse sequences, PA imaging can also be combined with other US modalities, such as color Doppler, harmonic imaging, and real-time elastography.

In addition to US B-mode images, we produced wavelength-compounded ($\Sigma\lambda$-PA) and component-weighted PA images (Supplementary Note 4). However, before generating these images, motion correction and laser fluence compensation were performed. Supplementary Note 3 graphically explains the specific pulse sequence used. Supplementary Notes 6 and 9 detail how fluence compensation and motion correction were implemented, respectively.

**Motion correction in fast-sweep PAUS.** Motion compensation, the first step in PA image reconstruction, is performed over each spectroscopic PAUS frame by estimating the relative displacement among 10 US B-mode images corresponding to PA images with 10 different wavelengths. The displacement between two successive B-mode images is computed and then accumulated relative to the first image in the composite spectroscopic frame using a recently developed speckle-tracking approach[56]. It is relatively efficient because it leverages a randomized search called PatchMatch[75] (Supplementary Note 9). Even with significant deformation over the entire multi-wavelength data acquisition interval greatly decorrelating speckle from sequence start to end, interframe correlation coefficients remain high and tissue motion can be accurately tracked over the entire interval. Notwithstanding modest SNR, displacements from US speckle tracking provide sufficient accuracy for robust image alignment. Estimated displacements are applied to each wavelength PA image before fluence compensation.

The specific motion compensation scheme used here processes US images acquired at a 50 Hz frame rate. This was sufficient to track physiologic motion in the present study. As shown in Supplementary Note 9, 2-D motion estimation is robust, and out-of-plane motion caused by respiration can be neglected at such high frame rates. It may be insufficient, however, for other applications including faster motion, especially near large pulsatile vessels. Fortunately, the current fast-sweep PAUS system can be programmed to interleave US images at frame rates greater than the 50 Hz PA frame rate to accurately track fast motion.

**Compensation for wavelength-dependent laser fluence in the fast-sweep approach.** The PA signal amplitude is proportional to the product of the local light absorption coefficient and laser fluence at an image pixel. As mentioned above, the fluence distribution in biological tissue varies with depth and wavelength, depending on many factors such as tissue absorption and scattering, the irradiation diagram, and boundary conditions[76]. Thus, for true spectroscopic imaging of molecular constituents, whether endogenous or exogenous, laser fluence spectral variations must be assessed and compensated in the PA image.

Most compensation methods rely on a priori knowledge of tissue optical properties (light absorption $\mu_a$ and reduced light scattering coefficient $\mu_s'$)[41–49]. They can be estimated ex vivo or even in vivo but require additional equipment and significant measurement time not compatible with real-time imaging[40,41,46,49].

We also considered laser fluence correction techniques that do not assume prior knowledge of tissue optical properties. Most rely on absorbing structures restored by full-view/bandwidth array detectors. Typically, these model-based schemes exploit the inversion of background tissue optical properties with approximated light transport models[77–79]. This makes them, however, very sensitive to the geometry of both the background and targets.

Deep-learning reconstruction has the potential to access more complex tissue structures and heterogeneous optical properties[80,81]. The challenge, however, is generating realistic training data/images. Even though successful in simulation, they have not been shown to broadly work for in vitro or in vivo studies.

Here, the spectral dependence of laser fluence is measured during PAUS imaging without additional equipment and delays. Because tissue illumination (see Fig. 2d) is performed sequentially with 20 individual fibers to form 20 partial PA images, the local amplitude of partial images is a function of fiber index, i.e. the distance between the fiber source and target (see Fig. 2d). The PA image contains multiple individual pixels and, therefore, the amplitude dependence on the distance between any pixel and the source can be obtained for many points with partial PA image amplitudes over the noise floor. These measurements provide inputs to fluence reconstruction.

Fluence reconstruction exploits the light diffusion model (Supplementary Note 6) of optically macro-homogeneous media, which has been shown to properly describe light transfer within turbid media at distances exceeding the transport mean free path $l_{tr} = 1/\mu_s'$. For most biological tissues, $\mu_s' > 2$ cm$^{-1}$ [40], so it is valid at imaging depths larger than ~5 mm; at smaller depths, however, fluence correction may be not required at all (see section "In vivo spectroscopic PAUS in a small animal model"). Using many points for fluence estimation stabilizes the minimization procedure (Supplementary Note 7).

Although it is hard to find an example of optically micro-homogeneous biological tissue, the presence of micro-heterogeneities is usually smoothed by strong optical scattering, and 'global' (or macroscopic) light transport is formed at depths exceeding a few transport mean free paths. These macroscopic tissue optical properties are usually reported in the literature[40]. For instance, direct measurements of light attenuation in muscle and liver tissues showed quite smooth exponential functions for light fluence profiles[82].

In PA measurements, when the optical fluence is estimated inside tissue at depths where light is fully diffused, a macro-homogeneous medium model can be

quite accurate to describe how much light is delivered from source to target. This model, however, does not cover the situation when a large blood vessel partially blocks the beam. For this case, fluence compensation algorithms can be adjusted by proper sorting of fibers participating in the reconstruction. For example, one side (opposite to the vessel) can be used for laser fluence reconstruction. Co-registered US B-mode can be used to label zones shadowed by large blood vessels so that their influence is taken into account. Using partial fiber sources, ten for example, should be sufficient for proper fluence estimation. Detailed analysis of this problem, however, is beyond the scope of this work.

Computation times for laser fluence compensation are mainly determined by the optimization search over a pre-defined 2-D $\mu_{eff}$ and $\mu_s'$ space for all laser wavelengths, which can be restricted according to underlying tissue optical properties. The search space can be significantly reduced by employing an initial $\mu_{eff}$ estimate based on Beer's law (i.e., pure attenuation decay). When the diffusion coefficient $D = 1/3\mu_s'$ is small, only $\mu_{eff}$ must be searched to accurately define the diffusion function (Supplementary Note 6). Due to the small number of parameters, a simple brute force method can be employed to compute all possible solutions and select the best match. Since all possible fluences can be calculated once and stored off-line, the only process required for real-time imaging is finding the best fit for measurements. Also, total computation time can be considerably reduced by parallel processing using a graphics processing unit (GPU).

We emphasize here that fluence estimation is performed for each wavelength in real-time, i.e. between PAUS single-wavelength frames (processing time is less than 20 ms, Supplementary Note 3). Additional details on fluence compensation, its validation in phantom measurements, and its performance in ex vivo studies are presented in Supplementary Notes 6-8, respectively. Matlab scripts and experimental raw data for fluence estimation are shared in Supplementary Data and Software Library.

**Spectroscopic PAUS modalities.** Spectroscopic PA information was acquired using 10 different laser wavelengths (700 nm, 715–875 nm with 20 nm step) over a scan cycle. At 700 nm, laser energy was set to zero to estimate the noise floor. For each wavelength, one PAUS frame contains one B-mode and 20 PA sub-images, corresponding to medium illumination from individual fibers. Signal processing for spectroscopic PA modalities ($\Sigma\lambda$-PA and component-weighted PA imaging) followed motion correction over the 10 different wavelengths for every pixel in the image area. By identifying strong absorbers in the PA image, wavelength-dependent optical fluence was estimated (previous sub-section and Supplementary Note 6). A wavelength-compounded image ($\Sigma\lambda$-PA) is produced by coherent summation over all 9 wavelengths, significantly increasing the SNR of individual wavelength PA images through signal averaging. To accurately estimate component spectra, the noise level estimated by turning off the laser at 700 nm is subtracted from the measured spectrum. The fluence compensated, noise-subtracted spectrum is correlated with the reference spectrum (e.g., absorption spectrum measured independently with UV–VIS) using cross-correlation and then further processed (flowchart in Supplementary Note 3 and Supplementary Note 4) to produce correlation-based PA images directly related to a single molecular absorber. Since a new wavelength image is obtained every 20 ms, spectroscopic imaging can be updated at a 50 Hz rate using the most recent 10 wavelengths after appropriate fluence and motion compensation for that set of measurements.

**Preparation of solutions.**

(a) *Suspension of Intralipid.* 20% IV fat emulsion (2B6022, Baxter Healthcare Corp., IL, USA) was used as the tissue-mimicking optical scattering medium to explore wavelength-dependent fluence estimation and compensation. It contained 20% soybean oil, 1.2% egg yolk phospholipids, 2.25% glycerin, and water. In this study, we diluted the Intralipid 20% solution into 1%, 2% and 4% solutions in v/v to investigate optical fluence variation with different concentrations.

(b) *Black ink solution.* The black ink solution (#44011, Higgins, MA, USA) was used in phantom spectroscopic PA experiments. Its optical characteristics were measured using UV–VIS spectrophotometry (Supplementary Note 5).

(c) *Prussian blue solution.* Supplementary Note 5 details synthesis methods and the optical properties of Prussian blue nanoparticles. The absorption coefficient of Prussian blue mixed with water was measured (Supplementary Fig. 5d). The solution used in Fig. 3 contained 0.47 ml Prussian blue nanoparticles mixed with 380 ml de-ionized water and 20 ml 20% Intralipid.

(d) *Gold nanorods solution.* 40-nm PEG-coated GNRs manufactured by NanoHybrids (Austin, TX, USA) were used. The localized surface plasmon resonance (LSPR) had a longitudinal peak at 776 nm, with an 80% width of 68 nm. Detailed properties and measured optical spectrum are in Supplementary Note 5 and Table 1.

**Small animal model.** An 8-week-old nude female mouse (nu/nu, strain code: 088, Charles River Laboratories, MA, USA) was used to test needle guidance with GNR injection under a well-defined protocol. The needle is 21 gauge with 0.82-mm outer diameter (21G1, BD, New Jersey, NJ, USA). During injection into the right leg muscle, the mouse was positioned on a heat pad to maintain body temperature and anesthetized with isoflurane. It was euthanized via $CO_2$ asphyxiation at the end of the

experiment. The study was performed with the approval of the Institutional Animal Care and Use Committee of University of Washington (Proto201600723:4211-03).

**Reporting summary**. Further information on research design is available in the Nature Research Reporting Summary linked to this article.

## Data availability
The authors declare that all data from this study are available within the Article and its Supplementary Information. We share some experimental raw datasets and provide all processing routines in Supplementary Data and Software Library with detailed instructions on how to run the scripts. The full package of raw datasets generated during and/or analyzed during the current study is available from the corresponding author on reasonable request.

## Code availability
All processing scripts are available in Supplementary Data and Software Library (Supplementary Note 10) with detailed instructions and examples on how to run the scripts.

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

## Acknowledgements

The authors would like to greatly thank Professor Martin Frenz at University of Bern for several helpful discussions about the method of laser fluence estimation, Matthew Bruce in the Applied Physics Laboratory at University of Washington (UW) for help with in vivo animal experiments and Yi-Ting Lee in the Department of Chemical Engineering at UW for help with UV–VIS spectrophotometer measurements. We also acknowledge Mesa Schumacher (Mesa Studios) for help in illustrating our article. We thank our vendors for customizing their products for our needs: Laser-Export from Russia for development of a unique wavelength-tunable laser and TEM-Messtechnik from Germany for development of the fiber-optic delivery system. Meng-Lin Li is supported by Chairman Tai-Shun Ho, Novatek Microelectronics Corp., the Ministry of Science and Technology, Taiwan (MOST-108-2321-B-002-058-MY2 and MOST 106-2628-E-007-002-MY3) and the Brain Research Center under Higher Education Sprout Project funded by the Ministry of Science and Technology and Ministry of Education, Taiwan. This work was partially supported by GE Healthcare and the Department of Bioengineering at University of Washington.

## Author contributions

G.-S.J. assembled the experimental setup, developed and implemented the PAUS imaging protocol, integrated the optical system with a Vantage (Verasonics) US scanner, performed all experimental studies, performed PA and US image processing and reconstruction, developed and implemented motion correction algorithms for spectroscopic PA imaging, wrote the paper. M.-L.L. conducted experiments with G.-S.J., helped in image processing, participated in designing laser fluence compensation algorithms. M.K. conducted experiments with G.-S.J., developed and implemented the laser fluence compensation routine in spectroscopic PA imaging. S.J.Y. assembled early-stage PAUS system, did preliminary measurements, helped in designing imaging protocol for the final PAUS scanner. J.J.P. Jr. participated in designing laser fluence compensation algorithms, fast acquisition and image processing in Verasonics US scanner and article illustration. D.S.L. synthesized Prussian blue nanocubes, helped in auxiliary measurements. I.P. conceived the idea of the spectroscopic fast-sweep approach, designed the project with M.O.D., supervised the project, specified the laser and fiber-optic coupling modules for the PAUS system, worked with vendors for their development, assembled the PAUS system, worked with M.K. on laser fluence compensation algorithms, participated in experimental studies, designed and wrote the paper. M.O.D. conceived the idea of moving beam illumination with I.P, designed the project, and wrote the paper.

## Competing interests

The authors declare no competing interests.
