## [Peer Review File · Nature Communications]

Reviewers' Comments:

Reviewer #1:

Remarks to the Author:

The paper describes the development of a new system for simultaneous photoacoustic and ultrasound imaging in real-time. The aim is to facilitate the clinical translation of the photoacoustic technology by capitalizing on the advantages of the suggested approach. The implementation of the suggested embodiment is indeed very challenging and includes new technological developments such as a fast tunable laser at 1kHz rate. However, the claims of the paper were not properly demonstrated, particularly regarding its potential clinical use. More specific comments are provided below.

The main advantage claimed by the authors is that the new system provides quantitative molecular imaging. Achieving quantitative results represents a long standing goal in photoacoustics and so far no sufficiently robust approach has been suggested. The authors focused on quantifying the bio-distribution (concentration) of extrinsically-administered contrast agents, particularly by correcting for the effects of wavelength-dependent light fluence. Light attenuation is indeed an important factor to take into account but there are other important factors also affecting quantification. For example, linear arrays are known to be strongly affected by so-called limited-view artefacts (as acknowledged by the authors) and have been shown to provide very sub-optimal photoacoustic images. The detection bandwidth of the array and the frequency-dependent directivity of the elements are also important factors that have not been taken into account. Also, the delay-and-sum reconstruction method used by the authors is known not to be quantitative.

The methodology to correct for light fluence attenuation has already been suggested by the authors and by other groups. I understand that the main novelty is the fact that this is achieved in a very short time (real-time). Real-time imaging (as claimed by the authors) implies image visualization during the acquisition procedure, which appears to still be very challenging. Acquisition of 128 photoacoustic signal is performed at 1kHz rate, and ultrasound pulse-echo is further interleaved. It is not clear how all these data are processed. It would be good that the authors provide a clear description on what is the amount of data acquired per second and how these data are transferred and processed. Also, the authors explicitly acknowledge that the computations times for light fluence compensation are still relatively large.

The experiments performed are not related to the potential clinical applicability of the system. Molecular imaging generally implies the use of a contrast agent that targets molecular pathways involved in a given disease. However, there are very few FDA-approved optical contrast agents and they are not targeted, hence molecular photoacoustic imaging does not appear to be a feasible approach. The authors use gold nanorods in the experiments, which have not been approved for clinical use, let alone ink as used in the phantoms. It appears that the only benefit of the multi-wavelength approach in the clinical setting would be oxygen saturation mapping. This has not been mentioned or discussed in the manuscript and probably would not be feasible as the authors acknowledge that the probe's limited view and finite bandwidth is not suitable for imaging the background.

The authors compare in Fig. 1 their system with other commercially available systems in terms of imaging rate. This makes the impression that their system is much faster than others. However, this comparison is flawed. For example, frame rates of 100Hz have been achieved with lasers that can be tuned on a per-pulse basis. This is faster than their system, which requires 20 ms to form a single-wavelength image (50Hz). It is then basically not true that no commercial system runs faster than 12Hz for spectroscopy (2Hz rate for 5 wavelengths). Imaging rates in the kHz range have also been achieved by several groups in the field, and there is also a commercially available system.

It is mentioned that the MPE limits are $20\text{mJ}/\text{cm}^2$ and $1\text{W}/\text{cm}^2$, so that the optimal SNR is

achieved at 50Hz. These numbers are not correct, the MPE limits are $20\text{mJ}/\text{cm}^2$ and $200\text{mW}/\text{cm}^2$. These values change with the wavelength but in any case the optimal SNR is achieved at 10Hz. Above this pulse repetition frequency (PRF), the energy per pulse must be reduced. Considering that the photoacoustic pressure is proportional to the pulse energy, it is preferable to use relatively low PRFs than to use higher PRFs and average multiple frames. The authors actually use a PRF of 1kHz and very low energy per pulse (1mJ at the laser output and less at the fiber output). It appears to me that the SNR achieved is clearly sub-optimal.

I have concerns related to the fluence compensation method. First, the method was only tested in phantoms with uniform absorption and scattering, which do not represent actual heterogeneous biological tissues. I understand that the light diffusion model employed requires some input of optical properties, which are generally unknown in vivo. As I mentioned above, the computations are also not performed in real-time, hence it is not correct to say that real-time imaging was achieved. I am also wondering on the effect of cluttering. For linear arrays with light provided from the sides, it is known that skin signals interfere with signals at depths. Considering that the illumination spot is changing, this may lead to frame-dependent cluttering effects that may affect the light fluence calculations.

The motion correction method, which is also claimed as a novelty, was not properly explained. First, this and other methods have already been previously reported. The novelty would be the real-time performance, but there is no discussion on the computation time required for motion correction. I am also concern on the fact that out-of-plane motion may be produced. This is generally the case if the system is used as a hand-held scanner and cannot be corrected. Considering the irregular displacement maps in Fig. 5 (875nm), one may assume that it is actually out-of-plane motion what is detected in this case.

Reviewer #2:

Remarks to the Author:

This paper presents a system for spectroscopic photoacoustic (PA) imaging that incorporates a fluence correction scheme and motion compensation. Both features are important for the successful implementation of PA spectroscopic methods. Although much of the underlying methodology is based on previously reported work, constructing the system and making it work in a practical sense for in vivo use represents a significant engineering advance. In terms of performance, the system appears to work in that it can visualise specific strong absorbers such as an injected bolus of gold nanorods (GNRs) or a steel needle in tissue in vivo. This suggests it could be useful for tracking needle based procedures although the extent to which it might find broader clinical application is unclear.

Overall the paper is well structured, provides sufficient detail and is clearly written. However, there are a number of points that require attention. Too often, over-generalisations and simplifications are made. In some cases these are factually incorrect, in others the situation is nuanced and qualification is required. In addition, the challenges relating to small animal and human PA imaging are often incorrectly stated or inadequately qualified.

1. The title refers to quantitative molecular imaging yet all the images appear to be scaled in arbitrary units. A quantitative molecular image would be one in which the image intensity represents the absolute concentration of the absorber of interest.
2. Abstract; it is stated that successful human trials have been minimal. Although relatively few fully validated and verified clinical studies have been completed to date (which is not surprising for a still emerging imaging technology), there have been a significant number of human studies that at least fall into the "technical validation" category. These include studies to assess the utility of PA

imaging for assessing breast cancer, skin cancer, lymph node disease, inflammatory arthritis, Crohn's disease, dermatological conditions and cardiovascular disease to name a few. More than minimal it would seem.

It is implied that the relatively limited clinical translation of PA imaging to date is largely a consequence of the inadequacy of current PA spectroscopic methods and motion related-errors that the current work addresses. This seems overly simplistic. There are many barriers to clinical translation, some technical (and not necessarily only those described in the current work), others relate to the limited availability of imaging systems, alignment with clinical workflow and regulatory or financial hurdles. That is not to say the motion and fluence correction issues addressed in the current work are not important. They are and their solution will certainly facilitate clinical translation in some areas. However, to imply they represent the primary barrier to clinical translation is a significant over-generalization/simplification.

3. Abstract; there is a suggestion that quantitative clinical PA spectroscopy requires that the issue of wavelength dependent fluence changes be addressed. Why clinical PA spectroscopy in particular ? The issue is a general one and applies to any application of spectroscopic PA imaging whether the target is a live mouse, ex vivo tissues or some other spectrally heterogeneous medium.

4. Line 32; It is stated that optical spectroscopy is not used routinely because tissue scattering limits penetration depth to 1mm or less. This statement is not correct. Clinical near infrared (NIR) spectroscopy is widely used to measure hemodynamic changes and blood oxygenation (M.Wolf, et al. J. Biomed. Opt. 12(6), 062104 (2007)); in its simplest form, the ubiquitous pulse-oximeter for measuring arterial oxygen saturation found in every hospital relies on optical spectroscopy. Moreover, if NIR light is used, tissue penetration can extend to several cm and is not limited to 1mm.

5. Line 52; It is stated that most previous studies have employed small animal models but the associated methods have not translated well to humans. The meaning of this statement is not clear. If an imaging system can successfully image a mouse, why would it not be possible to image human tissue, assuming of course the detection geometry was suitable (ie a linear or 2D array used in backward mode) ? Again this relates to the point made previously about clinical translation. Just because previous studies have used mouse models rather than human subjects, it does not necessarily follow that the imaging performance is not suitable for human studies. There are many reasons why a system suitable for small animal imaging might not have been applied to human studies, including lack of inclination or opportunity on behalf of the investigators...

6. Figure 1; Shouldn't the comparison be between the current state-of-the-art irrespective of whether it originates from industry or an academic research lab ? It could be argued that commercially available systems will generally offer lower imaging performance than the latest experimental system developed in a university lab (as is the authors' system) so the comparison is not entirely fair.

7. Lines 82 - 89; The confounding influence of the wavelength dependent fluence on the quantification of absorbers using spectroscopic methods is discussed. It is good to see this so clearly stated as it is a crucial point. However, it is not a problem limited only to clinical PA imaging but applies equally to imaging mice or rats.

8. Line 90: it is stated that tissue motion affects clinical spectroscopic PA imaging. Again, why "clinical" specifically ? If a live mouse or rat is being imaged, then respiratory and cardiac induced motion will be present so wouldn't this also be similarly problematic ? Indeed the authors illustrate this nicely in figure 6 by evaluating their motion tracking method by imaging a mouse in vivo !

9. Line 94: the MPE usually depends on wavelength and total exposure time – shouldn't these be quoted as well ? Also, it is stated that the optimal SNR occurs at 50 Hz. Please provide a

justification for this claim.

10. Line 95; it is stated that at least 5 wavelengths are required for stable decomposition. This statement needs qualifying as it appears to be based on one particular study using a specific approach in a particular experimental situation. The minimum number of wavelengths depends on a variety of factors including the nature of the inversion scheme, noise levels and the wavelength range in which the wavelengths are chosen. Based on simple notions of uniqueness however, it can be expected to depend on the number of spectrally distinct chromophores in the tissue. So in the current study, if the principal absorbers are oxyhemoglobin, deoxyhemoglobin and the GNRs, then three wavelength should be sufficient, four perhaps if scattering is considered.

11. Line 96 – why limit the comparison to commercial systems only ? If there is a non-commercial system that provides a higher frame rate, this should be cited as well.

12. Line 99 – it is stated that “motion changes the local concentration of absorbers”. It is suggested that this sentence is re-worded (also the caption in Fig1(b)). Clearly motion itself has no effect on concentration but presumably it is meant that it corrupts the measurement of the concentration.

13. Line 108 – detailed specifications of the laser are provided in the supplementary information, rightly so as it is a unique and critical component of the system. However, it does not state how the wide tuning range is achieved. It is stated that it is a diode pumped device but what is being pumped, OPO, Ti-Sapphire or some other medium ? Please provide a brief explanation.

14. Lines 200-213; The fluence correction approach relies on measuring the PA signal generated in a local absorber (eg a well defined blood vessel or some other discrete absorber) as a function of the light propagation distance. In practice, this measurement is achieved by delivering the excitation light sequentially through a linear array of optical fibres. The success of this method is critically dependent upon a number of assumptions, none of which appear to be stated explicitly. First, the background optical properties of the target tissue must be homogenous, at least over the entire tissue volume that is irradiated. Second, the local absorber must not itself perturb the fluence distribution. These assumptions need to be stated and justified, especially the assumption of spatial homogeneity. The latter is straightforward to achieve in a phantom but will not always apply in tissue. For example a superficial blood vessel located at one end of the fibre array might only be traversed by the photons emitted by a small proportion of the optical fibres which would introduce errors in to the fluence correction.

15. Line 210; “If fluence is ignored, accurate molecular imaging based on spectral decomposition is nearly impossible. We believe that this is especially true for imaging in humans, although we note that for mouse imaging fluence correction can sometimes be ignored.” The first sentence is true and a crucial point. However, why is it “especially true” in humans ? It is equally true when imaging mice or rats since their tissues contain essentially the same spectrally distinct chromophores as humans (oxyhemoglobin, deoxyhemoglobin, water, lipids, melanin etc) that give rise to the wavelength dependent fluence variations that corrupt the measured PA spectra and confound spectral decomposition.

16. Lines 214 – 221 and supplementary note 4; The terminology relating to the spectral decomposition method is unclear. It appears that the component weighted images are essentially images that have been spectrally unmixed to reveal the presence of a specific absorber, the GNRs in most cases. However, this is referred to as the “projection” of the measured PA spectrum on to the known absorption spectrum of the absorber. This seems a confusing way of describing it. It might be clearer to state that the multiwavelength image data were spectrally unmixed or decomposed to provide the GNR (or other component) weighted images. Although it is suggested otherwise, the described method is essentially a spectral decomposition in that each component is separated based on its spectral characteristics, albeit implemented using a correlation based

approach rather than more commonly used matrix inversion based methods.

17. Line 217; why is it necessary to multiply the weighted map by the wavelength compounded image ? It is not clear why the weighted map alone is not displayed.

18. Line 254; it is stated that "the GNR-weighted PA image is poorly projected onto the true absorption spectrum (third row in Fig. 3b)." Again this relates to the questionable use of "projected". How can an image be "projected" on to a spectrum ? Presumably, this statement is intended to mean that the image (third row in Fig. 3b) is a poor representation of the GNR distribution but how can this be ascertained by inspecting this image alone ?

19. Line 261; the disagreement between the measured PA spectra and ground truth optical absorption spectra for the GNR and ink is highlighted. This is due to the wavelength dependent fluence issue referred to earlier. The text then goes on to state "Unfortunately, this serious problem is usually omitted in the literature or not discussed in detail.". This statement is incorrect. The corrupting effect of the wavelength dependence fluence on the measured PA spectrum and its negative impact on identifying and quantifying specific absorbers is very well known by knowledgeable practitioners in the field and has been for many years. Indeed, many of the papers cited in the current work describe approaches to mitigate this problem – eg see 34,35,36,37,38,39,40,41.

20. Line 291; It is stated that "spectroscopic PA molecular imaging" in small animals has not "translated well into clinical tools" with the implication that this is due to some sort of technical inadequacy. This is yet another overly simplistic statement that needs elaboration. Just because a method has not been translated to the clinic that does not mean it can't be. Perhaps the authors of the cited prior studies were less interested in clinical applications than developing tools for imaging mouse models, a worthy objective in its own right. In general, if a method of recovering chromophore concentrations using PA spectroscopic techniques works in mice, there is no reason why it should not also work in humans (with the usual caveat that the instrumentation geometry is suitable of course).

21. Line 348; it is stated that light scattering is much lower in small animals than humans (also stated in Line 388). When applied to small mammals such as mice or rats this seems questionable. Optical scattering is a consequence of refractive index changes at the cellular/intracellular level and these are similar for most human and rodent tissues so optical scattering is likely to be similar as well. If the authors disagree, please provide supporting references.

It is then suggested that light "easily penetrates" mouse/rat tissue. Light penetration depends on absorption as well as scattering. Since the main absorbers in mouse tissue (blood, water, lipids, melanin etc) are essentially the same as those in human tissue and scattering is comparable, the optical penetration depth should be similar. Thicker skin in humans might reduce optical penetration a little, but not excessively so.

22. Line 401; this sentence could be clearer. A bulky laser system is clearly not conducive to clinical translation but large size itself does not preclude the implementation of motion and fluence correction.

23. Line 411 – again, it is not clear why fluence compensation is not required when imaging animal models, unless they are non-mammalian relatively translucent animals, such as zebrafish.

24. Line 439; motion is motion and it is not clear why it can be neglected in live animals but not humans.

25. Line 456; states that "projections may not yield absolute constituent concentrations". Please replace "may" with "can not" or explain how absolute concentrations could be obtained with the

current method alone.

26. Lines 480-487 describes the potential application to the assessment of reconstructive surgery in that the PA spectroscopy could be used to measure blood oxygenation. Can the authors comment on the suitability of their system for this purpose, bearing in mind that the in vivo images in the current work do not appear to reveal any blood vessels, only extremely strong absorbers such as stainless steel needles or GNR distributions.

Reviewer #3:

Remarks to the Author:

In this manuscript, the authors have developed a novel system and corresponding algorithms to address one of the fundamental problems in optoacoustic imaging: the inability to perform spectral analysis in deep tissue due to fluence coloring and motion. The authors performed a series of well-planned measurements (phantom, ex vivo, in vivo) to demonstrate the utility of their approach. In my opinion, this is an outstanding work, which fits Nature Communications in both its high academic quality and interest to the optoacoustic community. Two features distinguish this work from previous studies. First, the authors offer a holistic solution to the challenge of spectral analysis, taking into account real-world behavior. While most papers are focused only on the issue of fluence coloring, the authors also address the issue of motion artifacts – a real issue in clinical systems that is often ignored. Second, the system developed by the authors is not limited to proof-of-concept studies (e.g. physical scanning, long acquisition time, difficult geometries), but is based on a design that could translated into the clinic.

While I strongly support the publication of this manuscript, I do have comments that I believe need to be addressed:

1) It is unclear to me whether the specific reconstruction algorithm for the fluence (or rather average μ_s and μ_a) is completely novel or relies on previous ideas. The authors do cite a large number of papers in the Introduction in which the multiple illumination configurations have been used. The authors correctly state that none of these works reached the level required for clinical testing. Nonetheless, if the theoretical foundations for the current work are directly based on previous studies, this should be clearly mentioned. If the current algorithm has some essential feature that did not appear in previous works, this should also be mentioned.

2) The key feature of the proposed system is clinical compatibility. Clearly, imaging is performed at a high rate thanks to highly advanced hardware. However, it is not clear to me how things currently are in terms of processing time. The system takes a set of optoacoustic and ultrasound images and turns them into a single, quantitative spectral optoacoustic image. This includes optoacoustic inversion, fluence estimate, and motion estimation and correction. It would be useful for the readers if the authors analyze the run time of all these algorithms in the Supplementary materials. Assuming these algorithms introduce a lag in the image formation, future steps for real-time analysis should be discussed.

3) Combined ultrasound-optoacoustic systems have been commercially available for many years now and surely many other researchers have used the feature of motion estimation from US to improve the optoacoustic images. One example is a work from the group of Martin Franz (Phys. Med. Biol. 56, 2011, 5889–5901). I believe it is important to mention this type of works even if they did not focus on optoacoustic spectral unmixing.

4) While the problem of fluence coloring has remained largely unsolved when it comes to sufficiently deep tissue, some important theoretical advances have been made by Stratis Tzoumas from the Ntziachristos group that alleviated the problem (Medical Imaging, IEEE Transactions on 33 (1), 48 – 60 and Nature Communications volume 7, Article number: 12121 ,2016). I believe these papers should be cited.

5) The use of lasers with higher repetition rates, and the selective illumination at each point (rather than uniform illumination) reduces the SNR and thus penetration depth. It seems like that the system could still image relatively deep despite this limitation. Could the authors assess the

price in SNR compared to a system in which all fibers illuminate at once at the MPE level and with lower (e.g. 50 Hz) repetition rate?

6) Regarding Comment 5, I believe the authors could improve their SNR in the future by using multiplexed illumination. Instead of firing from each fiber at a time, one could use a set of illumination patterns that span a basis. Then, one could algorithmically deduce the response that would have been obtained if the fibers were used sequentially. In such an approach, 50% of the fibers could be used with each pulse to deliver more light to the tissue.

Minor comments:

Line 94: The MPE is wavelength dependent. For 700 nm, the limit is 200 mW/cm² and it drops as the wavelength is increased. This means that the optimal SNR is obtained for 10 Hz.

Line 211: The authors write "If fluence is ignored, accurate molecular imaging based on spectral decomposition is nearly impossible." This statement is a bit strong and inconsistent with some of the results presented in the paper and previous works (see Comment 4). In some cases, one may ignore the fluence (superficial imaging, tissue that is not well perfused with blood, or above 950 nm, where the absorption spectra is mostly due to water). I would recommend making this statement more specific, and focus it on deep-tissue imaging.

Response to Reviewers' Comments

“Real-time spectroscopic photoacoustic/ultrasound (PAUS) scanning with simultaneous fluence compensation and motion correction for quantitative molecular imaging”

May 14, 2020

This letter addresses all comments made by the reviewers for the manuscript entitled **“Real-time spectroscopic photoacoustic/ultrasound (PAUS) scanning with simultaneous fluence compensation and motion correction for quantitative molecular imaging”**. We have revised the manuscript to address the reviewers' comments and would like to thank them for carefully reading the paper and providing very useful comments.

In this letter, we respond to **each comment** made by the reviewers.

Reviewer #1:

1. The paper describes the development of a new system for simultaneous photoacoustic and ultrasound imaging in real-time. The aim is to facilitate the clinical translation of the photoacoustic technology by capitalizing on the advantages of the suggested approach. The implementation of the suggested embodiment is indeed very challenging and includes new technological developments such as a fast tunable laser at 1kHz rate.

We thank the reviewer for this evaluation. We spent more than 8 years to build this system, starting from the idea of high-speed spectroscopic imaging and ending with a laboratory prototype. It did require development of unique technologies, such as a new, compact, kHz rate fast-tunable laser. When we first proposed the fast-sweep concept several years ago (J. Xia, C-W Wei, I.M. Pelivanov, and M. O'Donnell, “Photoacoustic Imaging using Narrow Beam Scanning,” Proceedings of the 2011 IEEE Ultrasonics Symposium IUS2011-000618, pp. 2380-2382 (2011)), most of our peers did not believe that the technology could be developed for real-time implementation, particularly the high rep-rate laser combined with a robust, fast, sequential fiber coupling system. We have also done all of the engineering work required to integrate the laser, fiber coupler and an US scanner into a single device. In addition, we adopted and corrected laser fluence compensation methods for in situ real-time implementation without using external equipment, such as an integrating sphere, or assuming known tissue optical properties. A very recent algorithm developed in our group for US speckle tracking was also adopted for real-time motion correction of spectroscopic PA frames. To our knowledge, this is the first time that all of these features have been demonstrated in a functioning, real-time PAUS system.

2. However, the claims of the paper were not properly demonstrated, particularly regarding its potential clinical use. More specific comments are provided below.

We strongly disagree with Reviewer #1 on clinical translation of our approach. Although our system is not optimal for clinical translation compared to other PA approaches for some applications, such as breast imaging, it is appropriate for a range of applications where tight integration between PA and US modalities is required. In particular, we believe that it is very well suited to guiding interventional procedures and drug delivery, areas of rapidly emerging clinical and commercial interest in molecular imaging. Indeed, our pilot studies were partially supported by GE Healthcare (see acknowledgements) and we continue to partner with GE on developing a clinical system using the fast-sweep approach reported here.

We support our statement with multiple arguments below and, thus, will try to convince Reviewer #1 to change his/her opinion.

- (i) As it is evident from this Reviewer's specific comments below, his/her main criticism is that the limited view and limited bandwidth of the US transducer array does not allow quantitative analysis. We absolutely agree with the Reviewer, in part, that full-view and full-bandwidth PA detection is superior to limited view/bandwidth because PA reconstruction becomes mathematically accurate and there are no questions related to quantitation. But, unfortunately, technical and cost limitations make it very difficult to produce a real-time integrated PAUS system that maintains truly full view/bandwidth detection for clinical applications.

As we mention in the Introduction and emphasize here one more time, PA imaging for mice and humans is very different. A mouse can be easily surrounded by an US transducer array and, thus, full view detection can be achieved. Ultra-broadband PVDF, or relatively broadband piezo-composites, provide bandwidths approximating full bandwidth detection. However, mouse body size is about a light penetration depth, which means that high SNR PA signals can be recorded throughout the volume with an US transducer array completely surrounding the mouse. In contrast, the human body is more than 40 light penetration depths and cannot be surrounded by an US transducer array for efficient full-view reconstruction except in limited applications (e.g., breast imaging).

There are many papers on functional PA imaging of the whole mouse, including brain, liver, heart and other organs, but PA images of the whole human body have not been demonstrated. One of the few applications where full view/bandwidth can be performed in humans is breast imaging. For most other applications, we have to deal with limited view detection.

Note that for limited view PA imaging, reconstruction of structural information is impossible without artifacts even using ultra-broadband detectors. Thus, to explore PA imaging for applications other than breast imaging, limited view transducers must be used. **Thus, most PA studies focused on clinical applications, not only ours (except human breast imaging), must deal with the limited view problem. In our work, this is why we focus on limited view transducers, because only they can translate PA imaging to clinical use for a large number of applications. Thus, we strongly disagree with the reviewer on the future potential of our technology.**

(ii) I spent many years developing ultra-broadband PVDF detectors for PA imaging (see [R1-5] below). The transducers I designed and built have been used by many groups for a broad range of imaging applications. I also spent many years developing signal processing and reconstruction algorithms for PA imaging using ultra-broadband PVDF detectors (see some references below). Again, broad bandwidth/view detectors are superior to conventional US transducers in situations where they can be used. However, in most clinical settings they cannot be utilized, primarily because the same transducers must serve as transmitters for pulse-echo US imaging. PVDF is a very inefficient US transmitter and is not used at all in commercial US imaging systems. This is why piezoceramic and single crystal piezoelectric transducers are used in the clinic, with broad enough bandwidths for US imaging but markedly limited bandwidth for PA imaging. The choice of these devices is a classic engineering tradeoff for simultaneous PAUS imaging.

[R1] T.D. Khokhlova, I.M. Pelivanov, A.A. Karabutov, "Optoacoustic tomography utilizing focused transducers: resolution study." *Appl. Phys. Lett.*, **92**, p. 024105-1-3 (2008).

[R2] T.D. Khokhlova, V.V. Kozhushko, I.M. Pelivanov, A.N. Zharinov, V.S. Solomatin, A.A. Karabutov, "Optoacoustic imaging of absorbing objects in a turbid medium: ultimate sensitivity and application to breast cancer diagnostics." *Appl. Opt.* **46**, p.262-272 (2007).

[R3] T.D. Khokhlova, V.V. Kozhushko, I.M. Pelivanov, A.N. Zharinov, A.A. Karabutov, "Spatial resolution of a focused array transducer for laser optoacoustic tomography." *Laser Physics*, **14**(1), p.106-113 (2004).

[R4] T.D. Khokhlova, V.V. Kozhushko, I.M. Pelivanov, A.N. Zharinov, V.S. Solomatin, A.A. Karabutov, "Focused array transducer for 2D optoacoustic tomography." *J. Acoust. Soc. Am.*, **116**(3), p.1498-1506 (2004).

[R5] T.D. Khokhlova, V.V. Kozhushko, I.M. Pelivanov, V.S. Solomatin, A.A. Karabutov, "Wideband focused film transducer for optoacoustic tomography." *Acoustical Physics*, **49**(6), p.682-688, (2003).

(iii) From a commercial perspective, the most successful companies selling PA systems for clinical applications, VisualSonics and iTerra Medical (as well as Photosound), use limited view/bandwidth probes for human imaging. Furthermore, iTerra Medical and Photosound sell both small animal and human imaging systems; and they use full view/bandwidth for small animal models and limited view/bandwidth for human. Why? The reason is the same as we discuss above. Full view/bandwidth detection is possible only for a very limited number of human applications. Indeed, limited view/bandwidth PA detection is used now for the vast majority of human trials. Below we list several publications using limited view/bandwidth probes to image human subjects or aim at that goal. Our technology is more advanced over the conventional one, because we compensate for the wavelength dependence of laser fluence and perform motion correction. In addition, the developed laser is much cheaper than that conventionally used in PA imaging and has a much more compact footprint, both very important issues for commercialization. **Thus, we strongly disagree with the reviewer with his statement on the future clinical potential of our technology.**

3. The main advantage claimed by the authors is that the new system provides quantitative molecular imaging. Achieving quantitative results represents a long standing goal in photoacoustics and so far no sufficiently robust approach has been suggested.

Molecular imaging as defined by the Reviewer focuses on quantitation of molecular sources. We use the term molecular imaging in a broader sense to mean specific image contrast based on the molecular signatures of sources. We do not claim to measure **absolute** concentrations of molecular sources. Molecular targeting and the development of new contrast agents are definitely outside the scope of this paper. In contrast, our work focuses on alternative methods of PA imaging using new lasers and scanning approaches.

We do note that new contrast agents for PA molecular imaging is a very hot topic. Indeed, many major advances in PA contrast agents were published in Nature Journals [R6-11].

[R6] R.E. Borg, J. Rochford, “Molecular photoacoustic contrast agents: design principles and applications”. *Photochem. Photobiol.* **94**(6), p.1175-1209 (2018)

[R7] J. Weber, P. Beard, S. Bohndiek, “Contrast agents for molecular PA imaging”. *Nature Methods.* **13**, p. 639-650 (2016).

[R8] K. Wilson, K. Homan, S. Emelianov, Biomedical photoacoustics beyond thermal expansion using triggered nanodroplet vaporization for contrast-enhanced imaging”. *Nature Communications.* **3**, 618 (2012).

[R9] Y.-S. Chen, S.J. Yoon, W. Frey, M. Dockery, S. Emelianov, “Dynamic contrast-enhanced photoacoustic imaging using photothermal stimuli-responsive composite nanomodulators”. *Nature Communications.* **8**, 15782 (2017).

[R10] D. Razansky, A. Buehler, V. Ntziachristos, “Volumetric real-time multispectral optoacoustic tomography of biomarkers”. *Nature Protocols* **6**(8), p.1121-1129 (2011).

[R11] Y.-S. Chen, Y. Zhao, S.J. Yoon, S.S. Gambhir, S. Emelianov. “Miniature gold nanorods for photoacoustic molecular imaging in the second near-infrared window”. *Nature Nanotechnology.* **14**(5), p. 465-472 (2019).

Again, our definition of molecular imaging differs from the Reviewer’s, where we view it as spatially resolved identification of a substance in the human body. The fast-sweep PA scanner presented here makes this identification from the measured spectrum. Gold nanorods and ink have very different spectra and, thus, we demonstrate that the compensation for the wavelength dependence of laser fluence works independent of the absorber’s spectrum. If these spectra can be identified in vivo, then the spectra of other agents, such as FDA-approved ICG for example, can also be identified.

Although we differ with the Reviewer’s definition of molecular imaging, we consider his/her concern as very serious because other readers may interpret the phrase molecular imaging in the same way. To resolve the issue, we propose a compromise that, hopefully, will introduce less misunderstanding.

Guidance of interventional procedures is an important example of molecular imaging. In contrast to the broader term molecular imaging, the phrase “guidance of interventional procedures” is much more specific, and we believe that our real-time PAUS scanner is ideally suitable for it.

Indeed, this paper presents proof-of-concept measurements for this application. Thus, we narrowed the scope of our work and have changed the manuscript's title to "**Real-time spectroscopic photoacoustic/ultrasound (PAUS) scanning with simultaneous fluence compensation and motion correction for interventional procedure guidance**".

Even if the scope is narrowed, a real-time instrument to molecularly guide interventional procedures is a significant scientific advance with great potential for clinical applications.

...The authors focused on quantifying the bio-distribution (concentration) of extrinsically-administered contrast agents, particularly by correcting for the effects of wavelength-dependent light fluence. Light attenuation is indeed an important factor to take into account but there are other important factors also affecting quantification. For example, linear arrays are known to be strongly affected by so-called limited-view artefacts (as acknowledged by the authors) and have been shown to provide very sub-optimal photoacoustic images. The detection bandwidth of the array and the frequency-dependent directivity of the elements are also important factors that have not been taken into account. Also, the delay-and-sum reconstruction method used by the authors is known not to be quantitative.

We agree with the Reviewer that linear arrays' limited view/bandwidth restricts the accuracy of PA images. However, as discussed above, only limited view and US compatible arrays have a chance to work for a range of applications where it is nearly impossible to surround a human body with US detectors and light penetration is greatly reduced. We also agree that delay-and-sum reconstruction of data from a linear array reconstruction produces significant artifacts for large-scale objects. However, these methods are appropriate to image small-scale objects (such as capillary networks or non-uniform distribution of contrast agent) with less structural artifacts (please see our recent paper submitted to IEEE TMI [R12] and published in arXiv at the link below). We also address this issue in answering Reviewer's comment #7 below.

[R12] M. Kim, G.-S. Jeng, I. Pelivanov, M. O'Donnell. "Deep-learning image reconstruction for real-time photoacoustic system". IEEE TMI, 2020 (accepted for publication, available as "early access" <https://ieeexplore.ieee.org/document/9091172>).

Even though the structural image is distorted for this system, structural artifacts are similar at different wavelengths. The brightness of a pixel will change proportionally to its local absorption. Thus, it is possible to use spectroscopic information to identify whether a given pixel's spectrum corresponds to blood, ICG, or another extrinsic contrast agent; i.e. we can identify the 'molecular' constituent of this pixel. We clearly show that the 'projection' of the measured PA spectrum (after fluence compensation and motion correction) to that of the molecular agent works quite well. In phantom, ex vivo and in vivo experiments, we clearly identify ink, GNR and needle.

Again, we agree that that structural PA image may not be fully accurate, but the dependence of pixel brightness on wavelength should be proportional to the dependence of light absorption with wavelength for the same pixel. We also reference here multiple studies by S. Emelianov [R8-11, R14-15], M. Kolios [R13] and other authors, as well as numerous papers on PA microscopy where a limited view/bandwidth single element probe is usually scanned (note that we only

reference a very small fraction of the enormous literature on PA microscopy). All these studies showed that the pixel brightness depends on wavelength in proportion to the absorption dependence on wavelength. We do not think that there is a reason to ignore all these previous studies.

On the other hand, we agree with the Reviewer that determining the absolute absorption, e.g. in cm^{-1} , or the absorbed laser energy in J/cm^3 , and, therefore, the absolute concentration of an agent in every image pixel is nearly impossible using US linear arrays (limited view & bandwidth), at least with a simple DAS algorithm. However, identifying substances should be quite accurate if both fluence and motion corrections are applied to remove potential artifacts in PA spectra measured at a pixel. We added some material to the Discussion addressing this point. Please also see the answer to this Reviewer's concern #7 on quantitative concentration measurement.

[R13] E. Hysi et al. "Photoacoustic signal characterization of cancer treatment response: correlation with changes in tumor oxygenation". *Photoacoustics*. **5**, p. 25-35 (2017).

[R14] K.P. Kubelick, S.Y. Emelianov. "Prussian blue nanocubes as a multimodal contrast agent for image-guided stem cell therapy of the spinal cord". *Photoacoustics*. **18**, 100166 (2020).

[R15] I.C. Sun, C.H. Ahn, K. Kim, S. Emelianov. Photoacoustic imaging of cancer cells with glycol-chitosan-coated gold nanoparticles as contrast agents". *J. Biomed. Opt.* **24**(12), 121903 (2019).

[R16] Lavaud, J. et al. Noninvasive monitoring of liver metastasis development via combined multispectral photoacoustic imaging and fluorescence diffuse optical tomography. *Int. J. Biol. Sci.* **16**, p. 1616–1628 (2020)

4. The methodology to correct for light fluence attenuation has already been suggested by the authors and by other groups. I understand that the main novelty is the fact that this is achieved in a very short time (real-time).

We disagree with the reviewer's view on novelty of the method. The fast-sweep approach, i.e. using multiple low-energy illuminations instead of a single large energy illumination, was indeed demonstrated before in [R17].

[R17] C-W Wei, T-M Nguyen, J. Xia, B. Arnal, E. Y. Wong, I. Pelivanov, M. O'Donnell, "Real-time Integrated Ultrasound and Photoacoustic Imaging System," *IEEE Transactions on Ultrasonics, Ferroelectrics and Frequency Control* 62, no. 2, pp. 319-328 (2015).

The current realization, however, greatly differs from this initial description of the approach.

In our 2015 publication, we used scanning mirrors for beam delivery, which was impractical, and fast coupling into fibers was not discussed in any way.

Second, here we demonstrate the spectroscopic approach, which is much more complicated, especially in wavelength switching. In our concept, switching must be performed between firings. We did not discuss this possibility at all originally.

Third, we never published our approach to laser fluence compensation. We certainly understood the advantages of fast-sweep illumination for fluence correction, and developed the complete

system described here to implement real-time fluence compensation, but we intentionally did not publish anything on our specific method until we developed the complete system to demonstrate real-time implementation. The Reviewer is correct that other groups published phantom experiments on fluence correction using point source illumination, but even the idea of integrating this into an imaging probe was not proposed. Other investigators proposed to switch between narrow-beam and broad-beam illumination, i.e. measure laser fluence before measurements, point-by-point with single fiber illumination. It's not at all clear how this will work for clinical applications where tissue optical properties can change with time due to blood oxygenation change, for example. This is especially true for guiding interventional procedures, where the optical properties of the medium can change rapidly with time as a result of the procedure itself.

Fourth, we also show here that an arbitrary object in the PA image with amplitude above the noise level can be used for fluence compensation. We did it for two different absorbers with very different optical absorption spectra. In general, more imaging points contributing to the fluence estimate produce more accurate estimates.

Fifth, eq. S14 (see Supplementary Note 6) was previously derived for the 2-D case of sources located in the imaging plane and did not account for fiber inclination. We adopted this expression for the 3D case and account for the inclination of all fibers relative to the imaging plane. In addition, we derived a more general equation (eq. S7 - see Supplementary Note 6) producing simultaneous estimates of the reduced scattering coefficient, which is necessary for fluence correction when scattering is not very large. All these important features have not been presented in the previous literature on fluence compensation.

Clearly, we have not invented photoacoustic imaging. However, all papers on photoacoustic imaging published over the last 25 years, many in Nature journals, can be considered derivative work from A. Bell's original 1881 publication on the photoacoustic effect. We believe that the scientific approach and technological developments presented here are highly novel and represent a significant step toward bringing PA spectroscopic imaging into the clinic for the important application of molecular guidance of interventional procedures.

5. Real-time imaging (as claimed by the authors) implies image visualization during the acquisition procedure, which appears to still be very challenging. Acquisition of 128 photoacoustic signal is performed at 1kHz rate, and ultrasound pulse-echo is further interleaved. It is not clear how all these data are processed. It would be good that the authors provide a clear description on what is the amount of data acquired per second and how these data are transferred and processed. Also, the authors explicitly acknowledge that the computations times for light fluence compensation are still relatively large.

We indeed omitted many details on real-time implementation, as pointed out by the Reviewer, because the paper was already very long with 9 Supplementary Notes. However, we agree that these details are needed to better understand the fast-sweep PAUS system and they further demonstrate its performance. We updated Supplementary Notes 6 and 9 with as many details on real-time implementation as possible. Thus, we thank the Reviewer for pointing this out.

Most importantly, in the revised paper, **we demonstrate real-time implementation of the entire cycle of data processing, including data transfer, interleaved PAUS beamforming, laser fluence compensation and motion correction.**

We have created a Supplementary Data and Software Library section where we partially share raw data and all developed software for real-time data acquisition and processing:

- Real-time acquisition and beamforming;
- Real-time motion correction;
- Real-time laser fluence compensation.

Data acquisition rate:

PAUS data acquisition and transfer to the PC were done in real-time using Verasonics' PCIe Gen2x16 lanes (i.e., the data transfer rate is at least 5.6 GBytes/s). As indicated in Fig. 2(b), the total data acquired for one PAUS frame that needs to be transferred is:

PA: 20 (number of fibers) x 128 (Rx channel) x 2048 (number of axial points) x 2 (bytes per sample)

B-mode: 64 (firings, dual Tx beam) x 128 (Rx channel) x 2048 (axial points) x 2 (bytes per sample)

Therefore, the data transfer rate is $(PA+B)/20ms \sim 2.2$ GB/s, which easily fits within the underlying PCIe bandwidth.

Note that the PC for our Verasonics scanner is not the fastest in the world and is certainly not unique. In partnership with GE, we plan to implement this fast imaging sequence with simultaneous real-time processing on a clinical-grade GE Vivid E95 machine. We expect much better data transfer and signal processing rates compared to our current lab-grade Verasonics Vantage.

Also, we note that US investigators have already demonstrated [R18] five years ago that much larger data flows can be processed in real-time, as is done for plane-wave imaging.

[R18] M. Tanter and M. Fink. "Ultrafast imaging in biomedical ultrasound," *IEEE Tran. UFFC*, **61**(1), p.102-119 (2014).

Beamforming:

We exploited phase rotation beamforming which is a commonly-used receive beamforming for ultrasound imaging [R19]. Due to advances in computing power, a CPU/GPU processor can fully support software-based beamforming. We implanted a pixel-based algorithm to reduce the computational burden without loss of image quality [R20]. To compute one pixel, it needs a mixer, low-pass filter, interpolation filter and phase rotator for every data channel. We generated a sparse matrix, **A**, which performs all these functions, so that the matrix operation produces an (IQ) image **x** from raw data **y** (See Fig. R1 (a)). The size of raw data is 2048x128 and the size of

an image is 512×128 . We made the matrix once and stored it in memory. The beamforming computation, $\mathbf{x} = \mathbf{A}\mathbf{y}$, was done on a GPU processor (NVIDIA GeForce RTX 2080 Ti). The computation time is dependent on the size of the image (the total number of pixels) and the number of non-zero entries per row in the sparse matrix. The number of non-zero entries is determined by the f-number and filter size. We selected a filter size (9) and image size (512×128) which produces clinical quality US images and applied a similar approach to PA reconstruction. Fig. R1 (b) shows the mean computation time versus f-number. **In every case, the computing time is less than the data acquisition time (1ms).**

[R19] Kim, Y., Lee, W., Lee, Y. and Yoo, Y., 2012, February. New direct pixel beamforming based on phase rotation. In *Medical Imaging 2012: Ultrasonic Imaging, Tomography, and Therapy* (Vol. 8320, p. 83201D). International Society for Optics and Photonics.

[R20] Daigle, R.E., Verasonics Inc, 2012. *Ultrasound imaging system with pixel oriented processing*. U.S. Patent 8,287,456.

Figure R1. (a) Beamforming operation using a sparse matrix. (b) Beamforming computation time using a GPU Processor.

Real-time implementation of inter-frame (inter-wavelength) motion estimation and correction:

Estimation:

Recently, we introduced an algorithm based on a randomized search called PatchMatch to speed up processing for near real-time and truly real-time implementation (see [R21] and Supplementary Note 9). The computational complexity and run time associated with motion compensation using PatchMatch have been analyzed by our group (See Table 3 in [R21]). This

algorithm has total arithmetic operations (or equivalently, floating point operations per second (FLOPS)) that can be expressed as

$$(1 + N_i(1 + N_s))(9r_k K_l K_a K_e + 3)$$

N_i is the number of iterations in PatchMatch (=4 in this case); N_s is the number of random searches indicated in Suppl. Fig. 9 (=1 in this case); $K_l K_a K_e$ are the lateral, axial, and elevational kernel size used for tracking (=7,17,1 in this case); $r_k = 1$. For a 2-D image with a size of 128 x 512 pixels doing speckle tracking every 20 ms, 32 GLOPS are required. Compared to a current off-the-shelf Intel Core i9-9900KF CPU processor which has 588 GLOPS, our motion correction is capable of easily running within 20 ms.

[R21] Jeng, G.S., Zontak, M., Parajuli, N., Lu, A., Ta, K., Sinusas, A. J., Duncan, J. S., O'Donnell, M. Efficient two-pass 3-D speckle tracking for ultrasound imaging. *IEEE Access* **6**, 17415-17428 (2018).

Implementation:

In-vivo mouse nanoparticle injection data were used for this demonstration. The Matlab code (PM_realtime.m, which we share in Supplementary Software) with the core function performing PatchMatch (called PM2DabsRT_mex.mexw64) accelerated using Matlab MEX can produce 9 different wavelength, motion-corrected PAUS images. Using the same searching parameters mentioned earlier and **without using GPU or parallel implementation**, we run the program on the predefined ROI (a size of 7.6 mm by 2.2 mm) on Matlab R2019b with an Intel Core i9-9900KF at 3.6GHz. The measured run times of motion estimation for 9 individual wavelengths are as follows, which are well below 20 ms.

- Elapsed time is 17.019 ms.
- Elapsed time is 15.991 ms.
- Elapsed time is 10.953 ms.
- Elapsed time is 15.169 ms.
- Elapsed time is 10.937 ms.
- Elapsed time is 11.088 ms.
- Elapsed time is 15.087 ms.
- Elapsed time is 15.028 ms.
- Elapsed time is 15.038 ms.

The results for 2nd frame (715nm), 5th frame (775nm), 7th frame (815nm), and 9th frame (855nm) are shown below. In each panel, the original PA and corrected PA image are compared. The estimated accumulated lateral and axial displacement results are also plotted together.

Figure R2. Implementation of motion correction in the fast-swept PAUS approach.

Real-time compensation for laser fluence wavelength dependence

Estimation and compensation for wavelength dependence of laser fluence uses data from 200 images (20 fiber sources and 10 wavelengths). The largest burden for this computation is estimating optical parameters of the scattering medium (See Eq. S7 in Supplementary Note 6). We simply use a brute force method which calculates all possible laser fluences and choose best parameters minimizing an error function. A table below describes the values for computing fluences. The computation time using the GPU processor is **23 ms** for estimating optical parameters (9 wavelengths) at a pixel position. (At the first wavelength, control data were recorded by assigning zero laser power to estimate the noise bias. Thus, the other 9 wavelengths were used for estimation.)

	A set for parameter searching (cm^{-1})	Cardinality of a set
μ_{eff}	$\{0.2, \dots, 4.0\}$	100

μ'_s	{2.0, ..., 15.0}	100
----------	------------------	-----

‘Challenging’ does not mean ‘impossible’, especially when several people have worked on implementing this system over an 8-year period. These clarifications and additions to the manuscripts should remove all concerns about real-time system performance.

6. The experiments performed are not related to the potential clinical applicability of the system. Molecular imaging generally implies the use of a contrast agent that targets molecular pathways involved in a given disease. However, there are very few FDA-approved optical contrast agents and they are not targeted, hence molecular photoacoustic imaging does not appear to be a feasible approach. The authors use gold nanorods in the experiments, which have not been approved for clinical use, let alone ink as used in the phantoms.

Questions related with molecular targeting and the development of new contrast agents are definitely outside the scope of this paper. Our work focuses on alternative methods of PA imaging using new lasers and scanning.

Note, however, that developing new contrast agents for molecular imaging is a very hot topic, including specific agents for PA imaging. Many of these studies were published in Nature Journals, as we referenced in addressing Concern #3 above.

As we said above, we narrowed the scope of our work and changed the title of our manuscript to “**Real-time spectroscopic photoacoustic/ultrasound (PAUS) scanning with simultaneous fluence compensation and motion correction for interventional procedure guidance**”.

Even if the scope is narrowed, a real-time instrument to molecularly guide interventional procedures is a significant scientific advance with great potential for clinical applications.

7. It appears that the only benefit of the multi-wavelength approach in the clinical setting would be oxygen saturation mapping. This has not been mentioned or discussed in the manuscript and probably would not be feasible as the authors acknowledge that the probe’s limited view and finite bandwidth is not suitable for imaging the background.

Oxygen saturation is definitely one of the targets for our future projects. This problem is more complicated because it requires not only identification of the substance (e.g. blood), but also quantitative assessment of the ratio of oxy/de-oxy hemoglobin. We definitely do not have sufficient results here to convince the Reviewer that it is feasible, but our future developments will, hopefully, change the reviewer’s opinion in the future. We mention here our new paper under consideration at IEEE TMI [R12], a copy of which is now available in arXiv at the link below:

[R12] M. Kim, G.-S. Jeng, I. Pelivanov, M. O’Donnell. “Deep-learning image reconstruction for real-time photoacoustic system”. IEEE TMI, 2020 (accepted for publication, available as “early access” <https://ieeexplore.ieee.org/document/9091172>). .

In paper [R12] we consider advanced processing based on deep learning and show great improvements in PA reconstruction of blood vessel network using both synthetic and experimental data for limited view/bandwidth imaging. For all simulations used in training, the details of the fast-sweep system used in the present study, including the measured impulse response of the transducer and realistic system noise levels, were applied. Because both spatial and temporal spectra of small-scale objects are not fully destroyed by a limited view/bandwidth probe, there is still the possibility for advanced reconstruction methods. The only objects unlikely to be reconstructed correctly with limited view/bandwidth probes (and we agree with the Reviewer on this point) are large scale objects.

Thus, we cannot demonstrate oxygen saturation mapping right now, but we firmly believe that we are on the way to quantitative oxygen saturation measurements in vivo.

...the authors acknowledge that the probe's limited view and finite bandwidth is not suitable for imaging the background

We are confused by this comment. Indeed, we have proposed and experimentally demonstrated that our system can compensate in real-time for wavelength-dependent laser fluence variations using signals obtained from the object itself (i.e., from the background). There is no reason to believe that similar measurements should not be valid for other applications given that the principle has been experimentally demonstrated here.

8. The authors compare in Fig. 1 their system with other commercially available systems in terms of imaging rate. This makes the impression that their system is much faster than others. However, this comparison is flawed. For example, frame rates of 100Hz have been achieved with lasers that can be tuned on a per-pulse basis. This is faster than their system, which requires 20 ms to form a single-wavelength image (50Hz). It is then basically not true that no commercial system runs faster than 12Hz for spectroscopy (2Hz rate for 5 wavelengths).

We did our table based on the specifications of MSOT-Acuity system (iThera Medical, <https://www.ithera-medical.com/products/msot-acuity/>). We recently contacted iThera and got an official response related to their best performance of the dual-modality OA-US Acuity system. The maximum repetition rate which can be reached is 25 Hz (not 100 Hz). We can publish iThera's response if required.

Furthermore, the reviewer explains a possible reason in his next comment “**the MPE limits are 20mJ/cm² and 200mW/cm². These values change with the wavelength but in any case the optimal SNR is achieved at 10Hz. Above this pulse repetition frequency (PRF), the energy per pulse must be reduced. Considering that the photoacoustic pressure is proportional to the pulse energy, it is preferable to use relatively low PRFs than to use higher PRFs and average multiple frames.**” This explains why PA systems based on broad-beam illumination operate at 10-25 Hz. Using 50 Hz will reduce SNR by a factor of $\sqrt{5}$ in the broad-beam configuration.

We apologize that we misunderstood the frame rate number in specifications of the Acuity system. Again, the reviewer can find 12 Hz as the frame rate of the Acuity system in on-line available

specifications. We decided to remove the table from Fig.1 to avoid confusion, and we extended the explanation of the narrow-beam approach in the Main section of the revised paper version noting advantages and disadvantages of the proposed fast-swept PAUS imaging.

However, the Reviewer is not completely correct in saying that the PA signal amplitude is proportional to laser energy; it is actually proportional to laser fluence. The Reviewer is absolutely correct saying that 1 kHz **broad-beam illumination** will reduce the SNR by a factor of $\sqrt{100}$, which is definitely not optimal. **In fact, we do not do that!**

Instead of broad-beam illumination, we exploit narrow-beam illumination with fast scanning. At every single beam illumination, we **maximize the laser fluence at the sample surface to 20 mJ/cm²**. Then, using multiple (20) illuminations, we achieve the required illumination area. Thus, we operate at optimal irradiation conditions to maximize the PA signal amplitude representing the sum of partial PA signals obtained from individual firings.

Nevertheless, we noted in the Discussion that our approach does have slightly reduced SNR compared to traditional broad-beam illumination.

“Although fast-sweep spectroscopic PA imaging has significant advantages, it also has limitations. As noted, the probe’s limited view and finite bandwidth produce image artifacts, especially for large objects with uniform absorption [56, 66]. In addition, the small footprint of individual laser firings reduces SNR compared to broad illumination. For the sequence used here, the SNR is reduced approximately by the square root of the number of fibers. That is, the current system has approximately 13 dB lower SNR compared to broad illumination with a 50 Hz high-power laser delivering the same surface fluence. Because of our high frame rates and laser stability, however, SNR can be recovered with averaging. For example, a $\Sigma\lambda$ -PA image can recover nearly 9 dB. With motion compensation, longer averaging periods can also improve SNR.”

In other words, our system has a lower SNR by a factor of \sqrt{N} , where N – number of fibers, **not** number of laser firings. Note, that some optimal weighting of partial image summation as a function of position and fiber index can reduce this difference. In addition, energy fluctuations in the laser pulse can be neglected compared to single pulse illumination because the signal is averaged over 20 fiber illuminations, which is very critical for spectral decomposition. **Of course, we have a lower SNR** (again, as mentioned in the Discussion), **but not dramatically and the fast-sweep approach enables full integration with clinical US, laser fluence compensation, motion correction, a much smaller system footprint, and much cheaper system cost.**

...Imaging rates in the kHz range have also been achieved by several groups in the field, and there is also a commercially available system.

Here **we strongly disagree with the Reviewer**. To our knowledge, the only laser, (not the PA imaging system!) which allows 1 kHz OPO operation is a NT252 (Altos Photonics). However, it cannot be used fast, pulse-to-pulse wavelength switching. The switching time is at least 20 ms, which is a major limitation in integrating PA with US frames at real-time rates. It also cannot be used for laser beam sequential fiber coupling, at least in our design, and it’s doubtful that it will be possible to perform robust and stable sequential coupling without breaks in the laser pulse sequence. Finally, this laser is cumbersome and must operate at a fixed repetition rate, which

makes it very difficult to seamlessly trigger from an US scanner and integrate other US modalities such as Doppler, harmonic imaging and shear wave imaging with a real-time PA imaging sequence. We have had multiple conversations with Ekspla (Altos Photonics in USA) about this laser and we can publish our conversations with them if required.

Because the laser is a critical component of our system, we will be happy if the Reviewer provides us with contacts of a laser vendor(s) capable of manufacturing lasers with similar specs. However, we believe that now there are no alternative lasers that can be integrated into our PAUS system.

9. It is mentioned that the MPE limits are $20\text{mJ}/\text{cm}^2$ and $1\text{W}/\text{cm}^2$, so that the optimal SNR is achieved at 50Hz. These numbers are not correct, the MPE limits are $20\text{mJ}/\text{cm}^2$ and $200\text{mW}/\text{cm}^2$. These values change with the wavelength but in any case the optimal SNR is achieved at 10Hz.

The MPE exposure is regulated by ANSI standards, ANSI Z136, Table 5 [R22]:

[R22] <https://www.lia.org/resources/laser-safety-information/laser-safety-standards/ansi-z136-standards>

Table 5: MPE for Skin Exposure to a Laser Beam (ANSI Z136.1-2007)¹

Wavelength λ (nm)	Exposure Duration t (s)	MPE		Notes
		J/cm	W/cm ²	
UV				
180-302	10 ⁻⁹ to 3 E4	3 E-3		For all wavelengths, MPE is listed value or 0.56t ^{0.25} whichever is lower. 3.5 mm limiting aperture.
303	10 ⁻⁹ to 3 E4	4 E-3		
304	10 ⁻⁹ to 3 E4	6 E-3		
305	10 ⁻⁹ to 3 E4	1.0 E-2		
306	10 ⁻⁹ to 3 E4	1.6 E-2		
307	10 ⁻⁹ to 3 E4	25 E-3		
308	10 ⁻⁹ to 3 E4	40 E-3		
309	10 ⁻⁹ to 3 E4	63 E-3		
310	10 ⁻⁹ to 3 E4	0.1		
311	10 ⁻⁹ to 3 E4	0.16		
312	10 ⁻⁹ to 3 E4	0.25		
313	10 ⁻⁹ to 3 E4	0.40		
314	10 ⁻⁹ to 3 E4	0.63		
315-400	10 ⁻⁹ to 10	0.56t ^{0.25}		
315-400	10 ⁻⁹ to 10 ³	1		
315-400	10 ³ to 3 E4		1 E-3	
Visible and Near IR				
400-1400	10 ⁻⁹ to 10 ⁻⁷	2 C _A E-2		3.5 mm limiting aperture.
400-1400	10 ⁻⁷ to 10	1.1 C _A t ^{0.25}		
400-1400	10 to 3 E4		0.2 C _A	
Far IR				
1400-1500	10 ⁻⁹ to 10 ⁻³	0.1		
1400-1500	10 ⁻³ to 10	0.56t ^{0.25}		
1400-1500	10 ⁻³ to 3 E4		0.1	
1500-1800	10 ⁻⁹ to 10	1.0		
1500-1800	10 to 3 E4		0.1	
1800-2600	10 ⁻⁹ to 10 ⁻³	0.1		
1800-2600	10 ⁻³ to 10	0.56t ^{0.25}		
1800-2600	10 ⁻³ to 3 E4		0.1	
2600-1 E6	10 ⁻⁹ to 10 ⁻⁷	1 E-2		
2600-1 E6	10 ⁻⁷ to 10	0.56t ^{0.25}		
2600-1 E6	10 to 3 E4		0.1	

When the irradiation occurs for more than 10 seconds, the maximum laser fluence can be 20 mJ/cm² at 700 nm wavelength, then growing up to reach 100 mJ/cm² at 1064 nm. At that, the max fluence rate is 200 mW/cm² at 700 nm then growing up to reach 1 W/ cm² at 1064 nm. Thus, to reach both requirements, 10 Hz should be used to keep the maximum possible fluence, i.e. maximize the amplitude of generated PA signals. **These MPE values are cited as a fact in many papers, but most of them forget to mention that irradiation should occur for more than 10 seconds.** Using 10 Hz with 10 wavelengths, however, will induce serious motion artifacts which will decorrelate US frames and make motion correction less accurate.

In fact, if we look at the ANSI standards carefully (see the table above), for exposures less than 10 seconds, the MRE fluence **rate** limits depend on time:

$$MPE = 1.1 C_A * t^{0.25} \text{ W/cm}^2,$$

C_A is a wavelength dependent coefficient, which is equal to 1 at 700 nm wavelength; and the fluence per pulse remains the same.

Thus, if the exposure can be controlled in a fixed position, the allowed exposure limits can be much higher than 200 mW/cm².

A reasonable value for the exposure can be 1 sec for a fixed spot, which will allow 1.1 W/cm^2 , this is what we referenced to. Starting and stopping the exposure can be fully managed in the proposed PAUS concept because, again, the laser is driven (triggered) by the US scanner.

We agree with the Reviewer that the information was not complete and have now added all necessary details.

Above this pulse repetition frequency (PRF), the energy per pulse must be reduced. Considering that the photoacoustic pressure is proportional to the pulse energy, it is preferable to use relatively low PRFs than to use higher PRFs and average multiple frames.

We absolutely agree with the reviewer that using a 1 kHz **broad-beam illumination** will reduce the SNR by a factor of $\sqrt{100}$, which is definitely not optimal. **In fact, we do not do that!**

The authors actually use a PRF of 1kHz and very low energy per pulse (1mJ at the laser output and less at the fiber output). It appears to me that the SNR achieved is clearly sub-optimal.

We **disagree** with the reviewer.

Instead of broad-beam illumination, we use narrow-beam illumination with fast scanning. For every laser pulse, we **maximize the laser fluence at the sample surface to 20 mJ/cm^2** . Then, using multiple (20) illuminations, we achieve the required illumination area. Thus, we operate at optimal irradiation conditions to maximize the PA signal amplitude representing the sum of partial PA signals obtained from individual firings.

Nevertheless, we noted in the Discussion that our approach does have slightly reduced SNR compared to traditional broad-beam illumination.

“Although fast-sweep spectroscopic PA imaging has significant advantages, it also has limitations. As noted, the probe’s limited view and finite bandwidth produce image artifacts, especially for large objects with uniform absorption [56, 66]. In addition, the small footprint of individual laser firings reduces SNR compared to broad illumination. For the sequence used here, the SNR is reduced approximately by the square root of the number of fibers. That is, the current system has approximately 13 dB lower SNR compared to broad illumination with a 50 Hz high-power laser delivering the same surface fluence. Because of our high frame rates and laser stability, however, SNR can be recovered with averaging. For example, a $\Sigma\lambda$ -PA image can recover nearly 9 dB. With motion compensation, longer averaging periods can also improve SNR.”

In other words, our system has a lower SNR by a factor of \sqrt{N} , where N – number of fibers, not number of laser firings. Note, that some optimal weighting of partial image summation as a function of position and fiber index can reduce this difference. In addition, energy fluctuations in the laser pulse can be neglected compared to single pulse illumination because the signal is averaged over 20 fiber illuminations, which is very critical for spectral decomposition. **Of course, we have a lower SNR** (again, as mentioned in the Discussion), **but not dramatically and the**

fast-sweep approach enables full integration with clinical US, laser fluence compensation, motion correction, a much smaller system footprint, and much cheaper system cost.

9. I have concerns related to the fluence compensation method. First, the method was only tested in phantoms with uniform absorption and scattering, which do not represent actual heterogeneous biological tissues.

We demonstrate our fluence compensation in both phantoms and biological tissues such as chicken breast and mouse leg. We believe that these subjects can be considered as ‘macro-homogeneous’. If optical micro-heterogeneities are smaller than a few photon transport mean free paths, the distribution of laser fluence in the background will be smooth and can be characterized with a model of a macro-homogeneous medium. We reference to our previous studies where direct measurements of laser fluence distribution were performed in many ex-vivo tissues, where we show that in fact the laser fluence is a quite smooth exponential function far from the surface [R23].

[R23] I.M. Pelivanov, S.A. Belov, V.S. Solomatin, T.D. Khokhlova, A.A. Karabutov, “Direct measurement of the spatial distribution of laser radiation intensity in biological tissues *in-vitro* by the laser optoacoustic method.” *Quantum Electronics*, **36**(12), p. 1089-1096 (2007).

It can be very different when the PA image frame contains several organs. For this case, we agree with the Reviewer that a more complex laser distribution model must be considered. This more complicated problem is outside the scope of this work. However, because US B-mode images are co-registered with PA images, the interfaces between organs can be taken into account and more complex models can be considered and more advanced algorithms of fluence reconstruction can be applied. In some cases, it should be feasible, in others it may not. There are numerous papers on optical property reconstruction for heterogeneous turbid media, and we definitely cannot solve all of them at once. However, we believe that the solution we propose for macro-homogeneous media is a significant step forward. This is, it should be much better than using a-priori known properties together with a model assuming a homogeneous medium, or completely ignoring the laser fluence wavelength. Our approach does not solve all issues, but it is more accurate than previous approaches for in vivo, real-time imaging.

10. I understand that the light diffusion model employed requires some input of optical properties, which are generally unknown in vivo. As I mentioned above, the computations are also not performed in real-time, hence it is not correct to say that real-time imaging was achieved.

The computations are performed in real-time! Real-time imaging is achieved! Please see our response to #5 above.

We also created Supplementary Data and Supplementary Software sections. Everyone can now run our processing scripts and make sure that they can operate at real-time rates.

11. I am also wondering on the effect of cluttering. For linear arrays with light provided from the sides, it is known that skin signals interfere with signals at depths. Considering that the illumination spot is changing, this may lead to frame-dependent cluttering effects that may affect the light fluence calculations.

This is a very important point! We thank the Reviewer for noting it.

The Reviewer is correct in saying that if light is partially absorbed by the skin, it will create US signals propagating deep into the tissue and interfere with PA signals.

However, we believe that this effect is either negligible or strongly reduced for our imaging geometry.

As seen in Supplementary Fig.2, the fiber illumination point is located 6 mm from the imaging plane. The beam diameter is about 2 mm at the tissue surface.

As is well known from the theory of PA signal generation at a medium surface, the induced signal will propagate perpendicular to the surface **independent** of the angle of the laser beam. The central frequency of our transducer is $f_c=15$ MHz, resulting in a diffraction length (for Gaussian beams) for the generated US signal equal to $l_d = \frac{\pi a_0^2}{c} f_c = 31.4$ mm, where a_0 is the laser beam radius and equivalent to the radius of the skin-generated US beam.

In other words, the US signal generated in skin by a 2 mm diameter source will slowly diverge to 4 mm in diameter at 31.4 mm from the surface. Thus, the major part of the skin-generated US signal will not reach the imaging plane within our imaging window. Remember also that an US signal scattered at a depth will appear at twice that depth in the PA image due to the roundtrip propagation path.

The divergence of a beam can be described by the solution of the parabolic equation of diffraction [R24]

$$A(z, r) = \frac{A(z = 0, r = 0)}{\sqrt{1 + D^2}} \exp\left(-\frac{r^2}{a_0^2(1 + D^2)}\right)$$

[R24] M. B. Vinogradova, O. V. Rudenko and A. P. Sukhorukov, “The Wave Theory”. (Moscow: Nauka, 1990)

In Fig.R3 below, we show diffraction of a Gaussian beam of 2 mm in diameter for 6 MHz and 15 MHz (central frequency of the probe, Supplementary Figure 2) frequencies. At 15 MHz, the US signal amplitude in the transducer imaging plane is less than 10^{-6} of its initial value at the medium (skin) surface. Even at 6 MHz (which is -20 dB relative to 15 MHz for this transducer array), the amplitude is less than 10^{-3} of its initial value at the medium (skin) surface up to a 25 mm depth. Thus, clutter should be markedly reduced for our imaging geometry and, therefore, can be simply ignored.

Again, we thank the reviewer for his comment and have added a paragraph to the Discussion section on clutter effects.

Figure R3. Diffraction of Gaussian beams. Linear (upper row) and logarithmic (mid row) are considered. A dashed blue line corresponds to the transducer imaging plane. Depth dependence of the US signal amplitude $A(z)$ in the imaging plane is shown in the lower panels.

12. The motion correction method, which is also claimed as a novelty, was not properly explained. First, this and other methods have already been previously reported. The novelty would be the real-time performance, but there is no discussion on the computation time required for motion correction. Considering the irregular displacement maps in Fig. 5 (875nm), one may assume that it is actually out-of-plane motion what is detected in this case.

The motion correction method has been detailed in Supplementary Note 9. It leverages a randomized and efficient speckle tracking approach called PatchMatch recently developed by our group. We also show its real-time implementation in this paper (see our response to Comment #5).

I am also concern on the fact that out-of-plane motion may be produced. This is generally the case if the system is used as a hand-held scanner and cannot be corrected.

We agree with the Reviewer's comment regarding the limitation of out-of-plane motion using 1-D transducer arrays and 2-D motion tracking methods. However, out-of-plane motion is not significant when the frame rate is high enough. That is, speckle decorrelation (including out-of-plane motion, depth-dependent focusing, etc.) can be minimized as the frame rate increases, which has been demonstrated in Supplementary Fig. 9.

The elevation beamwidth (thus defining the slice thickness) using our 15 MHz probe with elevation length of 1.5mm and focus at 6 mm (see Vermon transducer specifications below) is:

Beamwidth (the width between two zero crossings) = wavelength/aperture size * range * 2 = 0.1mm/1.5mm*6mm*2 = 0.8 mm.

PERFORMANCE DATA SHEET

Reference 28/1633		
002	NC 52	Serial Number 1633J1001

Acoustic Test AD	Date : 15/01/2014, 15:30	VERMON S.A. FRANCE Tel:02.47.37.42.78 Fax:02.47.38.15.45
	Operator : PM	Data Sheet Number 1000002776 Document Ref 050-96 REV01

Element 64 Time response		
Amplitude	Vpp = 507 mV	
Resolution -6dB	Ax.R = 101 ns	
-20dB	Ax.R = 203 ns	
Distance	Del = 8.1 mm	11.0 μs

Element 64 Frequency response	
	-6 dB
cut off Frequency	11.3 MHz
cut off Frequency	19.3 MHz
center Frequency	15.3 MHz
bandwidth	8.0 MHz
bandwidth/Fc	52 %

Elements sensitivity variation			
	Avg	Min	Max
Reso. -6dB	497 mV	429 mV	563 mV
Reso. -20dB	100 ns	91 ns	113 ns
center -6dB	198.0 ns	171.7 ns	302.3 ns
bandwidth -6dB	55 %	46 %	62 %
bandwidth	15.1 MHz	14.4 MHz	15.5 MHz

Element 64 Time and Frequency response

Elements sensitivity Delta (All Elts) = -2.4 dB Delta (4 to 125) = -2.4 dB

Electrical safety

1700 V AC Hipot : passed failed

250 V AC Leakage current : 2 μA

ent GEN020 OSC020 APA021 PL/18 IC129/K G20, E1, sur cible plane inox.

Table 3. Abdominal motion data. The mean range of motion and the (minimum–maximum) ranges in millimeters for each site and each cohort of subjects. The motion is in the superior–inferior (SI) direction.

Site	Observer	Breathing mode	
		Shallow	Deep
Pancreas	Suramo et al. ⁷⁴	20 (10–30)	43 (20–80)
	Bryan et al. ⁷⁵	20 (0–35)	--
Liver	Weiss et al. ⁸⁹	13 +/- 5	--
	Harauz et al. ⁹⁰	14	--
	Suramo et al. ⁷⁴	25 (10–40)	55 (30–80)
Kidney	Davies et al. ⁶⁸	10 (5–17)	37 (21–57)
	Suramo et al. ⁷⁴	19 (10–40)	40 (20–70)
	Davies et al. ⁶⁸	11 (5–16)	--
Diaphragm	Wade ⁸⁰	17	101
	Korin et al. ⁷⁹	13	39
	Davies et al. ⁶⁸	12 (7–28)	43 (25–57)
	Weiss et al. ⁸⁹	13 +/- 5	--
	Giraud et al. ⁷⁸	--	35 (3–95)
	Ford et al. ⁸⁶	20 (13–31)	--

[R25] “The Management of Respiratory Motion in Radiation Oncology” (American Association of Physicists in Medicine); https://www.aapm.org/pubs/reports/rpt_91.pdf

Considering the worst case where respiratory motion is completely out-of-plane, the averaged motion speed (i.e., 6-9 mm/s) is much less than the maximum allowable speed (i.e., 40mm/s). This implies that out-of-plane motion is still within the image elevation slice and speckle decorrelation is not significant. Note that in practice, respiratory motion is generally parallel to the image plane. Therefore, out-of-plane motion speed should be much less than 6-9 mm/s.

Considering the irregular displacement maps in Fig. 5 (875nm), one may assume that it is actually out-of-plane motion what is detected in this case.

We **do not correct** any out-of-plane motion. Instead, we show that under high frame rate (i.e., 50Hz), motion artifacts mainly caused by respiration can be regarded as in-plane motion. The out-of-plane displacement is much less than the slice thickness so that the speckle similarity is still relatively high enough to make sure in-plane motion estimation is reliable. On the other hand, if the frame rate is very low (we already showed this in Supplementary Figure 9), out-of-plane motion can cause severe speckle decorrelation that worsens "in-plane" motion estimation.

We are confused by the Reviewer’s statement on the "irregular displacement" for 875 nm. If you compare displacement results for 855 nm and for 875 nm (shown below in Fig R4), both displacement estimation results are very similar which means motion is NOT irregular.

Figure R4. Displacement estimation for 855 nm and 875 nm wavelengths.

Reviewer #2

This paper presents a system for spectroscopic photoacoustic (PA) imaging that incorporates a fluence correction scheme and motion compensation. Both features are important for the successful implementation of PA spectroscopic methods. Although much of the underlying methodology is based on previously reported work, constructing the system and making it work in a practical sense for in vivo use represents a significant engineering advance.

We thank the reviewer for this evaluation. We spent more than 8 years to build this system, starting from the idea of high-speed spectroscopic imaging and ending with a laboratory prototype. It did require development of unique technologies, such as a new, compact, kHz rate fast-tunable laser. When we first proposed the fast-sweep concept several years ago (J. Xia, C-W Wei, I.M. Pelivanov, and M. O'Donnell, "Photoacoustic Imaging using Narrow Beam Scanning," Proceedings of the 2011 IEEE Ultrasonics Symposium IUS2011-000618, pp. 2380-2382 (2011)), most of our peers did not believe that the technology could be developed for real-time implementation, particularly the high rep-rate laser combined with a robust, fast, sequential fiber coupling system. We have also done all of the engineering work required to integrate the laser, fiber coupler and an US scanner into a single device. In addition, we adopted and corrected laser fluence compensation methods for in situ real-time implementation without using external equipment, such as an integrating sphere, or assuming known tissue optical properties. A very recent algorithm developed in our group for US speckle tracking was also adopted for real-time motion correction of spectroscopic PA frames. To our knowledge, this is the first time that all of these features have been demonstrated in a functioning, real-time PAUS system.

We disagree with the reviewer's evaluation that '**much of the underlying methodology is based on previously reported work**'. The fast scanning approach, i.e. using multiple low-energy illuminations instead of a single large energy illumination was indeed demonstrated before in [Ref 1]

[Ref 1] C.-W. Wei, T.-M. Nguyen, J. Xia, M B. Arnal, I. Pelivanov, M. O'Donnell, "Real-time integrated photoacoustic and ultrasound (PAUS) imaging system to guide interventional procedures: *ex vivo* study." *IEEE Tran. UFFC*. **62**(2), p. 319-328 (2015).

The current realization, however, greatly differs from this initial description of the approach.

In our 2015 publication, we used scanning mirrors for beam delivery, which was impractical, and fast coupling into fibers was not discussed in any way.

Second, here we demonstrate the spectroscopic approach, which is much more complicated, especially in wavelength switching. In our concept, switching must be performed between firings. We did not discuss this possibility at all originally.

Third, we never published our approach to laser fluence compensation. We certainly understood the advantages of fast-sweep illumination for fluence correction, and developed the complete system described here to implement real-time fluence compensation, but we intentionally did not publish anything on our specific method until we developed the complete system to demonstrate real-time implementation. The Reviewer is correct that other groups published phantom

experiments on fluence correction using point source illumination, but even the idea of integrating this into an imaging probe was not proposed.

As an example of previous work, [Ref 2] below (which we cite in our paper) proposed only a very simplified approach with a source located in the imaging plane, i.e. using a 2D geometry without fiber inclination. Limitations applied to the reduced light scattering coefficient were not discussed and a more general equation (**eq. S7** from Supplementary Note 6) was not considered.

[Ref 2] R. J. Zemp. “Quantitative photoacoustic tomography with multiple optical sources”. *Appl. Opt.* **49**(18), p. 3566–3572 (2010).

In [Ref 3], only a single illumination source was considered and the imaging transducer translated. Measured phantoms represented liquids with absorbing tubes embedded. Tissue illumination was still considered in a 2D geometry. That paper definitely helped us, but we propose a much more developed approach in all theoretical, experimental and engineering aspects. We also acknowledge Professor M. Frenz for in-person discussions and suggestions.

[Ref 3] K. G. Held, M. Jaeger, J. Rička, M. Frenz, H. G. Akarçay. Multiple irradiation sensing of the optical effective attenuation coefficient for spectral correction in handheld OA imaging. *Photoacoustics* 4(2), p. 70-80 (2016).

Fourth, we also show here that an arbitrary object in the PA image with amplitude above the noise level can be used for fluence compensation. We did it for two different absorbers with very different optical absorption spectra. In general, more imaging points contributing to the fluence estimate produce more accurate estimates.

Fifth, eq. S11 (see Supplementary Note 6) was previously derived for the 2-D case of sources located in the imaging plane and did not account for fiber inclination. We adopted this expression for the 3D case and account for the inclination of all fibers relative to the imaging plane. In addition, we derived a more general equation - eq. S7 (see Supplementary Note 6) producing simultaneous estimates of the reduced scattering coefficient, which is necessary for fluence correction when scattering is not very large. All these important features have not been presented in the previous literature on fluence compensation.

Finally, we have shown that all proposed algorithms can be integrated into a **real-time** scanner. In particular, we developed a fast beamformer to reconstruct a PA image in only 0.6 ms (i.e. between partial PA frames); laser fluence compensation for the whole spectroscopic cycle taking only 23 ms (i.e. much faster than the 200 ms required for real-time implementation); and motion correction performed faster than the required 20 ms using an US speckle tracking technique developed by our group.

We modified Supplementary Notes 6 and 9 to emphasize **real time implementation**, added a Supplemental Software Library, and uploaded some raw data (see Supplementary Data and Software Library). Thus, anyone can try to run all processing stages to ensure that they work in real-time, and use this software in their own studies. Real-time implementation is also shortly described below.

Data acquisition rate:

PAUS data acquisition and transfer to the PC were done in real-time using Verasonics' PCIe Gen2x16 lanes (i.e., the data transfer rate is at least 5.6 GBytes/s). As indicated in Fig. 2(b), the total data acquired for one PAUS frame needing to be transferred is:

PA: 20 (number of fibers) x 128 (Rx channel) x 2048 (number of axial points) x 2 (bytes per sample)

B-mode: 64 (firings, dual Tx beam) x 128 (Rx channel) x 2048 (axial points) x 2 (bytes per sample)

Therefore, the data transfer rate is $(PA+B)/20ms \sim 2.2$ GB/s, which easily fits within the underlying PCIe bandwidth.

Note that the PC for our Verasonics scanner is not the fastest in the world and is certainly not unique. In partnership with GE, we plan to implement this fast imaging sequence with simultaneous real-time processing on a clinical-grade GE Vivid E95 machine. We expect much better data transfer and signal processing rates compared to our current lab-grade Verasonics Vantage.

Also, we note that US investigators have already demonstrated [R18] five years ago that much larger data flows can be processed in real-time, as is done for plane-wave imaging.

[Ref 4] M. Tanter and M. Fink. "Ultrafast imaging in biomedical ultrasound," *IEEE Tran. UFFC*, **61**(1), p.102-119 (2014).

Beamforming:

We exploited phase rotation beamforming which is a commonly-used receive beamforming for ultrasound imaging [Ref 5]. Due to advances in computing power, a CPU/GPU processor can fully support software-based beamforming. We implanted a pixel-based algorithm to reduce the computational burden without loss of image quality [Ref 6]. To compute one pixel, it needs a mixer, low-pass filter, interpolation filter and phase rotator for every data channel. We generated a sparse matrix, **A**, which performs all these functions, so that the matrix operation produces an (IQ) image **x** from raw data **y** (See Fig. 1R (a)). The size of raw data is 2048x128 and the size of an image is 512x128. We made the matrix once and stored it in memory. The beamforming computation, $\mathbf{x}=\mathbf{A}\mathbf{y}$, was done on a GPU processor (NVIDIA GeForce RTX 2080 Ti). The computation time is dependent on the size of the image (the total number of pixels) and the number of non-zero entries per row in the sparse matrix. The number of non-zero entries is determined by the f-number and filter size. We selected a filter size (9) and image size (512 x 128) which produces clinical quality US images and applied a similar approach to PA reconstruction. Fig. 1R (b) shows the mean computation time versus f-number. **In every case, the computing time is less than the data acquisition time (1ms).**

[Ref 5] Kim, Y., Lee, W., Lee, Y. and Yoo, Y., 2012, February. New direct pixel beamforming based on phase rotation. In *Medical Imaging 2012: Ultrasonic Imaging, Tomography, and Therapy* (Vol. 8320, p. 83201D). International Society for Optics and Photonics.

[Ref 6] Daigle, R.E., Verasonics Inc, 2012. *Ultrasound imaging system with pixel oriented processing*. U.S. Patent 8,287,456.

Figure 1R. (a) Beamforming operation using a sparse matrix. (b) Beamforming computation time using a GPU Processor.

Real-time implementation of inter-frame (inter-wavelength) motion estimation and correction:

Estimation:

Recently, we introduced an algorithm based on a randomized search called PatchMatch to speed up processing for near real-time and truly real-time implementation (see [Ref 7] and Supplementary Note 9). The computational complexity and run time associated with motion compensation using PatchMatch have been analyzed by our group (See Table 3 in [Ref 7]). This algorithm has total arithmetic operations (or equivalently, floating point operations per second (FLOPS)) that can be expressed as

$$(1 + N_i(1 + N_s))(9r_k K_l K_a K_e + 3)$$

N_i is the number of iterations in PatchMatch (=4 in this case); N_s is the number of random searches indicated in Suppl. Fig. 9 (=1 in this case); $K_l K_a K_e$ are the lateral, axial, and elevational kernel size used for tracking (=7,17,1 in this case); $r_k = 1$. For a 2-D image with a size of 128 x 512 pixels doing speckle tracking every 20 ms, 32 GLOPS are required. Compared to a current off-the-shelf Intel Core i9-9900KF CPU processor which has 588 GLOPS, our motion correction potentially can easily run within 20 ms.

[Ref 7] Jeng, G.S., Zontak, M., Parajuli, N., Lu, A., Ta, K., Sinusas, A. J., Duncan, J. S., O'Donnell, M. Efficient two-pass 3-D speckle tracking for ultrasound imaging. *IEEE Access* **6**, 17415-17428 (2018).

Implementation:

In-vivo mouse nanoparticle injection data were used for this demonstration. The Matlab code (PM_realtime.m, which we share in Supplementary Software) with the core function performing PatchMatch (called PM2DabsRT_mex.mexw64) accelerated using Matlab MEX can produce 9 different wavelength, motion-corrected PAUS images. Using the same searching parameters mentioned earlier and **without using GPU or parallel implementation**, we run the program on the predefined ROI (a size of 7.6 mm by 2.2 mm) on Matlab R2019b with an Intel Core i9-9900KF at 3.6GHz. The measured run times of motion estimation for 9 individual wavelengths are as follows, which are well below 20 ms.

Elapsed time is 17.019 ms.

Elapsed time is 15.991 ms.

Elapsed time is 10.953 ms.

Elapsed time is 15.169 ms.

Elapsed time is 10.937 ms.

Elapsed time is 11.088 ms.

Elapsed time is 15.087 ms.

Elapsed time is 15.028 ms.

Elapsed time is 15.038 ms.

The results for 2nd frame (715nm), 5th frame (775nm), 7th frame (815nm), and 9th frame (855nm) are shown below. In each panel, the original PA and corrected PA image are compared. The estimated accumulated lateral and axial displacement results are also plotted together.

Figure 2R. Implementation of motion correction in the fast-swept PAUS approach.

Real-time compensation for laser fluence wavelength dependence

Fluence estimation and compensation uses data from 200 images (20 fiber sources and 10 wavelengths). The largest burden for this computation is estimating optical parameters of the scattering medium (See Eq. S7 in Supplementary Note 6). We simply use a brute force method which calculates all possible laser fluences and choose best parameters minimizing an error function. A table below describes the values for computing fluences. The computation time using the GPU processor is **23 ms** for estimating optical parameters (9 wavelengths) at a pixel position. (At the first wavelength, control data were recorded by assigning zero laser power to estimate the noise bias. Thus, the other 9 wavelengths were used for estimation.)

	A set for parameter searching (cm^{-1})	Cardinality of a set
μ_{eff}	$\{0.2, \dots, 4.0\}$	100
μ'_s	$\{2.0, \dots, 15.0\}$	100

Of course, our technology is based on previous work, just like all newly developed technologies. However, we believe that the scientific approach and technological developments presented here are highly novel and represent a significant step toward bringing PA molecular imaging into the clinic for the important application of molecular guidance of interventional procedures.

In terms of performance, the system appears to work in that it can visualise specific strong absorbers such as an injected bolus of gold nanorods (GNRs) or a steel needle in tissue in vivo. This suggests it could be useful for tracking needle based procedures although the extent to which it might find broader clinical application is unclear

The reviewer probably means **monitoring interventional procedures** when saying ‘**tracking needle based procedures**’.

First, **monitoring interventional procedures** alone is a very large field. Real-time US is commonly used for procedure guidance, including needle visualization, peripheral intravenous placement and central venous cannulation, arterial access, suprapubic aspiration, abscess localization for incision and drainage, foreign body localization, arthrocentesis, pericardiocentesis, thoracentesis and paracentesis, puncture, drug delivery, and ethanol (EA) and radiofrequency (RFA) ablation. More than **two million patients** are treated annually with US guidance, helping reduce procedural errors and costs [Ref 8]

[Ref 8] R.E. Sharpe Jr, L.N. Nazarian, D.C. Levin, L. Parker, V.M. Rao. The increasing role of nonradiologists in performing ultrasound-guided invasive procedures. *J. Am. Coll. Radiol.* **10**(11), p.859-63 (2013).

Although growing at an annual compounded rate of more than 12% [Ref 8], there are still significant barriers to broader acceptance of effective and cost-saving US guidance protocols. In particular, great skill is needed to align devices with US and injections are hard to visualize and validate because the injected agent/drug is opaque to US. Thus, advancing US with the possibility to visualize and monitor the injected solution is a significant advance that will impact millions of procedures annually. Indeed, our pilot studies were partially supported by GE Healthcare (see acknowledgements) and we continue to partner with GE on developing a clinical system using the fast-sweep approach reported here. Helping millions of people every year is clearly important and is of very high medical significance. Saying that the importance of significantly improving millions of clinical procedures per year is unclear seems off the mark.

We support our statement with multiple additional arguments below and, thus, will try to convince Reviewer #2 to change his/her opinion even if our technology would only be limited to monitoring interventional procedures.

- (i) Most proposed PA imaging solutions to date, even if interleaved with US, are driven by PA and use US as an adjunct. The interleaved US in these systems is far from the quality of high-end clinical US scanners and cannot easily implement other US modalities such as color-flow Doppler, harmonic and shear wave imaging. Physicians must be convinced that PA imaging can be used more effectively than US and can be applied either without US altogether or with limited help from lower-quality US imaging. In addition, radiologists must be trained in a new technique and use lower-quality US with limited options. Based on our extensive experience as a group with clinical ultrasound scanning, we are very skeptical that conventional PAUS systems will be readily adopted and will only be used for a very limited range of applications (e.g., breast imaging).
- (ii) In contrast, US imaging is already a very well developed field, and interventional procedure guidance is its fastest growing application area. We developed our current integrated PAUS system with the philosophy of adding a new feature to a high-end, fully functioning clinical US system, namely spectroscopic PA imaging, to add molecular tools to interventional procedure guidance. In this paper, we demonstrated both a method to integrate PA imaging into an US scanner and a specific implementation for spectroscopic PA imaging using a fast-sweep architecture. Unlike previous approaches, our PAUS system is driven by the US scanner with all additional features available and with the possibility to ultimately use the highest quality clinical US images. If we show that PA

imaging can improve interventional procedure guidance by adding molecular tools, our PAUS approach will be rapidly translated clinically because it will be actively pursued by the US industry. Our work was partially supported by GE, one of the world leaders in US imaging. The next stage of our collaboration with GE will focus on integrating PAUS approach described in this paper into their Vivid E95 high-end clinical scanner.

Overall the paper is well structured, provides sufficient detail and is clearly written. However, there are a number of points that require attention. Too often, over-generalisations and simplifications are made. In some cases these are factually incorrect, in others the situation is nuanced and qualification is required. In addition, the challenges relating to small animal and human PA imaging are often incorrectly stated or inadequately qualified.

We thank the Reviewer for very careful reading our paper and for multiple important points indicated in his/her review. However, we disagree with some of his/her statements below and will try to convince him/her to change opinions. Please see our point-by-point responses below.

1. The title refers to quantitative molecular imaging yet all the images appear to be scaled in arbitrary units. A quantitative molecular image would be one in which the image intensity represents the absolute concentration of the absorber of interest.

We use ‘quantitative’ to mean that the spectrum of every pixel in the PA image can be identified according to the **true absorption spectra** of substances within it, not that the PA signal amplitude can yield the absolute concentration of an absorber of interest.

However, we generally agree with the Reviewer’s criticism. Providing a concentration measurement of absorbers is beyond the scope of our work and requires additional effort to compensate limited view/bandwidth artifacts, i.e. advanced image processing. We are working in this direction, and believe that this can be done in the future. Our very recent work on advanced reconstruction can be found in [Ref 9].

[Ref 9] M. Kim, G.-S. Jeng, I. Pelivanov, M. O’Donnell. “Deep-learning image reconstruction for real-time photoacoustic system”. IEEE TMI, 2020 (accepted for publication, available as “early access” <https://ieeexplore.ieee.org/document/9091172>)

Based on the Reviewer’s comment, we modified the title of the manuscript to “**Real-time spectroscopic photoacoustic/ultrasound (PAUS) scanning with simultaneous fluence compensation and motion correction for interventional procedure guidance**”.

As mentioned above, guiding interventional procedures is an important example of molecular imaging. In contrast to the ‘broad’ understanding of quantitative molecular imaging, the terminology ‘guidance of interventional procedures’ is much more specific, and we believe that our PAUS scanner is perfectly suited to this application. All proof-of-concept measurements have been performed.

Even if the scope is narrowed, a real-time instrument to molecularly guide interventional procedures is a significant scientific advance with great potential for clinical applications.

2. Abstract; it is stated that successful human trials have been minimal. Although relatively few fully validated and verified clinical studies have been completed to date (which is not surprising for a still emerging imaging technology), there have been a significant number of human studies that at least fall into the “technical validation” category. These include studies to assess the utility of PA imaging for assessing breast cancer, skin cancer, lymph node disease, inflammatory arthritis, Crohn’s disease, dermatological conditions and cardiovascular disease to name a few. More than minimal it would seem.

We meant that ‘minimal’ compared to small animal studies, i.e. translations from small animal to human studies are minimal. Please see below our arguments on limitations for translation. We modified our statement by exchanging ‘...minimal’ to ‘...limited’ to avoid confusion.

It is implied that the relatively limited clinical translation of PA imaging to date is largely a consequence of the inadequacy of current PA spectroscopic methods and motion related-errors that the current work addresses. This seems overly simplistic. There are many barriers to clinical translation, some technical (and not necessarily only those described in the current work), others relate to the limited availability of imaging systems, alignment with clinical workflow and regulatory or financial hurdles. That is not to say the motion and fluence correction issues addressed in the current work are not important. They are and their solution will certainly facilitate clinical translation in some areas. However, to imply they represent the primary barrier to clinical translation is a significant over-generalization/simplification.

We have to admit that we agree with the reviewer and we applied necessary corrections to the Main and Discussion sections.

3. Abstract; there is a suggestion that quantitative clinical PA spectroscopy requires that the issue of wavelength dependent fluence changes be addressed. Why clinical PA spectroscopy in particular? The issue is a general one and applies to any application of spectroscopic PA imaging whether the target is a live mouse, ex vivo tissues or some other spectrally heterogeneous medium.

We agree with the reviewer that the problems of motion correction and laser fluence compensation are indeed general problems. We emphasized ‘clinical PA spectroscopy’ because we believe that ex vivo and small animal studies are intermediate steps towards the ultimate goal of clinical translation. A mouse can be fixed in a special holder (see iThera Medical website, for example); transducers can surround the mouse and translate over the whole body; laser illumination can be performed from all directions to homogenize the distribution of light within the target. Most of these conditions are hard to replicate for humans in routine clinical use.

Consider a mouse, for example. A **mouse body cross-section is of the order of a few light penetration depths** and can be easily surrounded by the transducer array. Illumination of a mouse from all directions distributes **light in the background almost uniformly**. The typical size of the human body is more than **40 light penetration depths** and cannot be surrounded by the transducer array for efficient full-view reconstruction. Tissue illumination can be performed

from one side only, thus, laser fluence will decay rapidly with depth. There is a fundamental difference in PA imaging of humans rather than mice, with the one exception being the human breast.

Below is a result of a Monte-Carlo simulation for a 25 mm homogeneous cylinder (mimicking a mouse body) with light irradiation from all sides compared to the distribution of light in a semi-infinite medium (mimicking human body). Both media have the same optical properties typical for biological tissues. The large difference between these cases is very clear in Fig. 3R below. For the ‘mouse’ (simulated as a 25 mm in diameter cylinder), the fluence is almost uniform, but for a ‘human’ fluence decays exponentially (far from the surface). Note that we used a very conservative estimate for mean μ_a and μ'_s for a nude mouse – actual penetration is probably even more uniform.

Figure 3R. Comparison of single side tissue illumination (case of a handle-held probe) and uniform illumination of tissue from all directions (typical for PA imaging of mice).

Again, in small animal experiments, special illumination geometries can be applied to more evenly distribute light throughout the volume. These same geometries cannot be applied to human studies (again, with the possible exception being the human breast).

We modified the Main and Discussion sections to make this point clearer and thank the reviewer for pointing this out.

4. Line 32; It is stated that optical spectroscopy is not used routinely because tissue scattering limits penetration depth to 1mm or less. This statement is not correct. Clinical near infrared (NIR) spectroscopy is widely used to measure hemodynamic changes and blood oxygenation (M.Wolf, et al. J. Biomed. Opt. 12(6), 062104 (2007); in its simplest form, the ubiquitous pulse-oximeter for measuring arterial oxygen saturation found in every hospital relies on optical spectroscopy. Moreover, if NIR light is used, tissue penetration can extend to several cm and is not limited to 1mm.

The reviewer makes an incorrect citation from our manuscript:

Reviewer #2: “optical spectroscopy is not used routinely because tissue scattering limits penetration depth to 1mm or less. This statement is not correct”.

Manuscript: “optical spectroscopy is not used routinely *in vivo* because high tissue scattering typically limits **coherent penetration** to a millimeter or less”.

Our statement was about coherent light, not diffused light. The coherent light penetrates in tissue by about 1 mm depth. This defines the limit of OCT-based methods.

However, we understand the confusion because we meant **spatially-resolved** spectroscopy, which may be not evident from this sentence. We corrected the sentence in Main and thank the Reviewer for noticing the inconsistency.

5. Line 52; It is stated that most previous studies have employed small animal models but the associated methods have not translated well to humans. The meaning of this statement is not clear. If an imaging system can successfully image a mouse, why would it not be possible to image human tissue, assuming of course the detection geometry was suitable (ie a linear or 2D array used in backward mode) ? Again this relates to the point made previously about clinical translation. Just because previous studies have used mouse models rather than human subjects, it does not necessarily follow that the imaging performance is not suitable for human studies. There are many reasons why a system suitable for small animal imaging might not have been applied to human studies, including lack of inclination or opportunity on behalf of the investigators...

We already started this discussion above, but will repeat our arguments in more detail here because the ‘equivalence’ of small animal and ‘human’ PA studies is a consistent point noted by the Reviewer throughout his/her comments.

We **strongly disagree with this Reviewer’s opinion**. Our very detailed response on the comparison between small animal and human studies is below. However, we thank the Reviewer because this misunderstanding is probably the result of inadequate language in our manuscript.

We corrected the text in the Discussion section to clarify the principle differences between small animal and human studies.

- (i) PA tomography uses recorded acoustic signals (i.e., PA signals) to reconstruct heat release in the medium under study, i.e. the product of the light absorption coefficient and optical fluence at every observation point. The inverse problem can be solved with mathematical rigor when the detection surface represents an infinite plane, cylinder or sphere. In addition, the spectrum of the PA signal cannot be distorted by a single channel of the transducer array. That is, because PA signals are ultra-broadband, the detector bandwidth should also be ultra-broadband. **Thus, for ‘full view’ and ‘full bandwidth’ data acquisition, the problem of PA tomography can be solved exactly. To accurately recover the distribution of optical absorption, and the wavelength dependence of the absorption (i.e., spectrum), from even ideal source reconstructions, the optical fluence distribution at each pixel/voxel must also be known at all wavelengths over the band of interest.** Of course, exact reconstruction of the absorption coefficient also requires acoustic and thermal homogeneity, i.e. the Gruniesen parameter and sound speed are constant throughout the medium.

In the Fig. 4Rb below, reconstruction of a uniformly heated sphere is demonstrated under full view/bandwidth conditions.

- (ii) Now consider the case where full view detection is not possible for some reason. For this case (see Fig.4Rc), PA reconstruction will be highly distorted. There are multiple publications investigating the effect of limited view detection. Furthermore, the transducers usually used for PA signal detection have limited bandwidth. The reconstruction is very different compared to ground truth (see Fig.4Rd). Only PVDF and recently developed optical resonators provide sufficient bandwidth to not introduce bandwidth-related distortions. I personally spent multiple years in designing and characterizing PVDF transducers for PA tomography and investigating artifacts induced by limited view/bandwidth.

Figure 4R. PA Reconstruction of uniformly heated sphere.

- (iii) Again, PA tomography reconstructs the distribution of heat release, not light absorption coefficient. To separate these two depth- and wavelength-dependent functions, laser fluence should be known or made as uniform as possible.

(iv) Why are we talking about limited view/bandwidth problems and laser fluence distribution here? Because conditions (i)-(iii) can all be potentially satisfied for small animals. There are a few different approaches to obtain full view reception for mice [Ref 10-16]. A 1-D array can be rotated or a circular array can be moved over the mouse body. Light illumination can be performed from all sides to achieve a quasi-uniform distribution of laser fluence in the mouse body. Please see our MC simulations in Fig.R2 above.

[Ref 10] D. Razansky, A. Buehler, V. Ntziachristos, “Volumetric real-time multispectral optoacoustic tomography of biomarkers”. *Nature Protocols* **6**(8), p.1121-1129 (2011).

[Ref 11] <https://www.tomowave.com/products/lois-3d-pre-clinical-system/>

[Ref 12] <https://www.pst-inc.com/tritom/>

[Ref 13] <https://www.ithera-medical.com/products/msot-invision/>

[Ref 14] S. Gottschalk, O. Degtyaruk, B. Mc Larney, J. Rebling, M.A. Hutter, X.L. Dean-Ben, S. Shoman, D. Razansky. “Rapid volumetric optoacoustic imaging of neural dynamics across the mouse brain”. *Nature Biomedical Engineering* **3**, p.392-401 (2019).

[Ref 15] H.-C.A. Lin, X.L. Dean-Ben, M. Reiss, V. Schöttle, C. A. Wahl-Schott, I. R. Efimov, D. Razansky. “Ultrafast volumetric optoacoustic imaging of whole isolated beating mouse heart”. *Sci. Rep.* **8**, 14132 (2018).

[Ref 16] X. Ai, Z. Wang, H. Cheong, *et al.* “Multispectral optoacoustic imaging of dynamic redox correlation and pathophysiological progression utilizing upconversion nanoprobes”. *Nature Communications* **10**, 1087 (2019).

(v) Can conditions (i)-(iii) be fulfilled for human? NO in most cases! The only case where (i)-(ii) are fulfilled is human breast, but (iii) still is a significant issue. Of course, one can investigate cardiac diseases in mice with PA imaging or measure mouse brain neural activity, but these studies cannot be readily translated to human. **Humans (i) cannot be surrounded by US transducers (and if they are, this will be absolutely useless due to the limited light penetration depth); ultra-broadband transducers (ii) cannot be used for US signal transmission (not detection) and cannot be used to integrate PA imaging with clinical US imaging; light distribution (iii) cannot be considered as uniform as in mice. All these factors make direct translation from mice to humans nearly impossible in most cases. These statements are in our paper and we believe that they have been documented sufficiently.**

(vi) In contrast to the ‘small animal approach’, we assume that for most clinical applications of PA imaging, linear US arrays will be used. This is not because we are unfamiliar with full view/bandwidth approaches, but because only clinical arrays can be used in practice to integrate US and PA modalities. US imaging is already a well-developed, fast-growing field and we propose to add a PA dimension. We have fully integrated the PA modality into the US scanner. Unlike previous approaches, our

PAUS system is driven by the US scanner, ultimately providing simultaneous PA images and the highest quality clinical US images with all US modalities easily integrated. If we show that PA imaging can improve interventional procedure guidance by adding molecular tools, our PAUS approach will be rapidly translated clinically because it will be actively pursued by the US industry. We modified the Main section of our paper to make this goal clearer.

6. Figure 1; Shouldn't the comparison be between the current state-of-the-art irrespective of whether it originates from industry or an academic research lab ? It could be argued that commercially available systems will generally offer lower imaging performance than the latest experimental system developed in a university lab (as is the authors' system) so the comparison is not entirely fair.

Actually, iThera Medical was created by leading scientists in the field of PA imaging. Their image quality represents the state-of-the-art and, in our opinion, no better results have been achieved in research labs under the conditions mimicking clinical ones. But, yes, we cannot guarantee that there are no better systems available in a university lab; we also do not want to create enemies. Reviewer #1 also criticized this table (see above), but noted a different aspect. Thus, we decided to remove the table. We believe that the comparison and conclusions can be done by the readers.

7. Lines 82 - 89; The confounding influence of the wavelength dependent fluence on the quantification of absorbers using spectroscopic methods is discussed. It is good to see this so clearly stated as it is a crucial point. However, it is not a problem limited only to clinical PA imaging but applies equally to imaging mice or rats.

This Reviewer's comment is related with his/her Comment #5. Our detailed response to that comment applies here as well. In brief, the radiation diagram can be adopted for mice and rats to make the light distribution more or less uniform within the target, but it is impossible to realize in humans. In our revised version, we tried to make this clearer.

8. Line 90: it is stated that tissue motion affects clinical spectroscopic PA imaging. Again, why "clinical" specifically? If a live mouse or rat is being imaged, then respiratory and cardiac induced motion will be present so wouldn't this also be similarly problematic? Indeed the authors illustrate this nicely in figure 6 by evaluating their motion tracking method by imaging a mouse in vivo !

We thank the Reviewer for his/her opinion on the importance of the proposed motion correction technique for different areas of PA imaging. We applied some edits in corresponding places to emphasize the importance of this operation. However, as we discussed above, the conditions applied for small animal experiments are not so strict as for human, especially in monitoring dynamically changing procedures. A mouse can be positioned in a special holder to reduce

motion [Ref 10-16, see above], some recorded frames can be rejected [Ref 17], experiments can be repeated if required. The solution for clinical studies should be robust.

[Ref 17] A. Ron, N. Davoudi, X.L. Deán-Ben, D. Razansky. “Self-gated respiratory motion rejection for optoacoustic tomography. *Appl. Sci.* **9**, 2737 (2019).

9. Line 94: the MPE usually depends on wavelength and total exposure time – shouldn’t these be quoted as well? Also, it is stated that the optimal SNR occurs at 50 Hz. Please provide a justification for this claim.

We thank the Reviewer for this comment. The exposure time should indeed be quoted.

The MPE exposure is regulated by ANSI standards [Ref 18].

[Ref 18] <https://www.lia.org/resources/laser-safety-information/laser-safety-standards/ansi-z136-standards>

Table 5: MPE for Skin Exposure to a Laser Beam (ANSI Z136.1-2007)¹

Wavelength λ (nm)	Exposure Duration t (s)	MPE		Notes
		J/cm	W/cm ²	
UV				
180-302	10 ⁻⁹ to 3 E4	3 E-3		For all wavelengths, MPE is listed value or 0.56t ^{0.25} whichever is lower. 3.5 mm limiting aperture.
303	10 ⁻⁹ to 3 E4	4 E-3		
304	10 ⁻⁹ to 3 E4	6 E-3		
305	10 ⁻⁹ to 3 E4	1.0 E-2		
306	10 ⁻⁹ to 3 E4	1.6 E-2		
307	10 ⁻⁹ to 3 E4	25 E-3		
308	10 ⁻⁹ to 3 E4	40 E-3		
309	10 ⁻⁹ to 3 E4	63 E-3		
310	10 ⁻⁹ to 3 E4	0.1		
311	10 ⁻⁹ to 3 E4	0.16		
312	10 ⁻⁹ to 3 E4	0.25		
313	10 ⁻⁹ to 3 E4	0.40		
314	10 ⁻⁹ to 3 E4	0.63		
315-400	10 ⁻⁹ to 10	0.56t ^{0.25}		
315-400	10 ⁻⁹ to 10 ³	1		
315-400	10 ³ to 3 E4		1 E-3	
Visible and Near IR				
400-1400	10 ⁻⁹ to 10 ⁻⁷	2 C _A E-2		3.5 mm limiting aperture.
400-1400	10 ⁻⁷ to 10	1.1 C _A t ^{0.25}		
400-1400	10 to 3 E4		0.2 C _A	
Far IR				
1400-1500	10 ⁻⁹ to 10 ⁻³	0.1		
1400-1500	10 ⁻³ to 10	0.56t ^{0.25}		
1400-1500	10 to 3 E4		0.1	
1500-1800	10 ⁻⁹ to 10	1.0		
1500-1800	10 to 3 E4		0.1	
1800-2600	10 ⁻⁹ to 10 ⁻³	0.1		
1800-2600	10 ⁻³ to 10	0.56t ^{0.25}		
1800-2600	10 to 3 E4		0.1	
2600-1 E6	10 ⁻⁹ to 10 ⁻⁷	1 E-2		
2600-1 E6	10 ⁻⁷ to 10	0.56t ^{0.25}		
2600-1 E6	10 to 3 E4		0.1	

When the irradiation occurs for more than 10 seconds, the maximum laser fluence can be 20 mJ/cm² at 700 nm wavelength, then growing to reach 100 mJ/cm² at 1064 nm. For those conditions, the max fluence rate is 200 mW/cm² at 700 nm then growing to reach 1 W/cm² at 1064 nm. Thus, to reach both requirements, 10 Hz should be used to keep the maximum possible fluence, i.e. maximize the amplitude of generated PA signals. **These MPE values are cited as a fact in many papers, but most of them forget to mention that irradiation should occur for more than 10 seconds.** Using 10 Hz with 10 wavelengths, however, will induce serious motion artifacts which will decorrelate US frames and make motion correction less accurate.

In fact, if we look at the ANSI standards carefully (see the table above), for exposures less than 10 seconds, the MRE fluence **rate** limits depend on time:

$$MPE=1.1C_A*t^{0.25} \text{ W/cm}^2,$$

C_A is a wavelength dependent coefficient, which is equal to 1 at 700 nm wavelength; and the fluence per pulse remains the same.

Thus, if the exposure can be controlled in a fixed position, the allowed exposure limits can be much higher than 200 mW/cm².

A reasonable value for the exposure can be 1 sec for a fixed spot, which will allow 1.1 W/cm²; this is what we referenced to. Starting and stopping the exposure can be fully managed in the proposed PAUS concept because, again, the laser is driven (triggered) by the US scanner.

We agree with the Reviewer that the information was not complete and have now added all necessary details and additional references.

10. Line 95; it is stated that at least 5 wavelengths are required for stable decomposition. This statement needs qualifying as it appears to be based on one particular study using a specific approach in a particular experimental situation. The minimum number of wavelengths depends on a variety of factors including the nature of the inversion scheme, noise levels and the wavelength range in which the wavelengths are chosen. Based on simple notions of uniqueness however, it can be expected to depend on the number of spectrally distinct chromophores in the tissue. So in the current study, if the principal absorbers are oxyhemoglobin, deoxyhemoglobin and the GNRs, then three wavelength should be sufficient, four perhaps if scattering is considered.

We agree with the Reviewer that, mathematically, the notion of uniqueness requires the same number of wavelengths equal to the number of chromophores. However, this does not take into account noise and measurement inaccuracies; thus, the required number of wavelengths may need to increase. The number of wavelengths also depends on the similarity of different chromophore spectra and, again, SNR. In addition, the model of laser fluence distribution may be not exactly accurate because of tissue heterogeneities and, thus, local fluctuations of the PA signal amplitudes are possible.

Our goal is to reconstruct the spectral shape for each absorber to reach high correlation between ground truth and measured chromophore spectra. When the full spectrum is analyzed, error should go down.

Again, we thank the Reviewer for pointing out this inaccuracy in our description. We modified the paper to address his/her concern.

11. Line 96 – why limit the comparison to commercial systems only ? If there is a non-commercial system that provides a higher frame rate, this should be cited as well.

We already addressed this concern above; we fully agree with the Reviewer that the table may induce questions. Thus, we removed it from Figure 1.

12. Line 99 – it is stated that “motion changes the local concentration of absorbers”. It is suggested that this sentence is re-worded (also the caption in Fig1(b)). Clearly motion itself has no effect on concentration but presumably it is meant that it corrupts the measurement of the concentration.

We apologize for the confusion. We consider tissue motion in the coordinate system relative to the transducer. In other words, the concentration changes at a certain point of the PA image because chromophores may move from point to point relative to the fixed position of the transducer.

We understand the issue and have modified this sentence to make it clearer.

13. Line 108 – detailed specifications of the laser are provided in the supplementary information, rightly so as it is a unique and critical component of the system. However, it does not state how the wide tuning range is achieved. It is stated that it is a diode pumped device but what is being pumped, OPO, Ti-Sapphire or some other medium? Please provide a brief explanation.

A **Ti-Sapphire** crystal was used in the OPO laser resonator which was pumped by the second harmonic of the Nd:YLF laser. In other words, the 1053 nm radiation of the diode-pumped Nd:YLF laser was first converted to its second harmonic and then the second harmonic was used to pump the **Ti-Sapphire** crystal positioned in the resonator. Wavelength tuning was performed with an acousto-optic filter. This is why it can be done so fast (less than a ms) and any sequence of wavelengths can be used. Again, this is not just non-linear conversion as used in most OPO systems. This is a **Ti-Sapphire** laser.

Unfortunately, we cannot provide more detailed information on the laser because it is the proprietary information of Laser Export Co Ltd. Please contact the designer of the laser, Roman Biryukov (sales@laser-export.com), for more details.

14. Lines 200-213; The fluence correction approach relies on measuring the PA signal generated in a local absorber (eg a well defined blood vessel or some other discrete absorber) as a function of the light propagation distance. In practice, this measurement is achieved by delivering the excitation light sequentially through a linear array of optical fibres. The success of this method is critically dependent upon a number of assumptions, none of which

appear to be stated explicitly. First, the background optical properties of the target tissue must be homogenous, at least over the entire tissue volume that is irradiated. Second, the local absorber must not itself perturb the fluence distribution. These assumptions need to be stated and justified, especially the assumption of spatial homogeneity. The latter is straightforward to achieve in a phantom but will not always apply in tissue. For example a superficial blood vessel located at one end of the fibre array might only be traversed by the photons emitted by a small proportion of the optical fibres which would introduce errors in to the fluence correction.

We generally agree with the Reviewer's criticism and added necessary assumptions of a macro-homogeneous medium in Methods and corresponding Supplementary Note.

If optical micro-heterogeneities are smaller than a few photon transport mean free paths, the distribution of laser fluence in the background will be smooth and can be characterized with a model of a macro-homogeneous medium. We reference to our previous studies where direct measurements of laser fluence distribution were performed in many ex-vivo tissues, where we show that in fact the laser fluence is a quite smooth exponential function far from the surface [Ref 19]. In addition, one can find thousands of papers where biological tissues are considered as macro-homogeneous media with effective (average) optical properties (Refs. 20-21)

[Ref 19] I.M. Pelivanov, S.A. Belov, V.S. Solomatin, T.D. Khokhlova, A.A. Karabutov, "Direct measurement of the spatial distribution of laser radiation intensity in biological tissues *in-vitro* by the laser optoacoustic method." *Quantum Electronics*, **36**(12), p. 1089-1096 (2007).

[Ref 20] S.L. Jacques. "Optical properties of biological tissues: a review". *Phys. Med. Biol.* **58**(11), p. R37-61 (2013).

[Ref 21] W.-F. Cheong, S.A. Prahl, A.J. Welch. "A review of the optical properties of biological tissues". *IEEE J. Quant. Electronics*. **26**(12), p.2166-2185 (1990).

It can be very different when the PA image frame contains several organs or large blood vessels. For this case, we agree with the Reviewer that the vessel will affect the background fluence distribution (without vessel). This more complicated problem is outside the scope of this work. However, we performed some numerical simulations which show that our estimation is not too bad and quite accurately assess the laser fluence even if a large blood vessel presents in the image area.

Figure 5R. Schematic diagram of laser sources and a tissue medium. An arterial vessel (cylindrical absorber) is aligned parallel to the y -axis.

Consider the fluence $\Phi(x, y, z)$ when a medium has a large, strong absorber such as an arterial vessel (see Fig. 5R above). As shown in Fig. 5R, we assumed that an arterial vessel (cylindrical absorber) is aligned parallel to the transducer elevational direction (y -axis) so that it appears as a circular absorber in the z - x image plane. The z - x location of the absorber is (5 mm, -3.6 mm) and the diameter D is either 0.8, 1.6 or 2.4 mm. The absorption coefficient inside the vessel is 10 cm^{-1} . We also investigated the fluence when the tissue medium has no vessel as a control. The reduced scattering, absorption and effective attenuation coefficients for the background medium is 10 cm^{-1} , 0.027 cm^{-1} and 0.9 cm^{-1} , respectively.

Figure 6R shows the slice (z - x plane) of 3D fluence in the medium irradiated from each fiber position where the medium has (a) no vessel and (a) a vessel ($D=2.4$). We assumed that a point target of interest is located at (8 mm, 6 mm) in the z - x plane.

Figure 6R. Light fluence in a medium irradiated from a fiber. Every z-x plane is $20\text{ mm} \times 40\text{ mm}$.

Figure 7R (a) shows the fluence magnitude at the position as a function of the distance between fiber and target positions for every vessel diameter. The μ_{eff} estimates computed from the signal samples are 1.06 , 1.35 and 1.57 cm^{-1} when the diameter D is 0.8 , 1.6 and 2.4 mm respectively. As seen, the target does influence the attenuation of laser fluence in the background medium. The larger diameter of the vessel, the higher deviation of laser fluence is from the control (without vessel) because of higher shadowing.

Our primary purpose is estimating **true fluence** rather than estimating true **optical parameters of the background medium**. For example, when the diameter is 2.4 mm , our method (solid cyan line) gives the light attenuation in the background medium corresponding to $\mu_{\text{eff}} = 1.57\text{ cm}^{-1}$ (Fig. 7R (b)). However, the actual in-depth distribution of laser fluence through this point taken from MC simulation (red line) is much closer to our estimate than to the laser fluence distribution without the vessel present. Indeed, the best fit to the ground-truth (ranging from 5 to 10 mm) using the analytical fluence model corresponding to the blue dotted line gives $\mu_{\text{eff}} = 1.41\text{ cm}^{-1}$, much closer to our estimate, $\mu_{\text{eff}} = 1.57\text{ cm}^{-1}$ than to the original distribution, $\mu_{\text{eff}} = 0.9\text{ cm}^{-1}$, without vessel.

Thus, our estimation is not perfect, but it accounts for the influence of large absorbers (such as blood vessels) and provides estimates close to the actual fluence distribution in the medium. Furthermore, we use 20 different fiber illuminations which can potentially be used in sorting and grouping in different combinations to get more localized estimates of laser fluence in different parts of the medium. This can be very important at the interface between two organs, for example. The co-registered US structural image can help define these interfaces and guide reconstruction. **However, this work is far beyond the scope of the current paper, and we prefer to stay here within the model of macro-homogeneous media.**

Figure 7R. (a) Signal magnitude at the target position $(x,y,z)=(6 \text{ mm}, 0 \text{ mm}, 8 \text{ mm})$ as a function of the distance between source and target. (b) Fluence along the axial line crossing the target position $(x,y)=(6 \text{ mm}, 0 \text{ mm})$.

In addition, deep-learning algorithms can also be used to minimize distortions induced by limited view/bandwidth. In some cases, it should be feasible, in others it may not. However, we believe that the solution we propose for macro-homogeneous media is a significant step forward. That is, it should be much better than using a-priori known properties together with a model assuming a homogeneous medium, or completely ignoring the wavelength-dependent laser fluence distribution. Our approach does not solve all issues, but it is more accurate than previous approaches for in vivo, real-time imaging.

15. Line 210; “If fluence is ignored, accurate molecular imaging based on spectral decomposition is nearly impossible. We believe that this is especially true for imaging in humans, although we note that for mouse imaging fluence correction can sometimes be ignored.” The first sentence is true and a crucial point. However, why is it “especially true” in humans ? It is equally true when imaging mice or rats since their tissues contain essentially the same spectrally distinct chromophores as humans (oxyhemoglobin, deoxyhemoglobin, water, lipids, melanin etc) that give rise to the wavelength dependent fluence variations that corrupt the measured PA spectra and confound spectral decomposition.

We already discussed in detail our opinion on the difference between small animal and human studies above (please see our answers to comments #3, #5 and #7 above). In brief, the radiation diagram can be optimized for mice and rats to make the light distribution mostly uniform within the target, but it is impossible to realize in humans. In our revised version, we tried to make this clearer.

16. Lines 214 – 221 and supplementary note 4; The terminology relating to the spectral decomposition method is unclear. It appears that the component weighted images are

essentially images that have been spectrally unmixed to reveal the presence of a specific absorber, the GNRs in most cases. However, this is referred to as the “projection” of the measured PA spectrum on to the known absorption spectrum of the absorber. This seems a confusing way of describing it. It might be clearer to state that the multiwavelength image data were spectrally unmixed or decomposed to provide the GNR (or other component) weighted images. Although it is suggested otherwise, the described method is essentially a spectral decomposition in that each component is separated based on its spectral characteristics, albeit implemented using a correlation based approach rather than more commonly used matrix inversion based methods.

The idea of component-weighted imaging using a correlation-based approach is similar to spectral decomposition (or spectral unmixing) but with a slightly different purpose. For example, define the measured absorption coefficient with 10 different wavelengths (denoted as a vector $\vec{\mu}_a$) at a certain pixel as

$$\vec{\mu}_a = \sum_{l=1}^L C_l \vec{\alpha}_l \quad (1)$$

representing the linear combination of L different absorbers, each having a concentration C_l and spectrum $\vec{\alpha}_l$. The purpose of spectral decomposition is to obtain C_l where $l = 1, 2, \dots, L$, which can be obtained by

$$C_l = \frac{\vec{\mu}_a \cdot \vec{\alpha}_l}{\|\vec{\alpha}_l\|_2 \|\vec{\mu}_a\|_2} \quad (2)$$

where $\|\ \ \|_2$ means the L2 norm.

In contrast, component-weighted imaging (CWI) is produced by multiplying the original wavelength-compounded PA image by the correlation coefficient (or projection “angle” or NCC defined in this paper) between the measured and ground-truth spectrum of the absorber under investigation.

$$CWI \sim \|\vec{\mu}'_a\|_1 \frac{\vec{\mu}'_a \cdot \vec{\alpha}'_l}{\|\vec{\mu}'_a\|_2 \|\vec{\alpha}'_l\|_2} \quad (3)$$

where $\vec{\mu}'_a$ and $\vec{\alpha}'_l$ are the mean-subtracted version of $\vec{\mu}_a$ and $\vec{\alpha}_l$, respectively. Note that the original wavelength-compounded imaging is proportional to the L1 norm of $\vec{\mu}_a$. According to this equation, the CWI is **not** equal to the concentration of the absorber under investigation. However, similar to spectral decomposition, the CWI contains the inner product between the measured spectrum and the known spectrum to weight the original PA image by the contribution of the underlying absorber. Therefore, the purpose of CWI is to provide another quantitative representation and spectroscopic visualization of a specific absorber. For a certain pixel, when the NCC is 1 (which means it is 100% the desired absorber), the PA signal is fully preserved. On the other hand, when the NCC is less than 1, the PA signal is suppressed according to the NCC difference relative to unity. Note that the idea of component-weighted imaging is very similar to existing coherence-factor-based ultrasound or PA imaging where image quality is enhanced by weighting the coherence factor calculated on a per-pixel basis.

We agree with the reviewer that ‘projection’ is probably not the best word to describe the above procedure. In the revised version, we removed the word ‘projection’.

17. Line 217; why is it necessary to multiply the weighted map by the wavelength compounded image ? It is not clear why the weighted map alone is not displayed.

As explained earlier, the component-weighted image (CWI) is the product of the NCC (or projection angle) and the original PA image, which conveys similar information (but not exactly the same) regarding absorber concentration. Showing CWI instead of the NCC map has the following advantages: (1) the intensity of the original PA image is preserved as opposed to the NCC where the PA intensity is normalized. (2) The dynamic range of the CWI is much higher than the NCC map.

18. Line 254; it is stated that “the GNR-weighted PA image is poorly projected onto the true absorption spectrum (third row in Fig. 3b).” Again this relates to the questionable use of “projected”. How can an image be “projected” on to a spectrum ? Presumably, this statement is intended to mean that the image (third row in Fig. 3b) is a poor representation of the GNR distribution but how can this be ascertained by inspecting this image alone ?

When we refer to “poorly projected on the true absorption”, it means that the second term (i.e., NCC or projection “angle”) on the right of Eq. (3) defined above is low. The resulting CWI is therefore suppressed, which quantifies the absence of GNRs. This indeed means a poor representation of the GNR distribution without laser fluence compensation. Again, CWI is not exactly the same as the absorber concentration using spectral unmixing (i.e., Eq. (2) defined above) but still provides a quantitative representation of the underlying absorber.

19. Line 261; the disagreement between the measured PA spectra and ground truth optical absorption spectra for the GNR and ink is highlighted. This is due to the wavelength dependent fluence issue referred to earlier. The text then goes on to state “Unfortunately, this serious problem is usually omitted in the literature or not discussed in detail.”. This statement is incorrect. The corrupting effect of the wavelength dependence fluence on the measured PA spectrum and its negative impact on identifying and quantifying specific absorbers is very well known by knowledgeable practitioners in the field and has been for many years. Indeed, many of the papers cited in the current work describe approaches to mitigate this problem – eg see 34,35,36,37,38,39,40,41.

We agree with the Reviewer’s comment. The problem is discussed in a number of papers, but the correction was not accurately applied for ex vivo and in vivo studies. We modified the indicated phrase to resolve this confusion.

20. Line 291; It is stated that “spectroscopic PA molecular imaging” in small animals has not “translated well into clinical tools” with the implication that this is due to some sort of technical inadequacy. This is yet another overly simplistic statement that needs elaboration. Just because a method has not been translated to the clinic that does not mean it can’t be. Perhaps the authors of the cited prior studies were less interested in clinical applications

than developing tools for imaging mouse models, a worthy objective in its own right. In general, if a method of recovering chromophore concentrations using PA spectroscopic techniques works in mice, there is no reason why it should not also work in humans (with the usual caveat that the instrumentation geometry is suitable of course).

This comment is related to #3, #5, #7 and #15 above. Our answers to those previous comments apply here as well.

In brief, the problem is that the instrumentation geometry can be nearly optimal for small animals but cannot be realized for human.

... **“with the usual caveat that the instrumentation geometry is suitable of course”**. This caveat is in fact the principle issue. The instrument geometry for mouse imaging is **not** suitable for the vast majority of potential clinical applications of PA.

Again, we applied corrections in the text of the manuscript to make our point clearer.

21. Line 348; it is stated that light scattering is much lower in small animals than humans (also stated in Line 388). When applied to small mammals such as mice or rats this seems questionable. Optical scattering is a consequence of refractive index changes at the cellular/intracellular level and these are similar for most human and rodent tissues so optical scattering is likely to be similar as well. If the authors disagree, please provide supporting references.

It is then suggested that light “easily penetrates” mouse/rat tissue. Light penetration depends on absorption as well as scattering. Since the main absorbers in mouse tissue (blood, water, lipids, melanin etc) are essentially the same as those in human tissue and scattering is comparable, the optical penetration depth should be similar. Thicker skin in humans might reduce optical penetration a little, but not excessively so.

We agree with the Reviewer’s comment and apologize for the confusion. It is fixed in our revised version.

I personally spent years studying the optical properties of biological tissues and have multiple publications in that area (see below).

If you look at measured summary tables of optical properties [Ref 20-21], they vary considerably. Indeed, they can vary by an order of magnitude for the same tissue. We can cite some literature data supporting our statement, but then a different set of papers could be cited to counter the argument. We would like to avoid this situation.

[Ref 20] S.L. Jacques. “Optical properties of biological tissues: a review”. *Phys. Med. Biol.* **58**(11), p. R37-61 (2013).

[Ref 21] W.-F. Cheong, S.A. Prahl, A.J. Welch. “A review of the optical properties of biological tissues”. *IEEE J. Quant. Electronics.* **26**(12), p.2166-2185 (1990).

The statement in our paper is based on our own observations, which are not valid as a proper scientific response. I could say 'believe me', but that has no value! Thus, although we are sure that light scattering in nude mice skin, subcutaneous and muscle tissues is less than that in human, we prefer to remove this phrase from the paper. We will make a comprehensive study of this using the PA method of direct measurement of laser fluence in turbid media and publish comparison results in future work.

The phrase "**light "easily penetrates" mouse/rat tissue"** still applies because the mouse/rat cross-section is nearly a few light penetration depths; and tissue illumination can be performed from all sides to achieve a quasi-uniform distribution of laser fluence inside.

22. Line 401; this sentence could be clearer. A bulky laser system is clearly not conducive to clinical translation but large size itself does not preclude the implementation of motion and fluence correction.

We corrected the phrase – thanks for the suggestion.

23. Line 411 – again, it is not clear why fluence compensation is not required when imaging animal models, unless they are non-mammalian relatively translucent animals, such as zebrafish.

Please see our answers to #3, #5, #7, #15 and #21.

We applied necessary corrections in the rest of the paper to make this statement clearer.

24. Line 439; motion is motion and it is not clear why it can be neglected in live animals but not humans.

We thank the Reviewer for his opinion on the importance of the proposed motion correction technique for different areas of PA imaging. We applied some edits in corresponding places to emphasize the importance of this operation. However, as we discussed above, the conditions for small animal experiments are not as strict as for humans, especially in monitoring dynamically changing procedures. A mouse can be positioned in a special holder to reduce motion, some recorded frames can be rejected [Ref 10-16, see above], and experiments can be repeated if required. The solution for clinical studies should be robust.

25. Line 456; states that "projections may not yield absolute constituent concentrations". Please replace "may" with "can not" or explain how absolute concentrations could be obtained with the current method alone.

We agree with the reviewer and have corrected this phrase.

26. Lines 480-487 describes the potential application to the assessment of reconstructive surgery in that the PA spectroscopy could be used to measure blood oxygenation. Can the

authors comment on the suitability of their system for this purpose, bearing in mind that the in vivo images in the current work do not appear to reveal any blood vessels, only extremely strong absorbers such as stainless steel needles or GNR distributions.

Oxygen saturation is definitely one of the targets for our future work. This problem is more complicated because it requires not only identification of the substance (e.g. blood), but also quantitative assessment of the ratio of oxy/de-oxy hemoglobin. We definitely do not have sufficient results here to convince the Reviewer that it is feasible, but our future developments will, hopefully, solve this problem.

We note here our new work now under consideration at IEEE TMI [Ref 9, see above], where we consider advanced processing based on deep learning and show great improvements in PA reconstruction of blood vessel networks using both synthetic and experimental data for limited view/bandwidth imaging. For all simulations used in training, the details of the fast-sweep system used in the present study, including the measured impulse response of the transducer and realistic system noise levels, were applied. Because both spatial and temporal spectra of small-scale objects are not fully destroyed by a limited view/bandwidth probe, there is still the possibility for advanced reconstruction methods. The only objects unlikely to be reconstruct correctly with limited view/bandwidth probes are large scale objects.

Thus, we are on the way to quantitative oxygen saturation measurements, but not right now. And we think that it is correct to discuss “quantitative oxygen saturation measurements” as a potential application.

Reviewer #3

In this manuscript, the authors have developed a novel system and corresponding algorithms to address one of the fundamental problems in optoacoustic imaging: the inability to perform spectral analysis in deep tissue due to fluence coloring and motion. The authors performed a series of well-planned measurements (phantom, ex vivo, in vivo) to demonstrate the utility of their approach. In my opinion, this is an outstanding work, which fits Nature Communications in both its high academic quality and interest to the optoacoustic community. Two features distinguish this work from previous studies. First, the authors offer a holistic solution to the challenge of spectral analysis, taking into account real-world behavior. While most papers are focused only on the issue of fluence coloring, the authors also address the issue of motion artifacts – a real issue in clinical systems that is often ignored. Second, the system developed by the authors is not limited to proof-of-concept studies (e.g. physical scanning, long acquisition time, difficult geometries), but is based on a design that could translated into the clinic.

We thank the Reviewer for these comments.

While I strongly support the publication of this manuscript, I do have comments that I believe need to be addressed:

1) It is unclear to me whether the specific reconstruction algorithm for the fluence (or rather average μ_s and μ_a) is completely novel or relies on previous ideas. The authors do cite a large number of papers in the Introduction in which the multiple illumination configurations have been used. The authors correctly state that none of these works reached the level required for clinical testing. Nonetheless, if the theoretical foundations for the current work are directly based on previous studies, this should be clearly mentioned. If the current algorithm has some essential feature that did not appear in previous works, this should also be mentioned.

The fast-sweep approach, i.e. using multiple low-energy illuminations instead of a single large energy illumination, was indeed demonstrated before in [1R].

[1R] C-W Wei, T-M Nguyen, J. Xia, B. Arnal, E. Y. Wong, I. Pelivanov, M. O'Donnell, "Real-time Integrated Ultrasound and Photoacoustic Imaging System," IEEE Transactions on Ultrasonics, Ferroelectrics and Frequency Control 62, no. 2, pp. 319-328 (2015).

The current realization, however, greatly differs from this initial description of the approach.

In our 2015 publication, we used scanning mirrors for beam delivery, which was impractical, and fast coupling into fibers was not discussed in any way.

Second, here we demonstrate the spectroscopic approach, which is much more complicated, especially in wavelength switching. In our concept, switching must be performed between firings. We did not discuss this possibility at all originally.

Third, we never published our approach to laser fluence compensation. We certainly understood the advantages of fast-sweep illumination for fluence correction, and developed the complete system described here to implement real-time fluence compensation, but we intentionally did not

publish anything on our specific method until we developed the complete system to demonstrate real-time implementation. The Reviewer is correct that other groups published phantom experiments on fluence correction using point source illumination, but even the idea of integrating this into an imaging probe was not proposed.

As an example of previous work, [2R] below (which we cite in our paper) proposed only a very simplified approach with a source located in the imaging plane, i.e. using a 2D geometry without fiber inclination. Limitations applied to the reduced light scattering coefficient were not discussed and a more general equation (eq. S7) from Supplementary Note 6 was not considered.

[2R] R. J. Zemp. “Quantitative photoacoustic tomography with multiple optical sources”. *Appl. Opt.* **49**(18), p. 3566–3572 (2010). However, that was very far from the current realization.

In [3R], only a single illumination source was considered and the imaging transducer translated. Measured phantoms represented liquids with absorbing tubes embedded. Tissue illumination was still considered in a 2D geometry. That paper definitely helped us, but we propose a much more developed approach in all theoretical, experimental and engineering aspects. We also acknowledge Professor M. Frenz for in-person discussions and suggestions.

[3R] K. G. Held, M. Jaeger, J. Rička, M. Frenz, H. G. Akarçay. Multiple irradiation sensing of the optical effective attenuation coefficient for spectral correction in handheld OA imaging. *Photoacoustics* 4(2), p. 70-80 (2016).

Fourth, we also show here that an arbitrary object in the PA image with amplitude above the noise level can be used for fluence compensation. We did it for two different absorbers with very different optical absorption spectra. In general, more imaging points contributing to the fluence estimate produce more accurate estimates.

Fifth, eq. S11 (see Supplementary Note 6) was previously derived for the 2-D case of sources located in the imaging plane and did not account for fiber inclination. We adopted this expression for the 3D case and account for the inclination of all fibers relative to the imaging plane. In addition, we derived a more general equation (eq. S7 - see Supplementary Note 6) producing simultaneous estimates of the reduced scattering coefficient, which is necessary for fluence correction when scattering is not very large. All these important features have not been presented in the previous literature on fluence compensation.

Finally, we have shown that all the proposed algorithms can be integrated into a real-time scanner. In particular, we developed a fast beamformer to reconstruct a PA image in only 0.6 ms (i.e. between partial PA frames); laser fluence compensation for the whole spectroscopic cycle taking only 23 ms (i.e. much faster than the 200 ms required for real-time implementation); and motion correction performed faster than the required 20 ms using an US speckle tracking technique developed by our group.

We modified Supplementary Notes 6 and 9 to emphasize real time implementation, added a Supplemental Software Library, and uploaded some raw data (see Supplementary Data and Software Library). Thus, anyone can try to run all processing stages to ensure that they work in real-time, and use this software in their own studies.

Of course, our technology is based on previous work, just like all newly developed technologies. However, we believe that the scientific approach and technological developments presented here are highly novel and represent a significant step toward bringing PA molecular imaging into the clinic for the important application of molecular guidance of interventional procedures.

2) The key feature of the proposed system is clinical compatibility. Clearly, imaging is performed at a high rate thanks to highly advanced hardware. However, it is not clear to me how things currently are in terms of processing time. The system takes a set of optoacoustic and ultrasound images and turns them into a single, quantitative spectral optoacoustic image. This includes optoacoustic inversion, fluence estimate, and motion estimation and correction. It would be useful for the readers if the authors analyze the run time of all these algorithms in the Supplementary materials. Assuming these algorithms introduce a lag in the image formation, future steps for real-time analysis should be discussed.

We indeed omitted many details on real-time implementation, as pointed out by the Reviewer, because the paper was already very long with 9 Supplementary Notes. However, we agree that these details are necessary for better understanding the fast-sweep PAUS system and they further demonstrate its performance. We updated Supplementary Notes 6 and 9 with as many details on real-time implementation as possible. Thus, we thank the Reviewer for pointing this out.

Most importantly, in the revised paper version, **we demonstrate real-time implementation of the entire cycle of data processing, including data transfer, interleaved PAUS beamforming, laser fluence compensation and motion correction.**

We have created a Supplementary Data and Software Library section where we partially share raw data and all developed software for real-time data acquisition and processing:

- Real-time acquisition and beamforming;
- Real-time motion correction;
- Real-time laser fluence compensation.

The real-time implementation is also shortly described below.

Data acquisition rate:

PAUS data acquisition and transfer to the PC were done in real-time using Verasonics' PCIe Gen2x16 lanes (i.e., the data transfer rate is at least 5.6 GBytes/s). As indicated in Fig. 2(b), the total data acquired for one PAUS frame that needs to be transferred is:

PA: 20 (number of fibers) x 128 (Rx channel) x 2048 (number of axial points) x 2 (bytes per sample)

B-mode: 64 (firings, dual Tx beam) x 128 (Rx channel) x 2048 (axial points) x 2 (bytes per sample)

Therefore, the data transfer rate is $(PA+B)/20ms \sim 2.2$ GB/s, which is well below the underlying PCIe bandwidth.

Note that the PC for our Verasonics scanner is not the fastest in the world and is certainly not unique. In partnership with GE, we plan to implement this fast imaging sequence with simultaneous real-time processing on a clinical-grade GE Vivid E95 machine. We expect much better data transfer and signal processing rates compared to our current lab-grade Verasonics Vantage.

Also, we note that in the US field investigators have already demonstrated [4R] five years ago that much larger data flows can be processed in real-time, as is done for plane-wave imaging.

[4R] M. Tanter and M. Fink. "Ultrafast imaging in biomedical ultrasound," *IEEE Tran. UFFC*, **61**(1), p.102-119 (2014).

Beamforming:

We exploited phase rotation beamforming which is a commonly-used receive beamforming for ultrasound imaging [5R]. Due to an advance of computing power, a CPU/GPU processor can fully support software-based beamforming. We implanted a pixel-based algorithm to reduce the computational burden without loss of image quality degradation [6R]. To compute one pixel, it needs mixer, low-pass filter, interpolation filter and phase rotator for every data channel. We generated a sparse matrix, \mathbf{A} , which operates all the functions, so that the matrix operation produces an (IQ) image \mathbf{x} from raw data \mathbf{y} (See Fig. 1R (a)). The size of raw data is 2048x128 and the size of an image is 512x128. We made the matrix once and stored it in memory. The beamforming computation, $\mathbf{x}=\mathbf{A}\mathbf{y}$, was conducted by a GPU processor (NVIDIA GeForce RTX 2080 Ti). The computation time is dependent on the size of image (the number total pixels) and the number of non-zeros per row in the sparse matrix. The number of non-zeros is determined by f-number and filter size. We decided the filter size (9) and image size (512X128) which do not incur image degradation, compared with bigger sizes. Fig. 1R (b) shows the mean computation time over f-number. **Every case, the computing time is less than the data acquisition time (1ms).**

[5R] Kim, Y., Lee, W., Lee, Y. and Yoo, Y., 2012, February. New direct pixel beamforming based on phase rotation. In *Medical Imaging 2012: Ultrasonic Imaging, Tomography, and Therapy* (Vol. 8320, p. 83201D). International Society for Optics and Photonics.

[6R] Daigle, R.E., Verasonics Inc, 2012. *Ultrasound imaging system with pixel oriented processing*. U.S. Patent 8,287,456.

Figure 1R. (a) Beamforming operation using a sparse matrix. (b) Beamforming computation time using a GPU Processor.

Real-time implementation of inter-frame (inter-wavelength) motion estimation and correction:

Estimation:

Recently, we introduced an algorithm based on a randomized search called PatchMatch to speed up processing for near real-time and truly real-time implementation (see [7R] and Supplementary Note 9). The computation complexity and run time associated with motion compensation using PatchMatch have been analyzed by our group (See Table 3 in [7R]). The PatchMatch has the total arithmetic operations (or equivalently, floating point operations per second (FLOPS)) that can be expressed as

$$(1 + N_i(1 + N_s))(9r_k K_l K_a K_e + 3)$$

N_i is the number of iterations in PatchMatch (=4 in this case); N_s is the number of random searches indicated in Suppl. Fig. 9 (=1 in this case); $K_l K_a K_e$ are the lateral, axial, and elevational kernel size used for tracking (=7,17,1 in this case); $r_k = 1$. For a 2-D image with a size of 128 x 512 pixels doing speckle tracking every 20 ms, the FLOPS of PatchMatch are 32 GLOPS. Compared to the current off-the-shelf Intel Core i9-9900KF CPU processor which has 588 GLOPS, our motion correction is potentially capable of running within 20 ms.

[7R] Jeng, G.S., Zontak, M., Parajuli, N., Lu, A., Ta, K., Sinusas, A. J., Duncan, J. S., O'Donnell, M. Efficient two-pass 3-D speckle tracking for ultrasound imaging. *IEEE Access* **6**, 17415-17428 (2018).

Implementation:

An in-vivo mouse nanoparticle injection data was used for this demonstration. The Matlab code (PM_realtime.m, which we share in Supplementary Software) with the core function realizing PatchMatch (called PM2DabsRT_mex.mexw64) that is accelerated by using Matlab MEX can produce 9 different wavelengths, motion corrected PAUS images. Using the same searching parameters mentioned earlier and **without using GPU or parallel implementation**, we run the program on the predefined ROI (a size of 7.6 mm by 2.2 mm) on Matlab R2019b with an Intel Core i9-9900KF at 3.6GHz. The measured run times of motion estimates for individual 9 wavelengths are as follows, which are well below 20 ms.

Elapsed time is 17.019 ms.

Elapsed time is 15.991 ms.

Elapsed time is 10.953 ms.

Elapsed time is 15.169 ms.

Elapsed time is 10.937 ms.

Elapsed time is 11.088 ms.

Elapsed time is 15.087 ms.

Elapsed time is 15.028 ms.

Elapsed time is 15.038 ms.

The results for 2nd frame (715nm), 5th frame (775nm), 7th frame (815nm), and 9th frame (855nm) are shown below. In each panel, the original PA and corrected PA are compared. The estimated accumulated lateral and axial displacement results are also plotted together.

Figure 2R. To implementation of motion correction in the fast-swept PAUS approach.

Real-time compensation for laser fluence wavelength dependence

Processing following the computation of 200 images (20 sources and 10 wavelengths) is fluence estimation and compensation. The most burden part is estimating optical parameters of a scattering medium (See Eq. S7 in Supplementary Note 6). We simply use a brute force method which calculate all possible laser fluences and choose best parameters minimizing an error. A table below describes the values for computing fluences. The computation time using the GPU processor (NVIDIA GeForce RTX 2080 Ti) is **23 ms** for estimating optical parameters (9 wavelengths) of a pixel position. (At the first wavelength, control data were recorded by assigning zero laser power to estimate the noise bias. Thus, the rest 9 wavelengths were involved in the estimation.)

	A set for parameter searching (cm^{-1})	Cardinality of a set
μ_{eff}	{0.2, ..., 4.0}	100
μ'_s	{2.0, ..., 15.0}	100

Of course, our technology is based on previous work, just like all newly developed technologies. However, we believe that the scientific approach and technological developments presented here are highly novel and represent a significant step toward bringing PA molecular imaging into the clinic for the important application of molecular guidance of interventional procedures.

3) Combined ultrasound-optoacoustic systems have been commercially available for many years now and surely many other researchers have used the feature of motion estimation from US to improve the optoacoustic images. One example is a work from the group of Martin Franz (Phys. Med. Biol. 56, 2011, 5889–5901). I believe it

is important to mention this type of works even if they did not focus on optoacoustic spectral unmixing.

We agree with the Reviewer's criticism and added references to previous approaches in motion correction.

4) While the problem of fluence coloring has remained largely unsolved when it comes to sufficiently deep tissue, some important theoretical advances have been made by Stratis Tzoumas from the Ntziachristos group that alleviated the problem (Medical Imaging, IEEE Transactions on 33 (1), 48 – 60 and Nature Communications volume 7, Article number: 12121 ,2016). I believe these papers should be cited.

We agree with the Reviewer that these papers must have been cited. We apologize for not citing them; we have applied necessary corrections.

In addition, we fully agree that work [NC 12121] is outstanding. The approach of eigenspectra PA tomography can assess one of the most important tissue characteristics – its blood oxygenation. But the approach described in that paper solves a different problem; it is designed to evaluate the background. In our case, we also evaluate the background, but only for fluence compensation. It may be a good idea to explore the correlation between laser fluence determined in our method and the background oxygenation level.

We are not sure that the approach applied in [NC 12121] will work in the presence of additional absorbers with absorption much stronger than the background. Each additional absorber will introduce an additional eigen spectrum. Thus, reconstructing strong absorbers with eigenspectra PA tomography should be tested. We are also not sure how limited view and bandwidth problems will affect the reconstruction routine proposed in [NC 12121]. It is definitely worth exploring in the future, as well as the combination of our method and [NC 12121]. Indeed, one is designed to assess strong absorbers and the other to assess the background.

5) The use of lasers with higher repetition rates, and the selective illumination at each point (rather than uniform illumination) reduces the SNR and thus penetration depth. It seems like that the system could still image relatively deep despite this limitation. Could the authors assess the price in SNR compared to a system in which all fibers illuminate at once at the MPE level and with lower (e.g. 50 Hz) repetition rate?

In fact, we did this assessment in the Discussion section:

“Although fast-sweep spectroscopic PA imaging has significant advantages, it also has limitations. As noted, the probe's limited view and finite bandwidth produce image artifacts, especially for large objects with uniform absorption [56, 66]. In addition, the small footprint of individual laser firings reduces SNR compared to broad illumination. For the sequence used here, the SNR is reduced approximately by the square root of the number of fibers. That is, the current system has approximately 13 dB lower SNR compared to broad illumination with a 50 Hz high-power laser delivering the same surface fluence. Because of our high frame rates and laser stability, however,

SNR can be recovered with averaging. For example, a $\Sigma\lambda$ -PA image can recover nearly 9 dB. With motion compensation, longer averaging periods can also improve SNR.”

Instead of broad-beam illumination, we exploit narrow-beam illumination with fast scanning. At every single beam illumination, we **maximize the laser fluence at the sample surface to 20 mJ/cm²**. Then, using multiple (20) illuminations, we achieve the required illumination area. Thus, we operate at optimal irradiation conditions to maximize the PA signal amplitude representing the sum of partial PA signals obtained from individual firings.

In other words, our system has a lower SNR by a factor of \sqrt{N} , where N – number of fibers, not number of laser firings. Note, that some optimal weighting of partial image summation as a function of position and fiber index can reduce this difference. In addition, energy fluctuations in the laser pulse can be neglected compared to single pulse illumination because the signal is averaged over 20 fiber illuminations, which is very critical for spectral decomposition. **Of course, we have a lower SNR** (again, as mentioned in the Discussion), **but not dramatically and the fast-sweep approach enables full integration with clinical US, laser fluence compensation, motion correction, a much smaller system footprint, and much cheaper system cost.**

6) Regarding Comment 5, I believe the authors could improve their SNR in the future by using multiplexed illumination. Instead of firing from each fiber at a time, one could use a set of illumination patterns that span a basis. Then, one could algorithmically deduce the response that would have been obtained if the fibers were used sequentially. In such an approach, 50% of the fibers could be used with each pulse to deliver more light to the tissue.

No. This idea will not work out because it assumes a large energy laser, e.g. the energy equal to the sum of energies of single-fiber firings. If we split the laser energy between fibers, the laser fluence will drop down proportionally and thus makes the SNR much worse. Again, in each single illumination, we reach the maximum allowed laser fluence to maximize the acoustic pressure.

Theoretically, 2 lasers can be combined, but this will make the imaging system extremely expensive and cumbersome. We think that it would be a step backwards.

Minor comments:

Line 94: The MPE is wavelength dependent. For 700 nm, the limit is 200 mW/cm² and it drops as the wavelength is increased. This means that the optimal SNR is obtained for 10 Hz.

We agree with the Reviewer in general, but disagree with how this limit applies to our PAUS system.

The MPE exposure is regulated by ANSI standards [8R].

[8R] <https://www.lia.org/resources/laser-safety-information/laser-safety-standards/ansi-z136-standards>

Table 5: MPE for Skin Exposure to a Laser Beam (ANSI Z136.1-2007)¹

Wavelength λ (nm)	Exposure Duration t (s)	MPE		Notes
		J/cm	W/cm ²	
UV				
180-302	10 ⁻⁹ to 3 E4	3 E-3		For all wavelengths, MPE is listed value or 0.56t ^{0.25} whichever is lower. 3.5 mm limiting aperture.
303	10 ⁻⁹ to 3 E4	4 E-3		
304	10 ⁻⁹ to 3 E4	6 E-3		
305	10 ⁻⁹ to 3 E4	1.0 E-2		
306	10 ⁻⁹ to 3 E4	1.6 E-2		
307	10 ⁻⁹ to 3 E4	25 E-3		
308	10 ⁻⁹ to 3 E4	40 E-3		
309	10 ⁻⁹ to 3 E4	63 E-3		
310	10 ⁻⁹ to 3 E4	0.1		
311	10 ⁻⁹ to 3 E4	0.16		
312	10 ⁻⁹ to 3 E4	0.25		
313	10 ⁻⁹ to 3 E4	0.40		
314	10 ⁻⁹ to 3 E4	0.63		
315-400	10 ⁻⁹ to 10	0.56t ^{0.25}		
315-400	10 ⁻⁹ to 10 ³	1		
315-400	10 ³ to 3 E4		1 E-3	
Visible and Near IR				
400-1400	10 ⁻⁹ to 10 ⁻⁷	2 C _A E-2		3.5 mm limiting aperture.
400-1400	10 ⁻⁷ to 10	1.1 C _A t ^{0.25}		
400-1400	10 to 3 E4		0.2 C _A	
Far IR				
1400-1500	10 ⁻⁹ to 10 ⁻³	0.1		
1400-1500	10 ⁻³ to 10	0.56t ^{0.25}		
1400-1500	10 ⁻³ to 3 E4		0.1	
1500-1800	10 ⁻⁹ to 10	1.0		
1500-1800	10 to 3 E4		0.1	
1800-2600	10 ⁻⁹ to 10 ⁻³	0.1		
1800-2600	10 ⁻³ to 10	0.56t ^{0.25}		
1800-2600	10 ⁻³ to 3 E4		0.1	
2600-1 E6	10 ⁻⁹ to 10 ⁻⁷	1 E-2		
2600-1 E6	10 ⁻⁷ to 10	0.56t ^{0.25}		
2600-1 E6	10 to 3 E4		0.1	

When the irradiation occurs for more than 10 seconds, the maximum laser fluence can be 20 mJ/cm² at 700 nm wavelength, then growing to reach 100 mJ/cm² at 1064 nm. For those conditions, the max fluence rate is 200 mW/cm² at 700 nm then growing to reach 1 W/cm² at 1064 nm. Thus, to reach both requirements, 10 Hz should be used to keep the maximum possible fluence, i.e. maximize the amplitude of generated PA signals. **These MPE values are cited as a fact in many papers, but most of them forget to mention that irradiation should occur for more than 10 seconds.** Using 10 Hz with 10 wavelengths, however, will induce serious motion artifacts which will decorrelate US frames and make motion correction less accurate.

In fact, if we look at the ANSI standards carefully (see the table above), for exposures less than 10 seconds, the MRE fluence **rate** limits depend on time:

$$MPE = 1.1 C_A * t^{0.25} \text{ W/cm}^2,$$

C_A is a wavelength dependent coefficient, which is equal to 1 at 700 nm wavelength; and the fluence per pulse remains the same.

Thus, if the exposure can be controlled in a fixed position, the allowed exposure limits can be much higher than 200 mW/cm².

A reasonable value for the exposure can be 1 sec for a fixed spot, which will allow 1.1 W/cm², this is what we referenced to. Starting and stopping the exposure can be fully managed in the proposed PAUS concept because, again, the laser is driven (triggered) by the US scanner.

We have now added all necessary details and additional references.

Line 211: The authors write “If fluence is ignored, accurate molecular imaging based on spectral decomposition is nearly impossible.” This statement is a bit strong and inconsistent with some of the results presented in the paper and previous works (see Comment 4). In some cases, one may ignore the fluence (superficial imaging, tissue that is not well perfused with blood, or above 950 nm, where the absorption spectra is mostly due to water). I would recommend making this statement more specific, and focus it on deep-tissue imaging

We agree with the Reviewer about this statement, but of course it was made in the context of deep tissue imaging. We also mention multiple times that in many situations, like imaging of nude mice, the fluence change with wavelength can be ignored. Nevertheless, we corrected this sentence to make it clearer and thank the Reviewer for pointing it out.

On behalf of authors,

Dr. Ivan Pelivanov

05/14/2020

Reviewers' Comments:

Reviewer #1:

Remarks to the Author:

The manuscript describes a new approach for multi-wavelength photoacoustic imaging at a very fast rate, further including a motion correction step. The manuscript has previously been submitted to the journal and answers to the comments from three independent reviewers are also provided in the current version. It appears that significant engineering efforts have been devoted to the development of the new system (the authors mentioned 8 years of development in the answer to one of the reviewers). Based on my reading of the paper and the comments from the previous reviewers, I can make the following suggestions to improve the manuscript.

Generally, the authors provide a very comprehensive answer (62 pages) to the previous reviewers' comments and in many cases very valuable explanations are given. However, most of these explanations have not been included in the manuscript, at least not in the changes marked in red in the main text. I would recommend that these points are better discussed in the text. I try to be more specific in the following.

I agree with Reviewer #2 that the system can be used for imaging humans as well as small animals, e.g. mice, and it would be a valuable tool in both cases. In this regard, I would not focus in the clinical aspect, which in any case has not been demonstrated in the manuscript. For example, a more generic title like "Real-time spectroscopic photoacoustic/ultrasound (PAUS) scanning with simultaneous fluence compensation and motion correction" could be considered.

I also agree with Reviewer #2 that there are many challenges not discussed in the manuscript related to the clinical translation of photoacoustic imaging. The technology is there and although the system suggested by the authors may provide some advantages, availability of the technology, regulatory aspects and many other issues are also important. In the manuscript, the authors just make a very generic statement "methods have generally not translated well to humans due to serious challenges in the clinical environment". The challenges associated to clinical translation need to be much more thoroughly discussed.

I have trouble finding the technical novelties of the system with respect to what has been reported in the literature, including the authors' own work. I think the reason for this is that previous work has barely been mentioned and discussed in the manuscript. Again, this was pointed out by the previous reviewers and references were discussed in the authors' answers but not in the text. I can also suggest here some previous works that I think are relevant. Photoacoustic imaging at kHz rates has been achieved (e.g. Sivasubramanian et al. High frame rate photoacoustic imaging at 7000 frames per second using clinical ultrasound system. *Biomedical optics express*, 7(2), 312-323 (2016), Ozbek et al. Optoacoustic imaging at kilohertz volumetric frame rates. *Optica*, 5(7), 857-863 (2018)). Other arrays for clinical imaging based on concave or combined linear/concave geometries have also been reported for photoacoustic-ultrasound imaging (Schellenberg et al. Hand-held optoacoustic imaging: A review. *Photoacoustics*, 11, 14-27 (2018), Mercep et al. Combined pulse-echo ultrasound and multispectral optoacoustic tomography with a multi-segment detector array. *IEEE transactions on medical imaging*, 36(10), 2129-2137 (2017)). There is also an extensive literature on the acceleration of algorithms in graphics processing units and in multispectral and motion-correction approaches. I think there is definitely value in the method suggested by the authors, but it is essential that this is properly contextualized and previous work is properly discussed.

A common question of all 3 reviewers (and a common answer to all of them) was related to the timing of image formation, i.e., how much time it takes to acquire and reconstruct/process images. I believe this is very relevant and I suggest that a schematic description of this process and the associated timing be included in a figure.

As I mentioned before, I don't think any clinical application (or potential clinical application) has been demonstrated in the manuscript. As correctly pointed out by one of the previous reviewers, gold nanorods and ink (used in the manuscript) cannot be injected in humans. Generally, no human images have been reported in the manuscript. Therefore, I strongly suggest that the paper is defined as a new general tool for photoacoustic imaging (which has value) and not as a clinical imaging system.

Minor comment. The configuration of fibers in Fig. 2b does not represent the actual distribution on the linear array, which is confusing. I would suggest modifying this figure.

Reviewer #2:

Remarks to the Author:

1. Novelty and impact: It was stated previously that much of the work described in this manuscript is based on previously described methodology; by this it was intended to mean that the essential elements of the underlying principal concepts had been described before. For example, the principle of fluence compensation was described previously in reference 33 (Held, K. G., et al Photoacoustics, (2016), 4(2), 70–80). The authors have indeed made advances in the practical execution of this approach to achieve fast acquisition, a convenient array based implementation and modifications to the algorithmic framework. However, these are largely engineering developments that may have involved significant effort and technical ingenuity but do not necessarily equate to conceptual novelty. The use of a linear array to acquire dual photoacoustic and ultrasound images has been demonstrated previously extensively, although the current work goes beyond these demonstrations in terms of practicality, multiwavelength acquisition speed and motion correction. It is suggested that the laser is a novel component. This may be so but it was supplied by a company and few technical details relating to its design are provided on the grounds they are proprietary and cannot be disclosed.

The above does not call into question the significant time and effort that has been devoted to developing the system. However, the technical achievement (significant as it is) seems to derive more from the adaptation and application of existing concepts than the development of new ones. That in itself is not necessarily a showstopper however. The potential for clinical impact is also important and this is defined largely by the system performance. The photoacoustic images provided by the system seem only to reveal exogenous absorbers that are very strongly absorbing – eg the metal needles and the gold nanorod suspensions that were used in the current study. Unlike most photoacoustic imaging systems, the acquired in vivo images do not appear to reveal the vasculature or other anatomy based on endogenous contrast. This suggests limited sensitivity which limits the breadth of potential clinical applications. Visualizing the location of needles and other surgical implements is a potentially significant application as the authors suggest. However, needle visualization has been convincingly demonstrated previously using dual photoacoustic-ultrasound scanners (eg references 28 and 55). The spectroscopic approach adopted may indeed provide a more specific identification of the needle as evidenced by the images in figure 4 but is it really that much better than the above previous non-spectroscopic approaches? These previous demonstrations did not spectroscopically localize an absorbing contrast agent delivered by the needle as the current work does and this could be useful for visualizing the distribution of labelled injected drugs. However, if this is the main advance over previous work in terms of "user" functionality (user being the operative word here), then it seems quite a modest step.

2. The authors state in their response; "Saying that the importance of significantly improving millions of clinical procedures per year is unclear seems off the mark". If the implication is that the reviewer made this comment it is a misrepresentation – the reviewer made no such statement.

3. Page 2, line 32 – "However, spatially resolved optical spectroscopy is not used routinely in vivo because high tissue scattering typically limits coherent penetration to a millimeter or less" This

sentence remains problematic. As stated previously, near infrared optical spectroscopy is widely used in vivo for making hemodynamic measurements using single channel spectroscopy systems or multichannel topography/tomography instruments. Multichannel systems can provide spatially-resolved measurements (see section 3 of the following review article: Hillman, E. M. C. (2007). *Journal of Biomedical Optics*, 12(5), 051402) and are widely available commercially from companies such as Hitachi and Hamamatsu. Moreover, by using near infrared light these systems can provide penetration depths much greater than 1 mm, in some cases several centimetres. The relevance of coherence raised by the authors in this issue is not clear. Spatial or temporal coherence (beyond the need for reasonably narrow linewidth) is not a requirement for the above optical spectroscopic methods, in part because they are typically based on diffuse optical techniques. The authors refer to OCT-based methods in their response but in vivo optical spectroscopy does not usually employ an OCT-like approach.

As a general point, if temporally coherent light is emitted by a source, it can remain temporally coherent even after multiple scattering at depths well beyond 1mm in turbid media such as tissue. If that were not the case, it would not be possible to create a speckle pattern within tissue which is perfectly possible even when the light is diffuse. Hence the suggestion that coherent light penetration in tissue is limited to 1mm does not seem correct. Perhaps it is question of terminology and the authors are referring to ballistic photons which have a penetration limit of this order ?

4. Page 2, line 55 – "...and associated methods have generally not translated well to humans due to serious challenges in the clinical environment." The meaning of this sentence is vague. What are these serious challenges ?

5. Page 4, line 80 – the meaning of "molecular profile" in this context is not clear.

6. Page 5, line 91 – as stated in the previous review, the claim that 5 wavelengths are needed for a spectroscopic decomposition requires careful qualification but this has not been provided. It is fine to say that 5 (or however many) were used in the current study and provide a justification for this choice but not to suggest it is a generally applicable rule as the current text does.

7. Page 5, line 93 - it is stated that "motion changes the local concentration of absorbers". As discussed previously, alternative wording would be clearer.

8. Page 10, line 215 – it is implied that ignoring the fluence is more of a problem in humans than in mice in spectroscopic photoacoustic techniques. This issue was raised in the previous review and discussed at length in the authors' response. There are several issues arising here.

- From the authors' response, there appears to be the suggestion that because mice can be illuminated from all directions and optical attenuation is less than in humans, this somehow reduces the need to compensate for the wavelength dependence of the fluence distribution in spectroscopic methods. It is not clear why this should be. The wavelength dependence of the fluence originates from the spectral properties of tissue which are essentially the same for mice and humans; mouse tissues contains hemoglobin, water, lipids, cellular scattering etc just as human tissue does. Hence, even if it could be arranged somehow for the internal light distribution to be spatially invariant, the fluence would still be wavelength dependent and distort the measured photoacoustic spectrum compromising a spectral decomposition. To illustrate this, consider a blood vessel of interest at the centre of a mouse. Assume uniform illumination from all directions and that mouse tissue optical attenuation is very weak so that the 3D internal light distribution is near uniform. The light travels from the surface of the mouse through the tissue surrounding the vessel accumulating spectral information which is then imposed on the vessel photoacoustic spectrum thereby distorting it. The fact that the mouse is illuminated from all directions and the depth dependence of the fluence is negligible is irrelevant. It does not remove the corrupting influence of the wavelength dependence of the fluence, or even reduce it compared to a non-uniform fluence

distribution. It could even make it worse. The fluence wavelength dependence can only be neglected if all the constituents of the tissue surrounding the vessel have a flat absorption and scattering spectrum, which does not arise in mammalian tissues. The only other case where the fluence wavelength dependence can perhaps be ignored is for highly superficial depths ($<1\text{mm}$) as is often assumed (but rarely proven) in optical resolution photoacoustic microscopy. However even in this case it applies equally to both mice and humans.

- It is correct that the mouse offers the opportunity to reduce limited view problem by recording the photoacoustic signals over a larger solid angular aperture than possible in humans in most cases. As the authors point out, this yields a more accurate structural image with fewer artefacts. This is certainly conducive to increasing the accuracy of spectroscopic methods but not because it mitigates the negative impact of the fluence wavelength dependence - it has no impact on this. Even if the image is perfectly reconstructed, the deleterious effect of the wavelength dependence of the fluence remains.

- Mouse imaging is not only achieved using cylindrical or spherical detection geometries as implied. As evidenced by the widespread use of the popular Visualsonics photoacoustic imaging system, it is often achieved using a linear array, a geometry that is translatable to human use.

- It is suggested that mouse imaging is merely an intermediate step to clinical imaging. In fact mouse models are widely used in research and characterising them using imaging techniques to provide biological insights is an important application in its own right – presumably this is why companies such as iThera and Visualsonics sell dedicated small animal photoacoustic scanners.

9. Page 10, line 222 – perhaps make it clear that the “measured spectrum” is the spectrum after the fluence compensation has been applied.

10. Page 18, line 416-424 – again the statement suggesting the fluence wavelength dependence is “small for some animal models” is contentious (see earlier comment) as is the implication that the lower depth dependence of the fluence in mice compared to humans somehow reduces the negative impact of the fluence wavelength.

11. Page 21 – Methods; please state the laser is a Ti:Sapphire laser pumped by the second harmonic of the Nd:YLF laser. The authors response on this point refers to an OPO laser resonator but presumably this is in error as there is no OPO in the system.

12. Page 23, line 582 – it is claimed that the methods described in references 32-40 require a priori knowledge of the tissue absorption and scattering properties. Please consider whether this statement is strictly accurate for all of these references. If the tissue optical coefficients are already known then it is not clear what these methods seek to achieve.

13. Page 23, line 588 – it is suggested that methods that use the inversion of light transport models are very sensitive to geometrical parameters. If the model can account for an arbitrary 3D distribution of absorption and scattering coefficients, why should this arise ?

14. Page 24, line 606 – again, it is not clear why imaging at depths $<5\text{mm}$ should not require fluence compensation.

Reviewer #3:

Remarks to the Author:

The authors have made significant improvements to their manuscript, highlighting its innovation, and better explaining many of the technical details, some of which were, in my opinion, difficult to understand in the previous version. The imaging approach developed by the authors is

unconventional, both conceptually and in its hardware, and goes against the common wisdom in our field. Point-wise illumination is technically difficult, requiring higher-rates lasers and sampling, and leading to lower SNR. However, as the authors now explain, the reduction in SNR is not as bad as one might think owing to the higher MPE of their approach. In addition, the proposed method has the benefit of clutter reduction, which was not discussed in the previous version. In terms of performance, it is now clear that the authors have achieved real-time operation. This is crucial because the main objection to the proposed approach was that only its data-acquisition was real-time (rather than the entire operation), raising the concern that it could not be adapted to clinical applications.

In light of the changes made and the rebuttal letter, I feel more strongly that this work is of outstanding quality, and would be of high interest to our research community. First, it goes against the mainstream, demonstrating a clinically compatible system that uses ideas that had been previously thought as theoretically interesting but impractical. Second, it offers a solution to some of the fundamental challenges of our field. As other reviewers have mentioned, the solution is not perfect, and there are trade-offs (as there always are – a clinical transducer for pulse-echo US will never be fully optimized for optoacoustics, and an ideal optoacoustic system can't do US well), but the results demonstrated in this work clearly indicate that the system can perform molecular imaging under clinical settings, even if not fully quantifiable. Third, as now explained in the rebuttal letter, the work includes the theoretical innovation of extending the concept of illumination sweeping to 3D. Accordingly, I fully support the publication of this manuscript in Nature Communications.

I have only two minor comments the authors might want to address:

1) It is now much clearer in the text that the MPE we usually use in optoacoustic systems is not the same as the one relevant to this system owing to the short time each spot is illuminated. However, this subject should be treated with caution. Because of light and thermal diffusion, there could be some interaction between the illumination of neighboring spots, in which the same superficial region is sequentially heated, whereas the MPE quoted assumes that no additional illumination is provided in the vicinity of the region of interest. Of course, one can easily overcome this issue by alternating between the two sides of the transducer and partially randomizing the illumination sequence in each side. I believe that a short discussion of this point would be beneficial. Yes, illumination sweeping enables using higher MPEs (which is a novel feature that mitigates the loss of signal), but reaching the highest MPE theoretically possible might require re-engineering of the illumination sequence. In case of journal length limitations, this discussion could replace some of the details of the MPE calculations, which could be moved to the Supplementary Information.

2) I fully agree with the authors that there is a big difference between demonstrating a concept in a simple measurement, and actually making it compatible with clinical applications. Indeed, all the works that previously used swept illumination for quantification involved impractical setups, focusing on proof-of-concept. However, as the authors now explain in their rebuttal letter, that in order to achieve a clinically compatible device they had to change the geometry proposed in previous schemes and develop a new theory for off-axis illumination. I believe it should be mentioned in the main text that this is an entirely new illumination configuration, which is off-axis and therefore requires a new theoretical framework and a new quantification algorithm. At least when I first read the manuscript, my initial impression was that all the novelty was in the hardware, but this algorithmic novelty is important even if its main purpose is to accommodate the practical geometry of handheld US transducers.

Response to Reviewers' Comments

“Real-time spectroscopic photoacoustic/ultrasound (PAUS) scanning with simultaneous fluence compensation and motion correction”

This letter addresses all comments made by the reviewers for the manuscript entitled **“Real-time spectroscopic photoacoustic/ultrasound (PAUS) scanning with simultaneous fluence compensation and motion correction”**. We have revised the manuscript to address the reviewers' comments and would like to thank them for carefully reading the paper and providing very useful comments.

In this letter, we respond to **each comment** made by the reviewers.

Reviewer #1:

The manuscript describes a new approach for multi-wavelength photoacoustic imaging at a very fast rate, further including a motion correction step. The manuscript has previously been submitted to the journal and answers to the comments from three independent reviewers are also provided in the current version. It appears that significant engineering efforts have been devoted to the development of the new system (the authors mentioned 8 years of development in the answer to one of the reviewers). Based on my reading of the paper and the comments from the previous reviewers, I can make the following suggestions to improve the manuscript.

Generally, the authors provide a very comprehensive answer (62 pages) to the previous reviewers' comments and in many cases very valuable explanations are given. However, most of these explanations have not been included in the manuscript, at least not in the changes marked in red in the main text. I would recommend that these points are better discussed in the text. I try to be more specific in the following.

We thank the Reviewer for valuable comments. We have modified the text according to your suggestions below.

I agree with Reviewer #2 that the system can be used for imaging humans as well as small animals, e.g. mice, and it would be a valuable tool in both cases. In this regard, I would not focus in the clinical aspect, which in any case has not been demonstrated in the manuscript. For example, a more generic title like **“Real-time spectroscopic photoacoustic/ultrasound (PAUS) scanning with simultaneous fluence compensation and motion correction”** could be considered.

We thank the Reviewer for valuable comments. Using a more generic title is a good suggestion. Previously we narrowed our title to interventional procedure guidance because one of the reviewers has concerns about the terminology of molecular imaging. We think deleting **“for interventional**

procedure guidance” from the title is definitely more appropriate for readers. Thus, we also agree to change the title to

“Real-time spectroscopic photoacoustic/ultrasound (PAUS) scanning with simultaneous fluence compensation and motion correction”

I also agree with Reviewer #2 that there are many challenges not discussed in the manuscript related to the clinical translation of photoacoustic imaging. The technology is there and although the system suggested by the authors may provide some advantages, availability of the technology, regulatory aspects and many other issues are also important. In the manuscript, the authors just make a very generic statement “methods have generally not translated well to humans due to serious challenges in the clinical environment”. The challenges associated to clinical translation need to be much more thoroughly discussed.

From our point of view, one of the main advantages of PA imaging is the possibility to provide spectroscopic information; that is, to add another image dimension. However, as we show in our paper, wavelength- and depth-dependent laser fluence variations and motion artifacts may greatly limit spectroscopic accuracy. Thus, laser fluence compensation and tissue motion correction are the primary physical challenges.

We definitely agree with the reviewers that there are many other challenges in translating PA imaging to medical use, including regulatory issues, educating personnel, reducing system cost and others that must be solved. These challenges are well discussed in [13]. However, in our paper, these problems are not within the scope and, thus, discussing them in any detail within the introduction would not help readers understand the solutions we propose. We do not solve all the problems of PA imaging, but address two critical ones for clinical translation.

We modified the Introduction to make these points clearer (see below). We thank the reviewer for the help.

‘Although motion artifacts and wavelength-dependent fluence variations are the largest technical hurdles to clinical adoption of spectroscopic PA imaging, there are many other challenges, as detailed in [13]. In addition, regulatory issues, educating personnel, and high PA system cost are additional barriers to widespread clinical translation.’

I have trouble finding the technical novelties of the system with respect to what has been reported in the literature, including the authors’ own work. I think the reason for this is that previous work has barely been mentioned and discussed in the manuscript. Again, this was pointed out by the previous reviewers and references were discussed in the authors’ answers but not in the text. I can also suggest here some previous works that I think are relevant. Photoacoustic imaging at kHz rates has been achieved (e.g. Sivasubramanian et al. High frame rate photoacoustic imaging at 7000 frames per second using clinical ultrasound system. Biomedical optics express, 7(2), 312-323 (2016), Ozbek et al. Optoacoustic imaging at kilohertz volumetric frame rates. Optica, 5(7), 857-863 (2018)). Other arrays for clinical imaging based on concave or combined linear/concave geometries have also been reported

for photoacoustic-ultrasound imaging (Schellenberg et al. Hand-held optoacoustic imaging: A review. Photoacoustics, 11, 14-27 (2018), Mercep et al. Combined pulse-echo ultrasound and multispectral optoacoustic tomography with a multi-segment detector array. IEEE transactions on medical imaging, 36(10), 2129-2137 (2017)).

Previous PA imaging approaches at kHz rates were applied only for a single (or a few) wavelength. In addition, all previous kHz-rate approaches simply proposed signal averaging with broad-beam illumination, leading to dramatic (equal to \sqrt{N}) SNR reduction. PA and US modalities were not previously synchronized at high repetition rates to enable motion correction. Fast-scanning, narrow swept-beam illumination was not considered as well. Overall, our approach differs markedly from previous methods using kHz rate illumination.

However, we agree with the reviewer that citing previous kHz-rate approaches, even if they don't address motion correction and fluence compensation, will improve our paper. We appreciate this suggestion. We added these points into the Discussion section.

There is also an extensive literature on the acceleration of algorithms in graphics processing units and in multispectral and motion-correction approaches. I think there is definitely value in the method suggested by the authors, but it is essential that this is properly contextualized and previous work is properly discussed.

We briefly discussed existing motion correction/compensation methods in Main. The cited references are numbered [43-46] ([55-58] for the latest revision). Regarding execution acceleration with GPUs, please note that this is not the main objective of the paper. Our purpose is to develop computationally efficient algorithms for both proposed techniques (fluence compensation and motion correction), which we showed can work in real-time. Using GPUs to speed up signal processing is beneficial but not essential. We added the timing diagram (Suppl. Fig. 3b) to Supplementary Note 3 to make these points clearer in our latest version.

A common question of all 3 reviewers (and a common answer to all of them) was related to the timing of image formation, i.e., how much time it takes to acquire and reconstruct/process images. I believe this is very relevant and I suggest that a schematic description of this process and the associated timing be included in a figure.

We appreciate this suggestion – the timing diagram was added to **Supplementary Note 3**.

As I mentioned before, I don't think any clinical application (or potential clinical application) has been demonstrated in the manuscript. As correctly pointed out by one of the previous reviewers, gold nanorods and ink (used in the manuscript) cannot be injected in humans. Generally, no human images have been reported in the manuscript. Therefore, I strongly suggest that the paper is defined as a new general tool for photoacoustic imaging (which has value) and not as a clinical imaging system.

We agree with the reviewer that our current system cannot be called 'clinical'. We continue to work with GE on integrating the solutions demonstrated in this study into a high-end, real-time US scanner for clinical applications. Thus, the correct statement is that we developed important solutions and tools for clinical translation of PAUS imaging. We applied necessary edits into the text and thank the reviewer for pointing this out.

Minor comment. The configuration of fibers in Fig. 2b does not represent the actual distribution on the linear array, which is confusing. I would suggest modifying this figure.

Indeed, the transducer size/color is a bit different in Fig.2a than the actual transducer described in detail in Supplementary Information. Note that the custom transducer used in our study has a commercial twin that is white in color with a slightly different shape much closer to the one shown in Fig.2a. Figure 2 correctly describes the fast-scanning principle. Illustrating our 'conventional' transducer shape and color should make it much easier for the reader to recognize the US probe in this diagram. We do not see an issue here because: (i) the actual transducer is described in detail in

Supplementary Information, and (ii) the fast-scanning principle is not limited by the transducer shape. The probe can be easily exchanged.

Reviewer #2:

1. Novelty and impact: It was stated previously that much of the work described in this manuscript is based on previously described methodology; by this it was intended to mean that the essential elements of the underlying principal concepts had been described before. For example, the principle of fluence compensation was described previously in reference 33 (Held, K. G., et al Photoacoustics, (2016), 4(2), 70–80). The authors have indeed made advances in the practical execution of this approach to achieve fast acquisition, a convenient array based implementation and modifications to the algorithmic framework. However, these are largely engineering developments that may have involved significant effort and technical ingenuity but do not necessarily equate to conceptual novelty.

We addressed this concern (made by Reviewer 1) in our previous rebuttal letter. We repeat it below:

“We disagree with the reviewer’s view on novelty of the method. The fast-sweep approach, i.e. using multiple low-energy illuminations instead of a single large energy illumination, was indeed demonstrated before in [R1].

[R1] Wei C.-W., Nguyen T.-M., Xia J., Arnal B., Wong E. Y., Pelivanov I., O’Donnell M., “Real-time integrated ultrasound and photoacoustic imaging system,” IEEE Tran UFFC **62**, 319-328 (2015).

The current realization, however, greatly differs from this initial description of the approach.

In our 2015 publication, we used scanning mirrors for beam delivery, which was impractical, and fast coupling into fibers was not discussed in any way.

Second, here we demonstrate the spectroscopic approach, which is much more complicated, especially in wavelength switching. In our concept, switching must be performed between firings. We did not discuss this possibility at all originally.

Third, we never published our approach to laser fluence compensation. We certainly understood the advantages of fast-sweep illumination for fluence correction, and developed the complete system described here to implement real-time fluence compensation, but we intentionally did not publish anything on our specific method until we developed the complete system to demonstrate real-time implementation. The Reviewer is correct that other groups published phantom experiments on fluence correction using point source illumination, but even the idea of integrating this into an imaging probe was not proposed. Other investigators proposed to switch between narrow-beam and broad-beam illumination, i.e. measure laser fluence before measurements, point-by-point with single fiber illumination. It’s not at all clear how this will work for clinical applications where tissue optical properties can change with time due to blood oxygenation change, for example. This is especially true for guiding interventional procedures, where the optical properties of the medium can change rapidly with time as a result of the procedure itself.

Fourth, we also show here that an arbitrary object in the PA image with amplitude above the noise level can be used for fluence compensation. We did it for two different absorbers with very different optical absorption spectra. In general, more imaging points contributing to the fluence estimate produce more accurate estimates.

Fifth, eq. S11 (see Supplementary Note 6) was previously derived for the 2-D case of sources located in the imaging plane and did not account for fiber inclination. We adopted this expression for the 3D case and account for the inclination of all fibers relative to the imaging plane. In addition, we derived a more general equation (eq. S7 - see Supplementary Note 6) producing simultaneous estimates of the reduced scattering coefficient, which is necessary for fluence correction when scattering is not very large. All these important features have not been presented in the previous literature on fluence compensation.

Sixth, we discussed our algorithms with Professor M. Frenz (the corresponding author of the Ref [45], i.e., (Held, K. G., et al Photoacoustics, (2016), 4(2), 70–80)) who is acknowledged.

Seventh, we developed a system for fast, sequential coupling of laser radiation into 20 fibers which is not easy taking into account ~ 1 kHz laser pulse repetition rate. This concept was not discussed in Ref [45] and any other previous studies. Coupling was fully synchronized with laser (required the specific laser which could operate at variable pulse repetition rate) with use of fast controllers. Without these developments, the concept [45] could not be applied.

Eighth, we first achieved the technical and algorithmic level of the fluence correction to work in real-time, which was not even discussed previously.

Clearly, we have not invented photoacoustic imaging. However, all papers on photoacoustic imaging published over the last 25 years, many in Nature journals, can be considered derivative work from A. Bell's original 1881 publication on the photoacoustic effect, and anticipated by Bowen's paper in 1981 on thermoacoustic imaging [R2]. We believe that the scientific approach and technological developments presented in this paper represent a significant step toward bringing PA spectroscopic imaging into the clinic for the important application of molecular guidance of interventional procedures.

[R2] Bowen T. "Radiation-induced thermoacoustic soft tissue imaging," Proc. IEEE Ultrasonics Symposium 2, 817-822 (1981).

It is suggested that the laser is a novel component. This may be so but it was supplied by a company and few technical details relating to its design are provided on the grounds they are proprietary and cannot be disclosed.

The corresponding author, Prof. I. Pelivanov, participated in the design of the laser, its calibration and testing. Overall, he significantly contributed to the scientific concepts leading to the specific system developed by a commercial enterprise. Clearly, some system details are proprietary to Laser-Export, but the overall concept of a high rep-rate, pulse-to-pulse wavelength tunable laser system for real-time PAUS imaging was driven by our group at the University of Washington, not the company. The main purpose of this paper is to show how this laser can be used for real-time PAUS imaging with simultaneous fluence compensation and motion correction.

The use of a linear array to acquire dual photoacoustic and ultrasound images has been demonstrated previously extensively, although the current work goes beyond these demonstrations in terms of practicality, multiwavelength acquisition speed and motion correction.

The use of a linear array in PA imaging is definitely not new. The Reviewer correctly says “although the current work goes beyond these demonstrations in terms of practicality, multiwavelength acquisition speed and motion correction”. We believe that addressing these two issues is a major step in PA imaging. Again, we demonstrate not only a concept, but a working PAUS system with (i) fully synchronized US and PA frames, (ii) motion correction, (iii) laser fluence compensation, (iv) multi-wavelength operation and (v) real-time implementation not only for acquisition, but also for signal processing and imaging (see **Supplementary Note 3**). We have spent 8 years from initial concept to working system. Again, **all** photoacoustic publications after the original studies of Alexander Graham Bell and Ted Bowen can be considered incremental improvements on the original concept according to the reviewer’s logic. Given the large number and variety of high impact photoacoustic publications over the last two decades, it’s hard to justify this logic.

The above does not call into question the significant time and effort that has been devoted to developing the system. However, the technical achievement (significant as it is) seems to derive more from the adaptation and application of existing concepts than the development of new ones. That in itself is not necessarily a showstopper however. The potential for clinical impact is also important and this is defined largely by the system performance.

We thank the Reviewer for his opinion.

The photoacoustic images provided by the system seem only to reveal exogenous absorbers that are very strongly absorbing – eg the metal needles and the gold nanorod suspensions that were used in the current study. Unlike most photoacoustic imaging systems, the acquired in vivo images do not appear to reveal the vasculature or other anatomy based on endogenous contrast. This suggests limited sensitivity which limits the breadth of potential clinical applications.

We recently published a paper where we show that vasculature can also be imaged with our system [R3].

The paper used deep-learning for advanced image reconstruction. Note that this paper was submitted to IEEE-TMI **after** the current paper had been submitted to Nature Communications. It was accepted very quickly compared to the long consideration process at NC. Thus, we do not cite this paper because of the logical priority of the current research in NC over our TMI paper. Sensitivity should not be an issue for depths up to a few cm, but the main issue is the artifacts produced by the limited view/bandwidth of the linear US probe. As demonstrated in our IEEE-TMI paper, such artifacts can be greatly reduced. Implementing this approach in our real-time PAUS system is definitely part of our future work.

[R3] Kim M., Jeng G.-S., Pelivanov I., O’Donnell M. “Deep-learning image reconstruction for real-time photoacoustic system”. IEEE TMI, 2020 (accepted for publication, available as “early access” <https://ieeexplore.ieee.org/document/9091172>).

Visualizing the location of needles and other surgical implements is a potentially significant application as the authors suggest. However, needle visualization has been convincingly demonstrated previously using dual photoacoustic-ultrasound scanners (eg references 28 and 55). The spectroscopic approach adopted may indeed provide a more specific identification of the needle as evidenced by the images in figure 4 but is it really that much better than the above previous non-spectroscopic approaches?

Yes, it is much better because of the high frame rates required for actual procedure guidance! In addition, we not only visualize the needle, but also the injected drug, which can be identified, imaged and characterized on a pixel-by-pixel basis. This is a much more robust approach that maps far better to real-time procedure guidance than previous methods presented in the literature.

These previous demonstrations did not spectroscopically localize an absorbing contrast agent delivered by the needle as the current work does and this could be useful for visualizing the distribution of labelled injected drugs. However, if this is the main advance over previous work in terms of “user” functionality (user being the operative word here), then it seems quite a modest step.

We believe that the identification of a drug, and its imaging and characterization on a pixel-by-pixel basis is a large step forward. Again, we do not present incremental ideas for procedure guidance, but rather a fully integrated PAUS imaging system enabling real-time data acquisition and image processing (including fluence compensation and motion correction, see **Supplementary Note 3**) that is well matched to the clinical needs for molecular procedure guidance.

2. The authors state in their response; “Saying that the importance of significantly improving millions of clinical procedures per year is unclear seems off the mark”. If the implication is that the reviewer made this comment it is a misrepresentation – the reviewer made no such statement.

We apologize for the misinterpretation.

3. Page 2, line 32 – “However, spatially resolved optical spectroscopy is not used routinely in vivo because high tissue scattering typically limits coherent penetration to a millimeter or less” This sentence remains problematic. As stated previously, near infrared optical spectroscopy is widely used in vivo for making hemodynamic measurements using single channel spectroscopy systems or multichannel topography/tomography instruments. Multichannel systems can provide spatially-resolved measurements (see section 3 of the following review article: Hillman, E. M. C. (2007). Journal of Biomedical Optics, 12(5), 051402) and are widely available commercially from companies such as Hitachi and Hamamatsu. Moreover, by using near infrared light these systems can provide penetration depths much greater than 1 mm, in some cases several centimetres. The relevance of coherence raised by the authors in this issue is not clear. Spatial or temporal coherence (beyond the need for reasonably narrow linewidth) is not a requirement for the above optical spectroscopic methods, in part because they are typically based on diffuse optical techniques. The authors refer to OCT-based methods in their response but in vivo optical spectroscopy does not usually employ an OCT-like approach.

We do not want to discuss the advantages and disadvantages of DOT since it is not truly germane to the overall approach presented in this paper. Note, however, that

- (i) Image reconstruction in DOT is an ill-conditioned and under-determined problem [R4-R8], whereas PA tomography is well conditioned and fully determined for full bandwidth/view acquisition when laser fluence is known. Even if the laser fluence is unknown, it does not affect image resolution; it affects quantification only.
- (ii) For image reconstruction in DOT, the detailed 3D structure of the object under study must be known [R4-R6], which cannot be obtained with DOT alone. PA imaging does not rely on the known structure because it does not require the solution of the forward model and uses US waves for reconstruction.
- (iii) DOT cannot image objects with high (sub-mm) resolution, which other clinical modalities (CT, PET, MR, US) can do. As discussed in [R5], the reliably resolved feature size for breast is 8-10 mm with a few cm penetration depths. In other words, the resolution is almost equal to depth in DOT. Even very optimistic arguments [R6] predict that a resolution of about 5 mm will be achieved in the next decade only for the best DOT imaging object – infant brain with well-mapped structure. Sub-mm resolution is not even discussed. In addition, DOT does not work near the surface because diffuse light is not yet formed. DOT cannot resolve vessels and capillaries. If the Reviewer means the 8-10 mm resolution is spatially resolved, we agree with him/her, but we report at least an order of magnitude higher resolution.
- (iv) DOT is very sensitive to motion and most head imaging is performed on sleeping infants.
- (v) DOT has definitely been successful in brain imaging, but it relies on a universal structural model of the neonatal head or one obtained with MRI. Extending DOT to other organs is not straightforward since it requires a detailed, patient-dependent model of the organ of interest and the overall imaging environment. However, DOT does have advantages over PA for brain imaging because PA has not yet been demonstrated for through-skull imaging of the human brain.
- (vi) A comprehensive discussion on other DOT issues can be found in [R8]

[R4] Puszka A, Di Sieno L, Mora AD, et al. “Spatial resolution in depth for time-resolved diffuse optical tomography using short source-detector separations”. *Biomed Opt Express*. **6**, 1-10 (2014).

[R5] Hoshi Y., Yamada Y., “Overview of diffuse optical tomography and its clinical applications”. *J. Biomed. Opt.* **21**(9), 091312 (2016).

[R6] Yamada Y., Okawa S. “Diffuse optical tomography: present status and its future”. *Opt. Rev.* **21**, 185–205 (2014).

[R7] Lee C., Cooper R. & Austin T. “Diffuse optical tomography to investigate the newborn brain”. *Pediatr. Res.* **82**, 376–386 (2017).

[R8] Boas D. A. et al., “Imaging the body with diffuse optical tomography”, in *IEEE Signal Processing Magazine*, **18**, 57-75, (2001).

Again, we believe this discussion is not relevant to our paper. DOT and PA methods were compared and discussed extensively in multiple previous studies. Thus, we modified the Introduction to resolve the Reviewer’s concern (mostly terminology).

‘Although recent progress in diffuse optical tomography (DOT) is encouraging, especially for functional brain imaging [2], this method faces fundamental challenges since image reconstruction is an ill-conditioned and under-determined problem and requires the detailed 3D structure of the object under study [3-5]. The spatial resolution in DOT is still very poor (8-10 mm) [3, 4] compared to other clinical imaging modalities (PET, CT, MRI and Ultrasound).’

As a general point, if temporally coherent light is emitted by a source, it can remain temporally coherent even after multiple scattering at depths well beyond 1mm in turbid media such as tissue. If that were not the case, it would not be possible to create a speckle pattern within tissue which is perfectly possible even when the light is diffuse. Hence the suggestion that coherent light penetration in tissue is limited to 1mm does not seem correct. Perhaps it is question of terminology and the authors are referring to ballistic photons which have a penetration limit of this order?

Yes, we mean ballistic photons because image reconstruction employing ballistic photons is similar to that in PA employing US waves; i.e., mathematically accurate and not ill-posed. We apologize for a misleading discussion far from the scope of this work. We edited the Introduction (mentioning DOT methods), focusing on the limited resolution of optical methods at large depths.

“Although recent progress in diffuse optical tomography (DOT) is encouraging, especially for functional brain imaging [2], this method faces fundamental challenges since image reconstruction is an ill-conditioned and under-determined problem and requires the detailed 3D structure of the object under study [3-5]. The spatial resolution in DOT is still very poor (8-10 mm) [3, 4] compared to other clinical imaging modalities (PET, CT, MRI and Ultrasound).”

4. Page 2, line 55 – “...and associated methods have generally not translated well to humans due to serious challenges in the clinical environment.” The meaning of this sentence is vague. What are these serious challenges?

We agree with the Reviewer and modified the Introduction to address this issue. We thank the Reviewer for pointing this out.

5. Page 4, line 80 – the meaning of “molecular profile” in this context is not clear.

To avoid confusion, we modify this wording to “**molecular constituent spectra**”.

6. Page 5, line 91 – as stated in the previous review, the claim that 5 wavelengths are needed for a spectroscopic decomposition requires careful qualification but this has not been provided. It is fine to say that 5 (or however many) were used in the current study and provide a justification for this choice but not to suggest it is a generally applicable rule as the current text does.

The original paper below suggests 5 wavelengths mainly based on ICG (indocyanine green).

[R9] Taruttis, A., Morscher, S., Burton, N. C., Razansky, D., Ntziachristos, V. Fast multispectral optoacoustic tomography (MSOT) for dynamic imaging of pharmacokinetics and biodistribution in multiple organs. *PloS One* 7(1), e30491 (2012).

To avoid confusion and be more specific, we modified the following sentence.

‘As suggested in [54], five (5) wavelengths are needed for stable spectral decomposition with FDA-approved ICG (indocyanine green) contrast agents.’

7. Page 5, line 93 - it is stated that “motion changes the local concentration of absorbers”. As discussed previously, alternative wording would be clearer.

We apologize that we forgot to modify the Figure caption. Motion, of course, does not change the concentration of absorbers in tissue, but it does change it in the PA image which is formed in the **coordinate system of the transducer**. In other words, the transducer is fixed in PAUS imaging; because tissue is moving, a pixel imaged at one wavelength moves to a different image point at a different wavelength. We modified the figure caption to make our statement correct and clear.

‘...As shown herein, tissue motion during spectroscopic acquisition can corrupt measurements of the concentration of different chromophores (blood and GNR, for example) in an image pixel...’

8. Page 10, line 215 – it is implied that ignoring the fluence is more of a problem in humans than in mice in spectroscopic photoacoustic techniques. This issue was raised in the previous review and discussed at length in the authors’ response. There are several issues arising here.

- From the authors’ response, there appears to be the suggestion that because mice can be illuminated from all directions and optical attenuation is less than in humans, this somehow reduces the need to compensate for the wavelength dependence of the fluence distribution in spectroscopic methods. It is not clear why this should be. The wavelength dependence of the fluence originates from the spectral properties of tissue which are essentially the same for mice and humans; mouse tissues contains hemoglobin, water, lipids, cellular scattering etc just as human tissue does. Hence, even if it could be arranged somehow for the internal light distribution to be spatially invariant, the fluence would still be wavelength dependent and distort the measured photoacoustic spectrum compromising a spectral decomposition. To illustrate this, consider a blood vessel of interest at the centre of a mouse. Assume uniform illumination from all directions and that mouse tissue optical attenuation is very weak so that the 3D internal light distribution is near uniform. The light travels from the surface of the mouse through the tissue surrounding the vessel accumulating spectral information which is then imposed on the vessel photoacoustic spectrum thereby distorting it. The fact that the mouse is illuminated from all directions and the depth dependence of the fluence is negligible is irrelevant. It does not remove the corrupting influence of the wavelength dependence of the fluence, or even reduce it compared to a non-uniform fluence distribution. It could even make it worse. The fluence wavelength dependence can only be neglected if all the constituents of the tissue surrounding the vessel have a flat absorption and scattering spectrum, which does

not arise in mammalian tissues. The only other case where the fluence wavelength dependence can perhaps be ignored is for highly superficial depths (<1mm) as is often assumed (but rarely proven) in optical resolution photoacoustic microscopy. However even in this case it applies equally to both mice and humans.

We strongly disagree with the Reviewer's analysis.

For simplicity, assume that the distribution of laser fluence is purely exponential with depth:

$$F(\lambda_1) = A_{01} * \exp(-\mu_{eff}(\lambda_1)d)$$

$$F(\lambda_2) = A_{02} * \exp(-\mu_{eff}(\lambda_2)d)$$

A_{01} and A_{02} are known constants and $F(\lambda_1)$ and $F(\lambda_2)$ are laser fluences at different optical wavelengths. Then, the ratio of these fluences is

$$F(\lambda_1)/F(\lambda_2) = \frac{A_{01}}{A_{02}} * \exp(-(\mu_{eff}(\lambda_1) - \mu_{eff}(\lambda_2))d)$$

Thus, the ratio of fluences taken for different wavelengths is a **depth-dependent function** (function of d). Let's assume that $\mu_{eff}(\lambda_1) = 1 \text{ cm}^{-1}$ and $\mu_{eff}(\lambda_2) = 2 \text{ cm}^{-1}$.

At the surface ($d=0$), the ratio of $F(\lambda_1)/F(\lambda_2) = 1$, i.e. fluence correction is NOT required.

At a depth of $d= 1 \text{ cm}$, the ratio of $F(\lambda_1)/F(\lambda_2) = 1/e$, i.e. fluence correction is required.

As depth increases, the ratio will diverge more and more from unity.

It is well-known in PA imaging that spectrum decomposition degrades with depth. Without any fluence correction it is still possible to perform spectral decomposition at shallow depths with manageable error.

The ratio of $F(\lambda_1)/F(\lambda_2)$ definitely depends on how strong these dependencies are with depth. We showed previously in Figure 3R (reproduced below), that the fluence distribution is much more homogeneous with depth for a small animal (when irradiated from all sides) rather than in a typical clinical situation (when irradiated from one side). The ratio of fluences in the center of the cylinder will vary between 1 and -4 dB for different wavelengths in a mouse, and will vary between 1 and -18 dB in a 'human' within the wavelength range under consideration. That is, at a different wavelength, it may be -2 dB versus -10 dB respectively, but within the max change (-4 dB for a mouse, and -18 dB for a human). A -4 dB error range over imaging wavelengths may still be small enough to decompose spectral constituents with reasonable error, but decomposition will definitely fail for -18 dB inaccuracies.

Figure 3R. Comparison of single side tissue illumination (case of a handle-held probe) and uniform illumination of tissue from all directions (typical for PA imaging of mice).

Coming back to the Reviewer’s concern... We agree that fluence compensation is also important for small animals, but has a much higher effect for human. **We thank the reviewer for emphasizing this point. Saying that proper fluence correction is important for small animals is definitely beneficial for our paper.** We just wanted to point out that this is a much more serious problem for a single-sided imaging geometry, which will be typical in the clinical environment. To avoid further discussion and not create a loop, we applied some edits in the paper which, we hope, will be acceptable. Now it reads as:

‘Even with PA’s remarkable success, validated clinical protocols, as well as system integration with clinical-grade probes, have been limited [35, 36]. The PA imaging geometry has been highly optimized for small animals [37-39], with several commercial systems developed specifically for

this application [40-42]. They provide almost uniform illumination of a mouse body with nearly full view signal detection, producing reconstructed PA images with spatial resolution approaching theoretical limits and minimum artifacts. Most of these conditions are hard to replicate for humans in routine clinical use for simple physical reasons. In particular, a mouse cross-section is on the order of a few light penetration depths and can be easily surrounded by the transducer array. Illumination from all directions distributes light in the background almost uniformly. The typical size of the human body is more than 40 light penetration depths and cannot be surrounded by the transducer array for efficient full-view reconstruction. Tissue illumination can be performed from one side only; thus, laser fluence will decay rapidly with depth. There is a fundamental difference in PA imaging of humans compared to mice, with the one major exception being the human breast' and

'Although compensating for wavelength-dependent light fluence is important even for the case of uniform illumination, this is a much more serious problem for a single-sided imaging geometry, which is typical of the clinical environment. To reconstruct the true absorption coefficient spectrum at large depths, local wavelength-dependent light fluence must be compensated.'

- It is correct that the mouse offers the opportunity to reduce limited view problem by recording the photoacoustic signals over a larger solid angular aperture than possible in humans in most cases. As the authors point out, this yields a more accurate structural image with fewer artefacts. This is certainly conducive to increasing the accuracy of spectroscopic methods but not because it mitigates the negative impact of the fluence wavelength dependence - it has no impact on this. Even if the image is perfectly reconstructed, the deleterious effect of the wavelength dependence of the fluence remains.

We absolutely agree with the Reviewer and we did not say anything different. We said that full view illumination reduces the effects of laser fluence. We also said that a different (full view/bandwidth) approach can work for small animals and is better than our limited view approach. That is, the approach we propose here is not optimal for small animal imaging but, vice versa, the full view/bandwidth approach (ideal for small animals) cannot be easily extended to humans.

- Mouse imaging is not only achieved using cylindrical or spherical detection geometries as implied. As evidenced by the widespread use of the popular Visualsonics photoacoustic imaging system, it is often achieved using a linear array, a geometry that is translatable to human use.

It is true that Visualsonics provides a photoacoustic system with a limited view/bandwidth transducer for small animal imaging, but they do not propose any fluence compensation. For Visualsonics' geometry, laser fluence will definitely face the same issues discussed in this paper. Systems from iThera are much more sophisticated in light delivery to small animals and nearly ideal for mice.

- It is suggested that mouse imaging is merely an intermediate step to clinical imaging. In fact mouse models are widely used in research and characterising them using imaging techniques

to provide biological insights is an important application in its own right – presumably this is why companies such as iThera and Visualsonics sell dedicated small animal photoacoustic scanners.

We agree with the Reviewer that small animal imaging is very important. However, again, iThera's full view/bandwidth system (inVision) with homogeneous illumination is a fully developed system appropriate for small animal imaging but is not easily translatable to humans. This is our point. Of course, our fluence compensation approach could be beneficial for iThera's inVision system, but the full view transducer geometry must be considered. We might collaborate with iThera in the future, but it is a different project. Note that Prof. Jeng (first author of our paper) already participated in some projects with iThera, and Prof. Pelivanov did some projects with Prof. Razansky (co-founder of iThera) in the past.

We applied some edits to the text to indicate the importance of fluence compensation for small animal imaging as well. We appreciate the Reviewer's comments.

9. Page 10, line 222 – perhaps make it clear that the “measured spectrum” is the spectrum after the fluence compensation has been applied.

The following sentence is modified according to the Reviewer's suggestion.

‘...component-weighted images are realized pixel-wise by the product of the $\Sigma\lambda$ -PA signal with the correlation of the **fluence compensated** spectrum with the ground-truth spectrum of a molecular constituent....’

10. Page 18, line 416-424 – again the statement suggesting the fluence wavelength dependence is “small for some animal models” is contentious (see earlier comment) as is the implication that the lower depth dependence of the fluence in mice compared to humans somehow reduces the negative impact of the fluence wavelength.

The lower depth dependence of laser fluence in mice compared to that for human (see Figure 3R above) does reduce the effect of laser fluence on absorption spectrum decomposition (see our explanation above). However, the word ‘small’ is not appropriate here – we agree with the Reviewer. We smoothed this phrase as well as similar phrases in the rest of the paper. We thank the Reviewer again for pointing this out and for reading our paper so carefully.

11. Page 21 – Methods; please state the laser is a Ti:Sapphire laser pumped by the second harmonic of the Nd:YLF laser. The authors response on this point refers to an OPO laser resonator but presumably this is in error as there is no OPO in the system.

Agreed. We apologize that we forgot to change it in the first revision. This time it is done in the Supplementary Note 1 as

‘A unique (see Specification sheet below), high pulse repetition rate (from single shot to 1000 Hz), wavelength tunable (700-900 nm) laser (Laser-Export, Moscow, Russia) was designed and built especially for the fast-sweep PAUS system. Its principle is based on pumping the Ti:Sapphire laser

head with the second harmonic of the Nd:YLF diode-pumped laser and using an acousto-optic filter for fast selection of wavelengths. All laser components are assembled within a compact block.'

In the main text we say '...a compact, high pulse repetition rate (from single shot to 1000 Hz), wavelength tunable (700-900 nm) Ti:Sapphire laser...'

12. Page 23, line 582 – it is claimed that the methods described in references 32-40 require a priori knowledge of the tissue absorption and scattering properties. Please consider whether this statement is strictly accurate for all of these references. If the tissue optical coefficients are already known then it is not clear what these methods seek to achieve.

We confirm. For laser fluence correction in a PA image, the knowledge of optical properties is required, but it is not enough. The optical properties are then used to model the laser fluence distribution inside tissue taking into account irradiation geometry, beam diameter and tissue structure.

13. Page 23, line 588 – it is suggested that methods that use the inversion of light transport models are very sensitive to geometrical parameters. If the model can account for an arbitrary 3D distribution of absorption and scattering coefficients, why should this arise?

These models can potentially handle arbitrary, but known, distributions, but that requires knowledge not only of the geometry of the background, but the geometry of all absorbers, which is usually unknown (without simultaneous information from 3D ultrasound, CT or MRI images). Because CT and MRI cannot be obtained in real-time, these inversion methods face severe translational issues.

14. Page 24, line 606 – again, it is not clear why imaging at depths <5mm should not require fluence compensation.

This is explained in answering #8 above. Briefly, this is because wavelength-dependent differences in fluence are small and manageable at shallow depths compared to deeper depths.

Reviewer #3:

The authors have made significant improvements to their manuscript, highlighting its innovation, and better explaining many of the technical details, some of which were, in my opinion, difficult to understand in the previous version. The imaging approach developed by the authors is unconventional, both conceptually and in its hardware, and goes against the common wisdom in our field. Point-wise illumination is technically difficult, requiring higher-rates lasers and sampling, and leading to lower SNR. However, as the authors now explain, the reduction in SNR is not as bad as one might think owing to the higher MPE of their approach. In addition, the proposed method has the benefit of clutter reduction, which was not discussed in the previous version. In terms of performance, it is now clear that the authors have achieved real-time operation. This is crucial because the main objection to the proposed approach was that only its data-acquisition was real-time (rather than the entire operation), raising the concern that it could not be adapted to clinical applications.

In light of the changes made and the rebuttal letter, I feel more strongly that this work is of outstanding quality, and would be of high interest to our research community. First, it goes against the mainstream, demonstrating a clinically compatible system that uses ideas that had been previously thought as theoretically interesting but impractical. Second, it offers a solution to some of the fundamental challenges of our field. As other reviewers have mentioned, the solution is not perfect, and there are trade-offs (as there always are – a clinical transducer for pulse-echo US will never be fully optimized for optoacoustics, and an ideal optoacoustic system can't do US well), but the results demonstrated in this work clearly indicate that the system can perform molecular imaging under clinical settings, even if not fully quantifiable. Third, as now explained in the rebuttal letter, the work includes the theoretical innovation of extending the concept of illumination sweeping to 3D. Accordingly, I fully support the publication of this manuscript in Nature Communications.

We thank the Reviewer for his/her comments on our revised manuscript.

I have only two minor comments the authors might want to address:

- 1) It is now much clearer in the text that the MPE we usually use in optoacoustic systems is not the same as the one relevant to this system owing to the short time each spot is illuminated. However, this subject should be treated with caution. Because of light and thermal diffusion, there could be some interaction between the illumination of neighboring spots, in which the same superficial region is sequentially heated, whereas the MPE quoted assumes that no additional illumination is provided in the vicinity of the region of interest. Of course, one can easily overcome this issue by alternating between the two sides of the transducer and partially randomizing the illumination sequence in each side. I believe that a short discussion of this point would be beneficial. Yes, illumination sweeping enables using higher MPEs (which is a novel feature that mitigates the loss of signal), but reaching the highest MPE theoretically possible might require re-engineering of the illumination sequence. In case of journal length limitations, this discussion could replace some of the details of the MPE calculations, which could be moved to the Supplementary Information.**

We agree with the Reviewer's comment in general. Indeed, the overlapping region inside tissue between two spatially neighboring fibers at a 1-kHz rate could be heated higher, and a randomized fiber scan sequence can mitigate this effect. However, MPE regulation applies only to the surface. We use a 550 um core fibers whereas the spacing between fibers is 1.5 mm, thus light density does not increase at the tissue surface. Thus, we are fine...

Tissue heating is usually extremely small, but if it is a concern, yes, rearranging fiber illumination can be easily done by changing the fiber sequence at the entrance to the fiber coupler.

Accordingly, based on the Reviewer's suggestions, we add the following sentences in Results:

'As mentioned in Main, because MPE depends on the irradiation time, we maximize the frame rate to be 50 Hz by sequentially scanning the fiber at a 1-kHz rate under MPE limits. When highly absorbing contrast agents are used, close illumination from successive spots may cause overheating on the tissue surface due to light and thermal diffusion. In this case, rearranging the fiber illumination sequence to 1, 15, 2, 14, etc. can be easily done at the entrance to the fiber coupler.'

- 2) I fully agree with the authors that there is a big difference between demonstrating a**

concept in a simple measurement, and actually making it compatible with clinical applications. Indeed, all the works that previously used swept illumination for quantification involved impractical setups, focusing on proof-of-concept. However, as the authors now explain in their rebuttal letter, that in order to achieve a clinically compatible device they had to change the geometry proposed in previous schemes and develop a new theory for off-axis illumination. I believe it should be mentioned in the main text that this is an entirely new illumination configuration, which is off-axis and therefore requires a new theoretical framework and a new quantification algorithm. At least when I first read the manuscript, my initial impression was that all the novelty was in the hardware, but this algorithmic novelty is important even if its main purpose is to accommodate the practical geometry of handheld US transducers.

We agree with the Reviewer's analysis and appreciate this comment. We added the following sentences in the MAIN accordingly.

'In addition, a novel off-axis fiber delivery system is developed as opposed to the clinically incompatible galvo-based scan approach [32] previously proposed by our group. Leveraging this configuration, we also present a new theoretical framework to quantify wavelength-dependent fluence variation among different fibers (Supplementary Note 6).'

Reviewers' Comments:

Reviewer #1:

Remarks to the Author:

The authors have addressed all my comments and suggestions

Reviewer #2:

Remarks to the Author:

Some points have been addressed, but in several instances misunderstandings seem to have arisen. Specific comments below:

1. On the question of novelty, the authors respond by simply copying and pasting much of the response from their previous rebuttal on this point. As noted before, the principal achievement of the current work appears to this reviewer to be one of incremental engineering steps in different areas, rather than the product of conceptual advances. Of course, the definition of incremental is open to debate but to imply that everything post Alexander Bell's discovery of the photoacoustic effect in 1880 is incremental in photoacoustic research is surely stretching a point. Examples of conceptual advances might be the exploitation of a new contrast mechanism or the development of new mathematics to reconstruct an image. By contrast, refining previously demonstrated concepts to make them suitable for practical in vivo use which is largely the case here represents more of an incremental engineering advance; the hardware required to implement the point source fluence correction method into the probe head would be just one such example of this. This is not a criticism of the significant engineering effort that underpins this work, nor is it to suggest that the advances made are not distinct from previous work or have not required elements of technical ingenuity. However, it remains the opinion of this reviewer that in conceptual terms, the overall novelty, which is one of the criteria for assessing manuscripts submitted to this journal, is modest. Of course, it is recognized there is a subjective element to this assessment although other reviewers have raised questions on this aspect.

2. In terms of user functionality, a question was raised as to how much additional clinical benefit derives from being able to localize a drug emerging from the tip of a delivery needle compared to visualising the needle alone. The authors respond by saying that they believe it is a large step forward but did not justify this claim. The point at issue here is that other photoacoustic imaging systems provide good real time visualization of needles so the key additional capability provided by the current system to the clinical user is the visualization of the drug distribution which previous systems did not. The question posed was how significant this specific additional functional capability is to the clinical end user - it is not a question about how fast the system is or any other performance specification or whether its workings are different to other systems.

3. Page 2 line 31 states "However, spatially resolved optical spectroscopy is not used routinely in vivo because high tissue scattering typically limits coherent penetration to a millimeter or less." The authors respond to the previous comment on this point by saying that a discussion of DOT is not relevant to the current paper. Indeed, it is not and was never suggested to be. The point made in the previous reviews is that the above sentence is misleading. The first part states that spatially resolved optically spectroscopy is not used routinely in vivo. This is very questionable. As mentioned before, there are many type of optical imaging instruments (a device that implements DOT is just one example) that provide spatially resolved optical spectroscopy in one form or another. Some indeed employ DOT to provide a 3D image, other are not DOT instruments as such but provide a 2D topographic image and yet others make measurements at just a few discrete points. These instruments have been developed over the last 50 years, an enormous number of in vivo research studies on animals and humans have been undertaken (on which there is a vast literature) and clinical instruments are commercially available and used clinically, eg in critical care. None of this is consistent with the blanket statement implying that spatially resolved optical

spectroscopy is not widely used in vivo. Of course optical spectroscopy has its limitations. One is that, although it can provide deep penetration (several cm) using near infrared (NIR) wavelengths, resolution is limited due to optical scattering. This prevents high resolution imaging and of course this is where photoacoustic imaging has a major advantage.

The second part of the sentence "high tissue scattering typically limits coherent penetration to a millimeter or less" is factually incorrect, especially in the context of the above methods. NIR optical spectroscopy techniques are limited by optical scattering but not primarily because scattering limits penetration depth (which can be multiple cm) but because it limits spatial resolution. It is not clear what is meant by "coherent penetration" but as mentioned in the previous review, the coherence of the light is not of great relevance here but in any case temporally coherent light can retain its coherence even after multiple scattering at depths well beyond 1mm in tissue.

There is no need for a lengthy discussion of this matter; it would suffice to simply point out that NIR optical spectroscopy is used in vivo but its resolution is limited by optical scattering and photoacoustic imaging overcomes this.

4. Point 8 of previous review. The issue here is whether a lower fluence depth dependence reduces the fluence wavelength dependence. The authors make a valid argument that this is true if the reduction in fluence with depth is due to reduced absorption/scattering alone. However it is not so obvious that it should also be the case when the depth dependence of the fluence is reduced only by careful choice of the illumination geometry since the fluence wavelength dependence is primarily a function of the spectral properties of tissue and light propagation distance. On the other hand, it could be argued that reducing the depth dependence of the fluence irrespective of how it is achieved compresses the extent of the wavelength dependence of the fluence and this would support the authors' assertion. To prove it either way would need a simulation, similar to that in Figure 3R but for two wavelengths for both single-sided and multi-sided illumination geometries*. That said, interesting as this point may be, since the claim in question is not explicitly made in the manuscript it is incidental now – the matter only arose in trying to understand the reasoning behind a now corrected questionable claim in a previous version of the manuscript.

*Perhaps this has been done? It is claimed that at the center of the cylinder, the ratio of fluences at different wavelengths for mouse and humans are different - 2dB versus -10dB - but no information is provided as to how these figures were estimated and what assumptions were made. It is not clear what the "human" circumstances of the simulations were. Illumination from all sides but with different optical properties as the "mouse" or same optical properties but single sided illumination or some combination of both? Without this information it is not possible to assess this analysis.

5. It is stated in several places that illuminating a mouse from all directions can distribute the light in the "background almost uniformly". If this is intended to suggest that the fluence in the center of a mouse is similar to that close to the surface, then it seems implausible. Illuminating from all sides can reduce the depth dependence of the fluence, compared to single side illumination but can it really be almost constant with depth in a real mouse in vivo? Prior work using multi-sided illumination does not support this with higher contrast evident at the surface compared to deeper regions – eg see references 59 and 39. It is recommended this claim is modified.

6. The work described in reference 59 on spectroscopically estimating blood oxygen saturation in mice clearly shows that accounting for the wavelength dependence of the fluence is essential even with multi-sided illumination; when the fluence wavelength dependence was ignored (as is the case when the linear unmixing method was used), the oxygen estimates exhibited very poor accuracy. This further undermines the suggestion that spectroscopic methods are much less challenging in mice because the fluence wavelength dependence is less of a problem. Indeed the entire mice vs human debate in the current work seems largely irrelevant and it is somewhat

baffling as to why it features so prominently. All discussion of mouse imaging could be removed entirely from the manuscript without diminishing the work in the slightest – in fact it would improve the clarity of the manuscript as this issue serves only to distract. The current system is an advance towards clinical photoacoustic imaging because it is a convenient fast hand-held probe suitable for human use. It also provides fluence compensation and this is very useful but that is not what makes it especially relevant for human use specifically as this feature is also needed for spectroscopic mouse imaging. That is OK - there is a still perfectly valid justification for authors' system without having to contrive an argument that fluence correction is needed more in humans than mice.

Reviewer #3:

Remarks to the Author:

The authors have addressed all my comments. I fully support this manuscript for publication.

Response to Reviewers' Comments

“Real-time spectroscopic photoacoustic/ultrasound (PAUS) scanning with simultaneous fluence compensation and motion correction”

This letter addresses all comments made by Reviewer #2 for the manuscript entitled **“Real-time spectroscopic photoacoustic/ultrasound (PAUS) scanning with simultaneous fluence compensation and motion correction”**. We have revised the manuscript to address Reviewer 2's comments and would like to thank them for carefully reading the paper and providing very useful comments.

In this letter, we respond to **each comment** made by Reviewer #2.

Reviewer #2:

1. On the question of novelty, the authors respond by simply copying and pasting much of the response from their previous rebuttal on this point. As noted before, the principal achievement of the current work appears to this reviewer to be one of incremental engineering steps in different areas, rather than the product of conceptual advances. Of course, the definition of incremental is open to debate but to imply that everything post Alexander Bell's discovery of the photoacoustic effect in 1880 is incremental in optoacoustic research is surely stretching a point. Examples of conceptual advances might be the exploitation of a new contrast mechanism or the development of new mathematics to reconstruct an image. By contrast, refining previously demonstrated concepts to make them suitable for practical in vivo use which is largely the case here represents more of an incremental engineering advance; the hardware required to implement the point source fluence correction method into the probe head would be just one such example of this. This is not a criticism of the significant engineering effort that underpins this work, nor is it to suggest that the advances made are not distinct from previous work or have not required elements of technical ingenuity. However, it remains the opinion of this reviewer that in conceptual terms, the overall novelty, which is one of the criteria for assessing manuscripts submitted to this journal, is modest. Of course, it is recognized there is a subjective element to this assessment although other reviewers have raised questions on this aspect.

This is a clear case of having to agree to disagree - we simply disagree with the Reviewer. We believe that the scientific approach and technological developments demonstrated in our paper represent a significant step toward bringing PA spectroscopic imaging into the clinic for the important application of molecular guidance of interventional procedures.

In the previous Rebuttal letter, we provided 8 reasons supporting both the scientific and engineering novelty of our paper. In addition, the corresponding author, Prof. I. Pelivanov, participated in the design of the laser, its calibration and testing. Overall, he significantly contributed to the scientific concepts leading to the specific system developed by a commercial enterprise. Clearly, some system details are proprietary to Laser-Export, but the overall concept

of a high rep-rate, pulse-to-pulse wavelength tunable laser system for real-time PAUS imaging was driven by our group at the University of Washington, not the company. Our arguments were certainly persuasive with the other two reviewers who had no further comments on novelty.

2. In terms of user functionality, a question was raised as to how much additional clinical benefit derives from being able to localize a drug emerging from the tip of a delivery needle compared to visualising the needle alone. The authors respond by saying that they believe it is a large step forward but did not justify this claim. The point at issue here is that other photoacoustic imaging systems provide good real time visualization of needles so the key additional capability provided by the current system to the clinical user is the visualization of the drug distribution which previous systems did not. The question posed was how significant this specific additional functional capability is to the clinical end user - it is not a question about how fast the system is or any other performance specification or whether its workings are different to other systems.

This comment from the Reviewer does not appear to make sense clinically. There is no question about the importance of image-guided drug delivery. Preclinically, image-guided drug delivery can be used for several different purposes, e.g. for monitoring biodistribution, target site accumulation, off-target localization, drug release and drug efficacy. Clinically, it holds significant potential for preselecting patients, can greatly improve spatially confined delivery of drugs and genes to target tissues while reducing systemic dose and toxicity, and opens many opportunities in oncology (for both diagnostics and therapy).

There are many papers published every year presenting both current and potential applications of image-guided drug delivery; the reviewer can easily find these paper using “image-guided drug delivery” to search.

The NIH in the United States has a full Study Section on “Imaging Guided Interventions and Surgery”, and many other Study Sections annually review multiple projects that focus on image-guided drug delivery. Overall, many institutes within the NIH fund numerous projects in this area. Clearly, this is a significant field of study as measured by NIH funding.

Many imaging modalities, including MRI and ultrasound, are used to guide procedures, not just photoacoustics. Clearly, this is a large field that is growing rapidly (e.g., see paper reference [68]).

Thus, we do not think that citing the benefits of directly visualizing the drug requires additional explanation.

3. Page 2 line 31 states “However, spatially resolved optical spectroscopy is not used routinely in vivo because high tissue scattering typically limits coherent penetration to a millimeter or less.” The authors respond to the previous comment on this point by saying that a discussion of DOT is not relevant to the current paper. Indeed, it is not and was never suggested to be. The point made in the previous reviews is that the above sentence is misleading. The first part states that spatially resolved optically spectroscopy in not used routinely in vivo. This is very questionable. As mentioned before, there are many type of optical imaging instruments (a device that implements DOT is just one example) that provide spatially resolved optical spectroscopy in one form or another. Some indeed employ

DOT to provide a 3D image, other are not DOT instruments as such but provide a 2D topographic image and yet others make measurements at just a few discrete points. These instruments have been developed over the last 50 years, an enormous number of in vivo research studies on animals and humans have been undertaken (on which there is a vast literature) and clinical instruments are commercially available and used clinically, eg in critical care. None of this is consistent with the blanket statement implying that spatially resolved optical spectroscopy is not widely used in vivo. Of course optical spectroscopy has its limitations. One is that, although it can provide deep penetration (several cm) using near infrared (NIR) wavelengths, resolution is limited due to optical scattering. This prevents high resolution imaging and of course this is where photoacoustic imaging has a major advantage.

We corrected our statement in the first paragraph of the Introduction as “Overall, infrared spectroscopy and tomography methods can provide a few cm light penetration into biological tissue, but their spatial resolution is limited by optical scattering and remains very poor (8-10 mm) [3, 4] compared to other clinical imaging modalities (PET, CT, MRI and ultrasound (US)).”...

Note, however, as the Reviewer correctly says, the 2D topography and DOT methods provide poor resolution. If the resolution is poor, how can the spectrum of a substance be correctly measured? It will be averaged with that of surrounding tissue preventing accurate spectroscopy. The Reviewer did not provide any references on the accurate spatially resolved spectroscopy inside biological tissues. Spectroscopy here means **accurate measurement of the optical spectrum in an arbitrary point of the investigated volume of tissue.**

Finally, as we also mentioned in our previous rebuttal, the discussion of the possibilities of optical spectroscopy methods is out of the scope of this work.

The second part of the sentence “high tissue scattering typically limits coherent penetration to a millimeter or less” is factually incorrect, especially in the context of the above methods. NIR optical spectroscopy techniques are limited by optical scattering but not primarily because scattering limits penetration depth (which can be multiple cm) but because it limits spatial resolution. It is not clear what is meant by “coherent penetration” but as mentioned in the previous review, the coherence of the light is not of great relevance here but in any case temporally coherent light can retain its coherence even after multiple scattering at depths well beyond 1mm in tissue.

There is no need for a lengthy discussion of this matter; it would suffice to simply point out that NIR optical spectroscopy is used in vivo but its resolution is limited by optical scattering and photoacoustic imaging overcomes this.

We corrected the sentence to make the statement clearer “Spatially resolved optical spectroscopy is not used routinely *in vivo* because high tissue scattering typically limits ballistic photon penetration to a millimeter or less”.

In addition we added a phrase as the Reviewer suggested (see our response to #3). This phrase covers what we wanted to state.

4. Point 8 of previous review. The issue here is whether a lower fluence depth dependence reduces the fluence wavelength dependence. The authors make a valid argument that this is true if the reduction in fluence with depth is due to reduced absorption/scattering alone. However it is not so obvious that it should also be the case when the depth dependence of the fluence is reduced only by careful choice of the illumination geometry since the fluence wavelength dependence is primarily a function of the spectral properties of tissue and light propagation distance. On the other hand, it could be argued that reducing the depth dependence of the fluence irrespective of how it is achieved compresses the extent of the wavelength dependence of the fluence and this would support the authors' assertion.

The depth dependence of laser fluence relates to its wavelength dependence. Imagine that tissue scattering and absorption do NOT depend on wavelength. In this case, there will be fluence attenuation with depth, but the normalized spectrum of a target located within the tissue will be measured correctly at all depths.

When tissue scattering and absorption are wavelength-dependent functions, this immediately reflects on their depth dependences, i.e. depth dependence of laser fluence becomes different for different wavelengths.

When the dynamic range of fluence variation with depth is reduced, the uncertainty of the target's spectral dependence is also reduced. An optimized illumination, which can be done for small animal PA imaging, can help minimize the dynamic range of laser fluence variation with depth. This statement is clearly demonstrated by the simulation below.

To prove it either way would need a simulation, similar to that in Figure 3R but for two wavelengths for both single-sided and multi-sided illumination geometries*. That said, interesting as this point may be, since the claim in question is not explicitly made in the manuscript it is incidental now – the matter only arose in trying to understand the reasoning behind a now corrected questionable claim in a previous version of the manuscript.

We performed an additional simulation to demonstrate our statement. In the geometry shown in Figure 3R of the previous review, we positioned a target at a 1 cm depth in both cases of tissue illumination: uniform multi-sided illumination (applicable for small animals) and single-sided illumination (clinically applicable for humans). We calculated the spectrum of the target positioned at a 1 cm depth inside a mouse (in the middle of cylinder, Fig 3R) and compared it with the ground-truth spectrum and with the spectrum of a the same target located within a 'human' (single-sided illumination) at the same, 1 cm depth. Figure 1RR clearly shows that the spectrum is disrupted less for the "mouse" compared to the "human".

Note that we used very low contrast in light absorption between 715 nm and 875 nm wavelengths, setting the light absorption as a linear function between 0.05 cm^{-1} for 715 nm wavelength and 0.3 cm^{-1} for 875 nm wavelength. Light scattering was assumed to be constant in the entire wavelength range for simplicity and equal to 3 cm^{-1} . The larger the wavelength dependence in optical properties of the background material, the more pronounced will be the difference between illumination geometries and the larger inaccuracy will be in the measured target's

absorption spectrum. Again, all-sided illumination smooths the light distribution within the target and reduces wavelength-dependent inaccuracy in the target's spectrum PA reconstruction.

We believe that the simulation below should end the discussion. Although the differences between the two geometries are small, they are significant and will grow much larger with increasing variation in wavelength dependent optical properties. Clearly, PA methods optimized for mice cannot easily be transferred to clinics.

At the same time, we also mention that fluence compensation and motion correction are important for pre-clinical studies in mice as well:

“Although compensating for wavelength-dependent light fluence is important even for the case of uniform **multi-sided** illumination, this is a much more serious problem for a single-sided imaging geometry, which is typical of the clinical environment.”

Figure 1RR. Geometry of single-sided (a) (“human” case) and (b) all-sided (“mouse” case) illuminations. (c) and (d) - Color-encoded optical fluence depth profiles for cases (a) and (b) respectively. (e) – Optical fluence at 10 mm depth as a function of background absorption (corresponding wavelength is shown below) in cases (a) (red curve) and (b) (blue curve). (f) - Normalized target’s spectrum measured photoacoustically at the 10 mm depth: ground truth spectrum (black curve), single-sided illumination (red curve) and all-sided illumination (blue curve). Reduced light scattering was assumed to be wavelength-independent and equal to 3 cm^{-1} ; light absorption changed linearly from 0.05 cm^{-1} at a 715 nm wavelength to 0.3 cm^{-1} at an 875 nm wavelength.

***Perhaps this has been done ? It is claimed that at the center of the cylinder, the ratio of fluences at different wavelengths for mouse and humans are different - 2dB versus -10dB - but no information is provided as to how these figures were estimated and what assumptions were made. It is not clear what the “human” circumstances of the simulations were. Illumination from all sides but with different optical properties as the “mouse” or same optical properties but single sided illumination or some combination of both ? Without this information it is not possible to assess this analysis.**

“Human” circumstances correspond to single-sided illumination as we mention throughout the entire paper and illustrate in Figs. 1 and 2 of the paper.

5. It is stated in several places that illuminating a mouse from all directions can distribute the light in the “background almost uniformly”. If this is intended to suggest that the fluence in the center of a mouse is similar to that close to the surface, then it seems implausible. Illuminating from all sides can reduce the depth dependence of the fluence, compared to single side illumination but can it really be almost constant with depth in a real mouse in vivo ? Prior work using multi-sided illumination does not support this with higher contrast evident at the surface compared to deeper regions – eg see references 59 and 39. It is recommended this claim is modified.

We modified the claim as suggested by the Reviewer. Now it says “Illumination from all directions **reduces the depth-dependence of laser fluence.**”

In another sentence - “**Illumination from all directions distributes light throughout the object ...**”

6. The work described in reference 59 on spectroscopically estimating blood oxygen saturation in mice clearly shows that accounting for the wavelength dependence of the fluence is essential even with multi-sided illumination; when the fluence wavelength dependence was ignored (as is the case when the linear unmixing method was used), the oxygen estimates exhibited very poor accuracy. This further undermines the suggestion that spectroscopic methods are much less challenging in mice because the fluence wavelength dependence is less of a problem. Indeed the entire mice vs human debate in the current work seems largely irrelevant and it is somewhat baffling as to why it features so prominently. All discussion of mouse imaging could be removed entirely from the manuscript without diminishing the work in the slightest – in fact it would improve the

clarity of the manuscript as this issue serves only to distract. The current system is an advance towards clinical photoacoustic imaging because it is a convenient fast hand-held probe suitable for human use. It also provides fluence compensation and this is very useful but that is not what makes it especially relevant for human use specifically as this feature is also needed for spectroscopic mouse imaging. That is OK - there is a still perfectly valid justification for authors' system without having to contrive an argument that fluence correction is needed more in humans than mice.

As we showed in Fig.1RR above, the spectral distortions are much more prominent for single-sided illumination, rather than for the all-side illumination, which can be optimized for mice. However, we softened our statements regarding the comparison of mice vs human.

“Although compensating for wavelength-dependent light fluence is important even for the case of uniform **multi-sided** illumination, this is a much more serious problem for a single-sided imaging geometry, which is typical of the clinical environment.”